# Clustering analysis of the *Sargassum* transport process: application to beaching prediction in the Lesser Antilles

Didier Bernard[1], Emmanuel Biabiany[2], Raphaël Cécé[1], Romual Chery[1], and Naoufal Sekkat[1]

[1]LARGE, University of the French West Indies, 97157 Pointe-à-Pitre, Guadeloupe, France.

[2]LAMIA, University of the French West Indies, 97157 Pointe-à-Pitre, Guadeloupe, France.

*Correspondence to:* Didier Bernard (didier.bernard@univ-antilles.fr)

**Abstract.** The massive *Sargassum* algae beachings observed over the past decade are the new natural hazard that currently impacts the island states of the Caribbean region (human health, environmental damages, and economic losses). This study aims to improve the prediction of the surface current dynamic leading to beachings in the Lesser Antilles, using clustering analysis methods. The input surface currents were derived from the Mercator model and the Hybrid Coordinate Ocean Model (HYCOM) outputs in which we integrated the windage effect. Past daily observations of *Sargassum* beaching on Guadeloupe coasts and satellite-based *Sargassum* offshore abundance were also integrated. Four representative current regimes were identified for both Mercator and HYCOM data. The analysis of the current sequences leading to beachings showed that the recurrence of two current regimes is related to the beaching peaks observed respectively in March and in August. The performance score of the predictive model showed that the HYCOM data seem more suitable to assess coastal *Sargassum* hazard in the Lesser Antilles. For one year of tests (i.e. 2021), the decision tree accuracy reached respectively 70.1% and 58.2% for HYCOM and Mercator with a temporal uncertainty range +/- 3 days around the forecast date. The present clustering analysis predictive system requiring lower computational resources compared to conventional forecast models would help improve this risk management in the islands of this region.

## 1 Introduction

During the periods 2011-2012, then 2014-2019, massive *Sargassum* beachings impacted most coasts of the Lesser Antilles (LA), mainly those facing east and southeast (Franks et al., 2012, Gower et al., 2013, Johnson. et al., 2014, Hu et al., 2016, Wang and Hu, 2016). The LA received large amounts of algae on the windward Atlantic coastline, while leeward Caribbean coastal areas remained slightly affected (Maréchal et al., 2017). These beachings in terms of frequency and intensity can now be considered as a new natural hazard for the Caribbean islands and American coasts.

Indeed, while it has been demonstrated that *Sargassum* algae provide ecosystem services, habitat and shelter for various organisms in a structurally sterile ocean ecosystem (Witherington et al., 2012; Bertola et al., 2020), the beachings over the past decade have induced health risks for the population and have had considerable socio-economic impacts (Franks et al., 2012). For example, when looking at the French West Indies, the Guadeloupe archipelago and Martinique, the findings are as follows:

(1)   Apart from 2013, the recent inflow of *Sargassum* rafts on the coasts of Guadeloupe and Martinique, although irregular,

has not ceased since 2011, reaching a paroxysm in 2015 (Florenne et al., 2016; Berline et al., 2020). State services estimated that the volumes stranded on the shores were in the order of 1.5 million m³, from October 2014 to October

2015 in Guadeloupe (Florenne et al., 2016). Only a third of these could be collected by the authorities and priority was given to areas at stake, such as inhabited areas, shores with economic or tourist activities and ecosystems or other environmental niches. The particularity and the difficulty lay in the fact that 60% of this coastline and/or of the volume stranded remained inaccessible to the techniques currently proven and at costs currently bearable.

(2) There was an impact on human health and ecosystems because in shallow and small bays, the accumulated algae degrade by fermentation and emit chemical compounds such as hydrogen sulfide ($H_2S$) and ammonia ($NH_3$) (Anses, 2017, Van Tussenbroek et al., 2017; Resiere et al., 2018).

(3) The survey conducted by the organizations responsible for socio-economic development estimated that the decline in tourism resulted in an economic loss of $5.5 million for the first half of 2015 (https://eos.org/features/*Sargassum*-watch-warns-of-incoming-seaweed).

The volumes to be collected were considerable compared to the size of these islands (< 1200 km$^2$ each) and the vulnerability of these territories. This new phenomenon has raised several scientific questions relating to *Sargassum* rafts, such as their transports, origins, the sources of nutrients promoting their growth but especially the physical factors that led to their occurrence and their development in the tropical and equatorial Atlantic.

Using large-scale observations with ocean color satellite remote sensing, historical hydrographic observations, time series of *Sargassum* volume collected on ships, multi-year reanalysis of wind and current, and numerical models; the roles of both subsurface nutrient supply and surface current transport were estimated. Several authors have contributed to the understanding of the mechanisms and physicochemical processes governing the phenomenon (Gower et al., 2006; Gower and King, 2011; Gower et al., 2013; Maréchal et al., 2017; Johns et al., 2020). Operational systems have been developed such as the satellite based *Sargassum* Watch System SaWS (Hu, 2009; Hu et al., 2015) and the *Sargassum* Early Advisory System (SEAS) (Webster and Linton, 2013). They provide a temporal and spatial assessment of annual seasonal increases and decreases in *Sargassum* algae amount over wide areas of the tropical Atlantic and Caribbean (Wang and Hu, 2016; Wang and Hu, 2017; Wang et al. 2019). Time series from remote sensing were coupled with spatial distribution models to determine the mechanisms that aggregate *Sargassum* algae along a zonal band in the tropical Atlantic considering possible nutrient sources promoting the observed annual blooms (Wang et al., 2018; Wang et al., 2019; Johns et al., 2020; Jouanno et al., 2021b).

Tropical Atlantic currents and winds seasonally aggregate and carry these algae towards the Caribbean (Franks et al., 2016; Brooks et al., 2018; Cuevas et al., 2018). Modelling studies mainly focused on the transport properties of *Sargassum* rafts by offshore currents (Wang and Hu, 2017; Brooks et al., 2018; Maréchal et al., 2017; Putman et al., 2018, 2020; Wang et al., 2019; Berline et al., 2020).

Johns et al. (2020) extended this analysis to highlight anomalous transport due to the 2009-2010 NAO anomaly and seasonal aggregation by the Inter Tropical Convergence Zone (ITCZ).

A combination of satellite-based Alternative Floating Algae Index (AFAI, Wang and Hu, 2016) fields with Hybrid Coordinate Ocean Model (HYCOM) surface current forecast data were used by Maréchal et al. (2017) to short-term predict *Sargassum* beachings for Guadeloupe and the French Antilles islands. Maréchal et al. (2017) showed that this short-term prediction system (i.e.

detection starting within 50-100 km of the coasts) worked efficiently during the year 2015 with a performance percentage of 62% and a beaching forecast date uncertainty below one day.

Trinanes et al. (2021) presented the *Sargassum* Inundation Reports (SIR), a product based on satellite observations to weekly predict *Sargassum* coastal inundation potential throughout the Caribbean Sea region, the Gulf of Mexico, and extending to the east coast of
Florida and the Bahamas. As described by Trinanes et al. (2021), the SIR algorithm uses the Floating Algae density values within 50 km of each coastal pixel to predict three inundation potential levels (low, medium, and high). This algorithm does not include ocean currents, winds, and waves which may modify the movement of *Sargassum*.

In the above works, the implementation of methods based on several independent data sets has led to the production of scientific knowledge and even to the development of large-scale forecasting systems. None of them used predictive modelling (Geisser, 1993;
Kuhn and Johnson, 2013) including classifiers (Friedl and Brodley, 1997) to determine the probability of repeatable patterns in a dataset so as to produce a decision for risk prevention managers. Predictive modelling refers to mathematical and computational methods of predicting future events based on the analysis of the repeatable patterns in the input dataset (Geisser, 1993; Friedl and Brodley, 1997; Kuhn and Johnson, 2013). Compared to other conventional forecasts, predictive modelling methods require low computational costs and are characterized by their flexibility, and their intuitive simplicity (Friedl and Brodley, 1997).

In this paper, we propose to use clustering and decision tree classifier methods, combining ocean surface current, wind reanalysis, an satellite-based *Sargassum* offshore abundance with past observed beachings to obtain a first predictive model of *Sargassum* beaching on the Caribbean coasts. This model will be used with forecast data as input to produce an operational decision support system.

As ocean data are spatio-temporal fields, machine learning methods such as K-Means (KMS) may be used to obtain a finite number
of possible k-cluster partitions of the surface currents. These methods have been widely used in weather forecasting (Michelangeli et al., 1995; Cassou et al., 2004; Boé and Terray, 2008) but are much less common in physical oceanography (Harms and Winant, 1998; Hisaki, 2013).

We focused on the offshore region covering either side of the Lesser Antilles, between 55-66°W and 8-17°N (Fig. 1a). Visual analysis of the monthly SaWS maps indicates that this region remains the primary pathway for *Sargassum* rafts from the Atlantic
Ocean to the Caribbean Sea. The North Equatorial Current (NEC), the Guiana Current (GC), the eddies and the retroflection front of the North Brazil Current (NBC) are the main contributors of this transport. Putman et al. (2018) modelled the percentage of *Sargassum* which follows these routes. Figure 1b describes the focused area divided into a first sub-set "LA1" for the Caribbean Sea, a second one, "LA2" between 18°N and 14.5°N (Guadeloupe, Dominica, Martinique, Saint Lucia) and a third one "LA3" south of 14.5°N (Saint Vincent, Barbados, Trinidad and Tobago).

The questions are as follows. Can dynamic patterns of surface currents in the Lesser Antilles be summarized as a discrete set of cases? What is their temporal recurrence? What combinations of currents enhance *Sargassum* rafts arrival and beachings on the Lesser Antilles coasts? What is the contribution of this type of predictive modelling to the prevention of this new natural hazard?

The overall methodology, database, clustering methods and decision tree used in this study are described in Sect. 2. The obtained current regimes, their relationship to *Sargassum* hazard and the decision support system performances are presented in Sect. 3.

These results are discussed in Sect. 4.

**2 Datasets and methods**

The overall methodology is presented in Fig. 2. The main goal of the first step was to use clustering analysis to identify the main current patterns in the Lesser Antilles during the period 2019-2020. The 30-day current patterns sequences leading to beachings were deduced based on beaching observations in Guadeloupe. An additional clustering analysis was conducted on these sequences

to study the main patterns (orange box in Fig. 2). A decision tree classifier was built with the following input data: current patterns, 30-day sequences before beaching, satellite-based Sargassum abundance offshore Guadeloupe, surface currents from HYCOM (HYCOM GLBy0.08 version) and Mercator (PSY4V3R1 Mercator 1/12-degree 3D analysis). This decision support system was tested for the period of the year 2021. The performance scores were assessed for each decision day and three temporal uncertainty ranges around this day, respectively: +/-1 days, +/-2 days, +/-3 days.

**2.1 HYCOM surface current dataset**

Daily (12 UTC, i.e. Coordinated Universal Time) surface current components from the 41-layer Hybrid Coordinate Ocean Model (HYCOM) at 1/12-degree, global analysis (HYCOM GLBy0.08 version, available at: https://www.hycom.org/data/glby0pt08/expt-93pt0, last access: 17 January 2022), were examined. The HYCOM surface forcing including 10-m wind velocities are extracted from Climate Forecast System Version 2 (CFSv2). The Navy Coupled Ocean Data Assimilation (NCODA) system is used to

assimilate available observational data: satellite altimeter sea surface height, satellite and in-situ sea surface temperature, temperature vertical profiles and salinity vertical profiles (Cummings, 2005; Cummings and Smedstad, 2013; Helber et al., 2013). The bathymetry used is the GEBCO8 (Becker et al., 2009) with 30 arc second of resolution. The HYCOM GLBy0.08 grid resolution is 0.08 degree in longitude and 0.04 degree in latitude. To perform the present study, the native HYCOM fields were first interpolated on the Mercator uniform lon/lat 0.08-degree grid with a bilinear method. Putman et al. (2018) and Johns et al. (2020) used a previous

version of HYCOM model including uniform lon/lat 0.08° scale grid to successfully simulate *Sargassum* trajectories.

**2.2 Mercator surface current dataset**

The daily (12 UTC) surface current components from the 50-layer PSY4V3R1 Mercator 1/12-degree 3D analysis system (Lellouche et al., 2018; Gasparin et al., 2019) were also analyzed (PSY4V3R1 Mercator 1/12-degree 3D analysis, available at: https://resources.marine.copernicus.eu/product-detail/GLOBAL_ANALYSIS_FORECAST_PHY_001_024/DATA-ACCESS, last

access: 17 January 2022). The atmospheric surface forcing is extracted from the 3-hourly ECMWF (European Centre for Medium-Range Weather Forecasts) IFS (Integrated Forecast System). This version of Mercator model includes assimilation of observational data quite similarly to HYCOM NCODA system (i.e. satellite altimeter sea surface height, satellite and in situ sea surface temperature, temperature vertical profiles and salinity vertical profiles). Unlike the HYCOM GLBy0.08 native grid including higher resolution in latitude (i.e. 0.04 degree), the Mercator native grid is uniform in longitude and latitude with 0.08-degree horizontal

grid resolution. This would suggest that HYCOM may better reproduce small scale patterns than Mercator. Moreover, as described by Lellouche et al. (2018), the Mercator bathymetry includes GEBCO8 data in regions shallower than 200 m and the coarse 1 arc-

minute ETOPO1 data (Amante and Eakins, 2009) in regions deeper than 300 m. The complex bathymetry of the Lesser Antilles Arc studied here could be less realistic in the Mercator than in the HYCOM fields.

## 2.3 ERA-5 dataset: surface winds

Surface wind influences the transport of floating seaweed rafts and a drag or windage coefficient must be added to the surface currents. Daily 12 UTC fields from the hourly 31-km horizontal resolution ERA-5 reanalysis dataset (Hersbach et al. 2020) was used (ERA-5 reanalysis, available at: https://cds.climate.copernicus.eu/cdsapp#!/dataset/reanalysis-era5-pressure-levels, last access: 17 January 2022). The wind data was integrated with Mercator and HYCOM ocean currents data following this formula:

$$u_s(x, t) = u_m(x, t) + C_w u_w(x, t) \tag{1}$$

where $u_s$ represents the oceanic surface currents with windage, $u_m$ the oceanic surface currents velocity, $C_w$ the windage and $u_w$ the surface winds velocity. This approach is consistent with Putman et al. (2018) and Johns et al. (2020) studies. The value of $Cw = 0.01$ was used, following Putman et al. (2018), Johns et al. (2020) and Berline et al. (2020). The use of other windage values should be investigated in a further study.

## 2.4 Beaching observational data (Guadeloupe)

A referencing database including observed beachings on Guadeloupe coasts was used in the present study. The selected time period is the same as the one for surface current data: from 1st January 2019 to 31 December 2020. During this period of 730 days, only 110 days of *Sargassum* beaching were recorded (i.e. 30 days in 2019 and 80 days in 2020). During the year of 2021, 78 days of beaching were observed in Guadeloupe. These observational data based on remote sensing and in situ data are archived online by the Regional Directorate for Environment, Development and Housing in Guadeloupe (http://www.guadeloupe.developpement-durable.gouv.fr/sargasses-r999.html).

## 2.5 Satellite-based offshore abundance of *Sargassum*

*Sargassum* satellite observations were included in the present decision support system. To quantify the abundance of *Sargassum* in an area of 100 km radius offshore Guadeloupe, the 7-day Floating Algae (FA) density fields derived from the Alternative Floating Algae Index (Wang and Hu, 2016) were analyzed. As described by Trinanes et al. (2021), the 7-day Floating Algae (FA) density fields are accumulated on 7 days and have a 0.1° resolution. Due to optical complexity in nearshore waters, the FA density fields are masked with missing values within 30 km from shoreline (Trinanes et al. 2021). The cumulative FA density values were added up in the area 30-100 km offshore Guadeloupe (Fig. 1) then averaged over the two years 2019 and 2020 for each day.

## 2.6 Clustering analysis with expert distance

Unsupervised learning methods such as Hierarchical Agglomerative Clustering (HAC) and K-means algorithms were used in the present study. The Ward method allows to identify homogeneous subsets of data (Ward, 1963). Besides the measures and the classes of distance between objects such as the Euclidean distance for K-means and the Ward method, a new metric was also added. The Expert Distance (ED) which integrates image analysis within unsupervised learning methods (Clustering) was used. This method allowed significant improvement in clustering analysis dealing with climate data characterized by high spatio-temporal variability, such as precipitation (Biabiany et al., 2020). Clustering methods using Euclidean distance (L2) can lead to group different physical

situations within the same cluster (Biabiany et al., 2020). The ED metric integrates a set of knowledge about the dynamics of the data to be partitioned as well as their spatio-temporal properties.

This ED is based on an empirical spatial subdivision and the use of Kullback-Leibler divergence, in order to quantify the similarity between two fields. Figure 3 shows the schematic of the Expert Distance process adopted here.

The LA study area was separated into three parts (Fig. 1b, Fig. 3) based on the *Sargassum* rafts transport centers of action reported in the literature (Franks et al., 2016; Berline et al., 2020). To the west of LA, the first zone, LA1, is centered on the Caribbean Sea. To the east, the Atlantic zone was split into two areas towards 13.5°N, just above the island of Barbados. To the south-east is the LA3 zone under the influence of the North Equatorial Recirculation Region (NERR) and its retroflection rings, while to the north-east is the LA2 zone, more representative of the North Equatorial Current. The analyzed daily fields include a total of 14 279 grid points (4 282 grid points in LA1, 3 407 grid points in LA2 and 4 536 grid points in LA3). The remainder corresponds to areas over land (e.g., islands).

The clustering results were evaluated using the Silhouette Index (Rousseeuw, 1987). The Silhouette (SaMk) index defined in Biabiany et al. (2020) was used. This allows to express the quality of a clustering, by the average of the quality of each cluster, which is itself the average of the silhouette indices *s(i)* over the cluster elements. This index is defined as follows:

$$SaMk = \frac{1}{k} \times \sum_{j=1}^{k} \frac{1}{|C_j|} \times \sum_{i \in C_j} s(i) \tag{2}$$

where $k$ is the number of clusters, $C_j$ the set of days from the cluster $j$, $i$ a day form $C_j$ and $s(i)$ the silhouette index (Rousseeuw, 1987) value of day $i$.

The current pattern clusters obtained are related by a set of days in common. Match percentages were calculated using the following formula.

$$p_{(m,h)} = \frac{|C_m \cap C_h|}{|C_m \cup C_h|} = \frac{N_{(m,h)}}{|C_m| + |C_h| - N_{(m,h)}} \tag{3}$$

where $p_{(m,h)}$ is the percentage of correspondence between cluster $C_m$ and cluster $C_h$ derived from Mercator and HYCOM datasets respectively. $N_{(m,h)}$ is the number of days shared by these two clusters.

## 2.7 Clustering analysis on current sequences leading to beachings

To better understand current dynamics which may lead to *Sargassum* beaching in Guadeloupe, we analyzed the 30-day current sequences before beaching. The 30 days duration corresponds to the empirical transport time of a passive particle moving from the main entrance location of *Sargassum* rafts in the Lesser Antilles area (i.e. in LA3 zone, 8° N; -55° E) to Guadeloupe (i.e. LA2 zone). Based on the mean current magnitude of 0.2 m s$^{-1}$ (average value over the LA zone, in HYCOM and in Mercator data) and the distance of 500 km between the main entrance location and the Guadeloupe coasts, 29 days are obtained for the transport. For simplicity, the duration of 30 days was selected instead of 29 days. While 110 observed beaching days were registered between January 2019 and December 2020, only 107 sequences were studied here. This is explained by the fact that beaching days registered in January 2019 were removed to avoid the sequences missing data of the December 2018 period. These 107 beaching sequences were examined with HYCOM fields. Dissimilarities between these sequences were calculated before dividing the sequences dataset into several groups using a hierarchical classification (Larmarange et al., 2015). The Longest Common Subsequence (LCS) method

was used to compute the distances between the sequences (Elzinga and Struder, 2015; Studer and Ritschard, 2016). A dendrogram was calculated using Ward's algorithm (Ward, 1963). The highest relative inertia loss criterion allowed to determine the optimal number of partitions (TraMiner package (Gabadinho et al., 2011)). The stages of this clustering process were summarized in Fig. 4.

**2.8 Decision support system**

To determine the probability of *Sargassum* beaching, a decision tree was built using complementary elements called "modules" (Fig. 5). Each module generates information based on input data including surface currents (Mercator and HYCOM) with windage effects (ERA-5) and past observations of beachings in Guadeloupe. Thus, for a given day, the proposed system works as follow:

- Module A takes as input the week number of the selected day and returns the associated daily probability to reach the maximum offshore abundance of *Sargassum* (based on observational FA density values during the two years 2019 and 2020);

- Module B assigns a cluster number to the focused day after the ED clustering of the daily surface currents. Then, from this day, it builds empirical sequences of numbers between 1 and 4 (type of cluster) over a period of 30 past days;

- Module C takes as input the daily cluster number produced by module B and returns the probability (frequency) of beaching associated with the type of cluster. This probability is calculated, by cluster type, from the beachings observed on the coasts of the Guadeloupe archipelago. The system has 107 30-day current sequences before beaching. These sequences start on the day of beaching on the coasts of Guadeloupe. This set of referenced 30-day current sequences before beaching is called BASE (Fig. 5b);

- Module D compares the sequence of the given day to the referenced current sequences before beaching with Jaccard distance. Module D is interconnected to BASE and module B. It returns the percentage of correspondence between them.

In the literature, the average of the different modules is often used as the decision operator (Bo. et al., 2020; Swain and Hauska, 1977). In the present work, the percentage of beaching for a given day was determined using the percentages provided by modules A, C and D, according to the following formula:

$$P(i) = (A(i) + \frac{C(i)}{D(i)}) \tag{4}$$

where P(i) is the quantity used in the design of the decision rule. This rule is simply the linear combination of the percentages from modules A, C and D, calculated according to:

$$DECISION(i) = P(i) > Mean(P(j)) \tag{5}$$

where $j \in R$. The set of past days (2019-2020) and DECISION(i) is a (logical) response of the decision tree for a given day i.e. expressed in binary form.

The proposed tree in Fig. 5 was tested on the full year of 2021 except 31 December 2021 which was not included because of missing data, giving a total of 364 tests. The performance assessment of the decision support system was assessed for each decision day and three temporal uncertainty ranges around this day, respectively: +/-1 days, +/-2 days, +/-3 days.

**3 Results**

**3.1 Surface current patterns in the focused area**

In view of the lack of studies dealing with surface current patterns in the Lesser Antilles area, this preliminary analysis is presented here. The deciles of surface current velocities including windage, are presented in Table 1. The maximum surface velocity reaches

2.49 m s$^{-1}$ and 2.57 m s$^{-1}$, respectively for HYCOM and Mercator. For both models 90% of the velocity values remain below 0.65 m s$^{-1}$ (the respective 90th centile values are respectively 0.6515 m s$^{-1}$ and 0.6458 m s$^{-1}$ for HYCOM and Mercator). The Mercator data have a median of 0.28 m s$^{-1}$, the mean of 0.33 m s$^{-1}$, while for HYCOM these values are respectively equal to 0.32 m s$^{-1}$ and 0.36 m s$^{-1}$. The ratio between the first and the last decile is close to 6. Figure 6 shows skewed distributions with skewness equal to

1.31 and 1.21 for HYCOM and Mercator, respectively. A Gaussian kernel was applied to obtain these distributions. The distribution mass is concentrated on the left.

To assess the contribution of each of the three regions (i.e. LA1, LA2, LA3) to the deciles, the relative frequency against the decile thresholds given in Table 1 is shown in Fig. 7. Three different shapes can be seen. In the Caribbean Sea, the LA1 relative frequency distributions from HYCOM and Mercator are almost horizontal, indicating a quite constant contribution (~3%) over all velocity

classes. In the Atlantic Ocean (i.e. in the area including LA2 and LA3), HYCOM and Mercator current speed distributions are quite similar. The frequency distributions show two opposite behaviors respectively for LA2 and LA3. In the Atlantic north LA part, LA2 area, the frequency decreases with current speed. The current speeds above 0.65 m s$^{-1}$ are very uncommon. On the contrary, in the Atlantic south LA part, LA3 area, the frequency increase is observed with maximum frequency linked with current speeds above 0.65 m s$^{-1}$. These three significant specific current speed distributions associated with LA1, LA2 and LA3 confirm the need to

separate these three areas in the ED metric clustering process.

The differences between HYCOM and Mercator current vectors were also examined for each grid point (Fig. 8). Globally, at open sea, the current speed differences are relatively small and remain below 0.15 m s$^{-1}$. These differences between HYCOM and Mercator increase close to the islands with an average value of 0.3 m s$^{-1}$. In the South part of the LA arc, around Trinidad and Tobago, Mercator current magnitudes are globally higher than HYCOM current magnitudes. Thus, Mercator surface currents might

induce higher *Sargassum* influx from the Western Central Atlantic to the Caribbean Sea in this area.

At each grid point, the angular deviations found between the medians of the surface current velocity vector directions can be divided into three magnitude groups of 45° intervals. The current direction differences between 0 and 45° are the most frequent group in the region, while those between 45 and 90° remain localized downstream of the islands. Finally, those above 90° occur exclusively around Trinidad.

**3.2 Clustering analysis**

To identify surface current patterns in the region, and then those that lead the transport of *Sargassum* rafts to the LA islands coasts, the clustering of the gridded data according to equation (1) was performed.

**3.2.1 Clustering assessment**

One of the known uncertainties in the k-means method is induced by the selected number of clusters. To find an optimal number of

clusters and identify the best partition (Biabiany et al. 2020), the silhouette index (SaMk) evolution against the number of clusters, k. is shown in Fig. 9. The silhouette indices obtained by the KMS-ED method, are in general above 0.2 for any k<15, and remain higher than those obtained by the KMS-L2, HAC-L2 and HAC-ED methods. These values indicate that the quality of the clusters is much better with the KMS-ED method. The inflection point of the KMS-ED curve occurs for the same number of clusters, k=4,

for both Mercator and HYCOM data. This highlights four representative current regimes in the studied region, respectively named MC1, MC2, MC3, MC4 for Mercator and HC1, HC2, HC3 and HC4 for HYCOM.

### 3.2.2 Visual analysis of current regimes

The four types of surface current circulation, obtained in intensity and direction, are shown in Figs. 10 and 11, respectively for the Mercator and HYCOM analysis. The parangon which is the closest day to the centroid, was chosen to represent each type of cluster. The four clusters may be distinguished by the NBC expansion and by the induced retroflection ring locations. The surface current velocities and their associated streamlines are driven by the following structures:

- those which enter through the Caribbean Sea from the south, remaining almost parallel to the continental shelf. They occur in the LA3 and LA1 regions;
- Those due to the propagation of the eddies dynamic characteristics related to the retroflection rings of the NBC. They are coming from the south of the LA3 region, along the Atlantic side of the Lesser Antilles arc, before passing through the Caribbean Sea towards 12-14° N;
- Those generally coming from the northeast of the LA1 and LA2 regions, representing the southern limit of the subtropical gyre which cuts the Lesser Antilles at about 15° N. They keep their initial direction and are sheared by the South-East currents.

The number of days corresponding to each cluster is given in Table 2. MC1, HC2 and HC3 are the most common along the studied period. Each of them represents almost 30% of the daily output. However, none of the four clusters really stands out. For both analyses, the differences between cluster occurrences stay lower than 10%.

### 3.2.3 Matching days between clusters

The clusters found are also related by a set of days in common. Match percentages were calculated using the following Eq. 3. Table 3 shows the results of the match percentages (Eq. 3). MC4 - HC2 is the cluster pair with the highest match score (69.8%). It is followed by the pair MC2 - HC1 (60.4 %), then MC3 - HC3 (56.7 %) and MC1 - HC4 (50,6%). The cluster numbering does not take into account these match percentages (e.g. MC1 and MC2 main patterns respectively differ from HC1 and HC2 patterns).

### 3.2.4 Distribution and comparison of intensities

Deciles were used to study and analyze the velocity distributions characterizing each cluster. Evolutions of the relative frequency of Us(x,y,t) as a function of the deciles (Table 1) are shown in Figs. 12 and 13. For the entire analysis, the values of the deciles remain fixed and constant, and the curves are plotted for the three regions described in Fig. 1.

For both models, globally, three main patterns are identified. The first pattern includes the following clusters: MC1, MC3, HC1 and HC3. This pattern is characterized by the increase of the relative frequency curve in the LA1 and LA3 regions and its decrease in the LA2 region. The elements of these clusters include strong current velocities above the median of 0.28 m s$^{-1}$. The second pattern includes MC2 and HC2 clusters which are characterized by the decrease of the relative frequency for the three regions (i.e. LA1, LA2, LA3). The last pattern includes MC4 and HC4 clusters and corresponds to three concave curves with maximums located at different velocity thresholds depending on the region under study.

To examine possible relationships, for a given region, between the two variables, decile speed thresholds and identified clusters, contingency tables were constructed (not shown) and the chi-squared test was performed. For the three areas, the p-value was much lower than 0.01. The chi-squared test results indicated that for the LA1, LA2 and LA3 regions, the speed distribution depends on the identified cluster.

### 3.2.5 Seasonality

The monthly distribution of each cluster is plotted (Figs. 14 and 15). Differences are relatively clear for both model analyses with a marked seasonal variation. The MC3 and HC3 regimes are observed during the first half of the year with a maximum in March, followed by MC2 and HC1 from April to July. The last two regimes are observed from August to December. The pair MC4 HC2, reaches a maximum in September while MC1 and HC4 persist until February of the following year.

### 3.3 Links with *Sargassum* beachings

As with many floating objects, before coming ashore on the coasts of the LA, *Sargassum* algae accumulate on the ocean surface in large amounts and form slicks, or filamentary structures, interspersed with void areas, under the influence of currents. These dynamic structures regularly observed from satellites, aircraft, and ships, have a certain inertia (Maximenko et al., 2012).

Beyond biological production, it is therefore the specific dynamic conditions of the surface currents and the surface winds which may lead to massive *Sargassum* beachings on Caribbean coastal areas.

The monthly evolution of observed stranding days on the Guadeloupe coasts, the monthly evolution of *Sargassum* abundance in the area 30-100 km offshore Guadeloupe were also analyzed on the focused period 2019-2020 (Figs. 14 and 15). During these two years, the amount of *Sargassum* likely to enhance the beaching risk in Guadeloupe increased significantly from February to May, then decreased from May to November.

Two beaching peak values are found: one in March and the second in August. The beachings dates and the cluster occurrence dates were also compared in Table 4. The MC3 - HC3 pair gather the greatest number of similarities, followed by the MC1 and HC2 clusters.

These pairs of clusters would be favourable to the transport of these algae toward the coasts of the Lesser Antilles islands. MC2 and HC1 are the two clusters with the smallest number of beaching days.

### 3.4 Current sequences leading to beachings

The HAC clustering analysis on the current regime sequences leading to observed beaching days allowed to distribute the 107 sequences into four classes, respectively called Seq1, Seq2, Seq3 and Seq4. This analysis integrated only the HYCOM surface current data which have a greater resolution than Mercator. During the focused period (i.e. 2019-2020), Seq4 (39.3%) and Seq2 (37.4%) have the greatest occurrence (Table 5). Seq1 and Seq3 have a respective occurrence of 16.8% and 6.5%. Figure 16 shows that Seq2, Seq3, and Seq4 are characterized by the respective modal current regimes HC3, HC1, and HC2. For the Seq1 sequences, there is no clear prevalent current regime. The monthly distribution of the main sequence classes, Seq2 and Seq4, highlights a significant seasonal splitting (Fig. 17). The Seq2 sequences occurred from December to June while the Seq4 ones occurred from July to November. These two distributions also seem significantly correlated with the monthly occurrences of observed beachings. While the first beaching peak occurring in March is linked with the Seq2 maximum occurrences, the second beaching peak occurring

in August is linked with the Seq4 maximum occurrences.

## 3.5 Decision support system results

The behavior of each module is presented in Fig. 18. Globally, Module A probabilities based on *Sargassum* offshore abundance probability seem the most correlated with the DSS decision. However, during the months with low *Sargassum* offshore abundance (i.e. from September to December) Modules C and D related to current patterns are the main contributors to the decision. Module D based on the comparison between past observed sequences and the sequence corresponding to the forecast day remains with high probabilities above 0.5. These probabilities can reach 0.95 indicating strong similarities between the sequences. Module C associated with the percentages of beaching per cluster shows empirical probabilities close to 0.3 indicating that one third of the days in the concerned clusters are beaching days.

Table 6 presents the performance results of the predictive model (i.e. clustering + decision tree) for Mercator and HYCOM. "True positive/negative" respectively refer to the number of observed beaching/non-beaching days, predicted by the decision system. "Recall" refers to the ratio in percentage between these respective numbers of days and the total number of tests (i.e. 364 days). "Accuracy" corresponds to the number of days with a true prediction and its ratio in percentage was computed over the total number of tested days. Overall, the use of HYCOM data allows to improve the prediction of beaching and non-beaching days (Table 6). With forecast date uncertainty below one day, the HYCOM DSS has an accuracy of 54.1% (i.e. beaching and non-beaching days) and predicts 59.0% of the observed beachings (i.e. true positive) in the year 2021. At the same date precision, the Mercator DSS has an accuracy of 50.6% and predicts 55.1% of the observed beaching days. The performance differences between the two datasets tend to increase with the temporal uncertainty ranges around the forecast date. With a temporal uncertainty range +/-3 days, the HYCOM DSS reaches an accuracy of 70.1% and predicts 73.1% of the observed beachings in the year 2021. At the same date precision, the Mercator DSS presents an accuracy of 58.2% and predicts 65.4% of the observed beachings.

## 4. Discussion

### 4.1. Performance indices and clustering quality

The performance of the clustering and the quality of the clusters were assessed using the silhouette coefficient. The evolution of this coefficient (Fig. 9) clearly shows that on the one hand, the methods based on the HAC algorithm produce lower values than those obtained by the KMS algorithms. On the other hand, for ED, silhouette indices are largely above those found by the L2 distance as written by Biabiany et al. (2020). This silhouette coefficient evolution allows us to keep four representative types of current regimes in this part of the Caribbean region. However, due to the lack of works for this region, comparisons between the present results and other studies were very limited. In other studies, authors have proposed a similar number of dominant regimes on a large scale, in the tropical Pacific (Fereday et al., 2008), for the determination of robust modes of Northern Hemisphere Sea ice variability (Fučkar et al., 2016), or for ocean mapping from environmental data (Zhao et al., 2020).

In our case, the velocity distributions show four singular profiles confirming the good performance of the clustering. Each cluster also had distinct monthly distributions. This analysis allowed to better understand the variability of the surface current circulations in this region.

### 4.2. Surface current analysis

In terms of spatial distribution, clusters show notable differences for both types of model analysis and three variability factors can be identified.

The first one is the seasonal evolution of the NBC retroflection front (Baklouti et al. 2007). The NBC feeds the Guiana Current (GC) but also separates sharply, near 6°–8°N, from the South American coastline and retroflects to feed, this time, the eastward NECC. Isolated large rings move north-westward toward the Caribbean Sea, on a course parallel to the South American coastline, then interact with the Lesser Antilles (Fratantoni et al., 2002, 2006). These two dynamic structures, GC and NBC rings, contribute significantly to the transfer of South Atlantic surface water to the Caribbean. These dynamic structures were found on the four identified clusters and seem to work year-round with intensity variations.

Another part of this variability is caused by the rings of the NBC that move northwestward from the equatorial Atlantic and interact with the steep topography of the Lesser Antilles arc. MC2 and HC1 are two typical cases. Interactions with the island chain cause significant disturbances of the inflow through the southern passages with a blocking. This provides a meridional transport of surface water northward, along the LA arc (Fratantoni and Richardson, 2006; Huang et al., 2021). The Lesser Antilles arc clearly diverted the initially north-westward drift of the NBC rings to a more northward course parallel to the island arc. Johns et al. (2002) have shown that the crossing of the Atlantic inflow to the Caribbean Sea through the passages of the Windward Islands (i.e. Lesser Antilles south islands from Trinidad to Martinique) has a highly asymmetric seasonal cycle, with a maximum in June and a minimum in September–October. The annual distribution of MC2 and HC3 clusters is close to that found by Johns et al. (2002).

The last identified factor is related to the North Atlantic Gyre and the associated North Equatorial Current. As the seasons change from winter to summer, the gyre shifts South by a few degrees in latitude. In this part of the study area, several clusters show lower current speeds and areas with large angular deviations in direction have also been identified. In the LA2 area, under influence of the north-east Trade Winds (i.e. Atlantic area between 14.5°N and 18°N), the relative frequencies of above-average speeds are the lowest. The wind-current shear zones are also the most extensive. The wind-driven flow occurs from the subtropical gyre location to 15°N, near the Martinique island (Johns et al., 2002). Passages through the Leeward islands have a maximum inflow in September and a minimum one in June.

The comparison between the large-scale meteorological situations corresponding to the parangons showed that the main differences between the current regime clusters are related to the location and the extension of the high-pressure centers, the positioning of the ITCZ, and the intensity of the low Caribbean Level Jet.

**4.2. *Sargassum* beachings**

All clusters contain beaching days in relative abundance, 12 to 36 % of beaching days for the two years 2019, 2020. The first peak of beachings, in March, seems linked with the maximum frequency of MC3 and HC3 clusters. The second peak of observed beachings occurs in August and seems associated with the MC1, HC2 and HC4 clusters. Johns et al (2020) found that windage forcing induced by the wind convergence accumulates *Sargassum* rafts within the ITCZ between April and September. This accumulation would contribute to the observed beaching peak in August. The clustering analysis on the beaching current sequences confirmed that the recurrence of HC3 (between December and June) and HC2 (between July and November) would induce large beachings on the Guadeloupe coasts during these respective periods. The HC2 current regime is characterized by the prevalence of

the North Atlantic gyre with weak velocities in the Western Central Atlantic and zonal streamlines. As for the HC3 current regime, it is characterized by strong Guiana Current with high velocities in the LA3 region and meridional streamlines almost parallel to the Lesser Antilles Arc.

## 4.3 Predictive model performance

A machine learning based method for predicting *Sargassum* beaching was proposed and was built from a decision tree. This method has already been used for other parameters and it allows to improve both the prediction accuracy and the fully black-box effect of the neural network. Compared to usual parametric statistical methods, it can effectively overcome the multicollinearity of explanatory variables (e.g., ocean current and surface wind). Depending on the temporal uncertainty ranges, the accuracy of the present decision tree is between 54.1% and 70.1% for HYCOM against 50.6% and 58.2% for Mercator (Table 6). The best performance scores are reached with the largest temporal uncertainty range +/-3 days. Similar performance scores were found for decision trees predicting summer rainfall in Chongqing (China) (Bo et al., 2020) or landslide hazard in the Yen Bai Province (Vietnam) (Pham et al., 2020).

During the year of testing (i.e. 2021) only 78 beachings days were observed. Despite this large difference between beaching and non-beaching occurrences, the predictive model produces quite symmetric performances for both true positives and true negatives (Table 6). This fact highlights the good ability of the model to handle the different chosen datasets. This stability of the decision support system tends to increase with the temporal uncertainty range. For HYCOM, this asymmetric performance difference drops below 4% at +/-3 days precision.

Several ways to improve the predictive model were identified. The lack of observational data in time (i.e. only two years) may weaken the final decision and induce overfitting. The tree could also be improved by weighting and prioritizing the different modules so as to increase their relevance. The improvement of the results can be found by optimizing the proposed decision calculation rule (3) to better integrate the characteristics of the observed phenomenon. The daily probability of Sargassum offshore abundance produced by Module A would also be improved with better quality *Sargassum* remote sensing observations, particularly the FA density data gap within 30 km of the coasts. The present study does not take into account the effects of other factors (e.g., presence of nutrient, sinking of algae and waves) which would allow a more realistic understanding of the *Sargassum* beachings.

## 5. Conclusion

For a decade, the Caribbean countries, and particularly the LA, have suffered from the impacts induced by the massive and regular arrival of *Sargassum* on their coastal areas. This study presents the application of a clustering approach to determine the types of surface current circulations integrating the additional wind drift and their possible links with the *Sargassum* beachings observed on the LA coasts. The Guadeloupe archipelago was chosen as beaching observational site for the period 2019-2020. This analysis was performed using the most recent versions of ocean current analysis, Mercator and HYCOM. The surface wind speed data from the ERA-5 model were also used. The Clustering of the spatiotemporal surface current fields including windage was produced using the K-Means algorithm combined with the expert distance metric. Silhouette index was used to determine the optimal number of clusters.

For this region (8-17°N, 66-55°W) divided into three sub-regions, we identify four coherent patterns from data sets. They contain

the current structures related to the Guiana currents, the branches of the subtropical Atlantic gyre, the front and the retroflection rings related to the NBC.

The finer resolution of HYCOM analysis provided more detailed information on surface current velocities near the islands than Mercator fields (i.e. mean local velocity difference of 0.3 m s$^{-1}$). Offshore, these differences remain very small.

Links between clusters and observed beachings in Guadeloupe were studied considering windage, parangon velocity distributions and monthly abundance maps. The surface current circulations characterizing the (MC3; HC3) and (MC4; HC2) cluster pairs seemed the most favorable for the transport and the beaching of *Sargassum* on the Lesser Antilles coasts.

The clustering analysis on the beaching current sequences based on HYCOM fields confirmed that the recurrence of HC3 (Seq2, between December and June) and HC2 (Seq4, between July and November) would induce large beachings on the Guadeloupe coasts during these respective periods. While the HC2 current regime is characterized by the prevalence of the North Atlantic gyre with weak zonal velocities, the HC3 current regime is marked by the influence of the NBC, the induced retroflection rings and strong Guiana Current leading to higher meridional velocities in the LA3 region.

Machine learning algorithms (KMS, ED, decision tree classifier) were applied to estimate the probability of *Sargassum* beachings in Guadeloupe, based on: surface current forecasts, current regime sequences and several combinations of probabilities. The performance score of this predictive model showed that HYCOM seems more suitable to reproduce small-scale current patterns inducing or not beachings in the Lesser Antilles. For one year of tests (i.e. 2021), the decision tree accuracy reached respectively 70.1% and 58.2% for HYCOM and Mercator with a temporal uncertainty range +/-3 days around the forecast date. This accuracy

could be improved by weighting and prioritizing the different modules. The daily probability of *Sargassum* offshore abundance used in the decision tree would also be improved with better quality *Sargassum* remote sensing observations, particularly the FA density data gap within 30 km of the coasts.

Due to the very recent availability of the selected HYCOM new generation version, the present study was conducted only on two years (i.e. 2019-2020). The studied period could be extended to more years to integrate the inter-annual variability of the surface

currents.

Nevertheless, the obtained results are very encouraging and open new possibilities for the forecasting of this natural hazard type. Machine learning methods developed in this analysis proved to be useful in the prevention of a natural risk depending on physical multifactorial combinations.

The present clustering analysis predictive system could be applied to other islands of the Lesser Antilles changing the observational

beaching site. The association of clustering methods and decision trees requiring low computational costs may enhance existing operational systems to help decision-makers in the *Sargassum* risk management. Maréchal et al. (2017) restrained the starting point of their operational short-term forecast system within 50-100 km of the LA coasts in order to reduce prediction errors. This geographical limit would correspond to a forecast period of 1-2 days before beaching. The present regional information on current dynamics leading to the arrival of *Sargassum* near the islands would be useful to extend this limit. In this way, it could be easier to

anticipate the implementation of the resources needed to collect the *Sargassum* algae on the shorelines. Like the *Sargassum* Inundation Reports (Trinanes et al., 2021), the present small-scale *Sargassum* beaching predictive model may contribute to the

region-wide efforts to help coastal communities managing this hazard.

**Data availability.** Data from this research are not publicly available. Interested researchers can contact the corresponding author of this article.

**Author contributions.** The study was mainly conceptualized and written by DB and EB. RC1, RC2, NS provided comments for the results and reviewed the manuscript. RC2 and NS helped with beaching observational data processing.

**Competing interests.** The authors declare that they have no conflict of interest.

**Acknowledgements.** This study was supported by the ERDF/C3AF project (grant number: CR/16-115). The authors wish to thank Danièle Frison who helped with the translation.

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

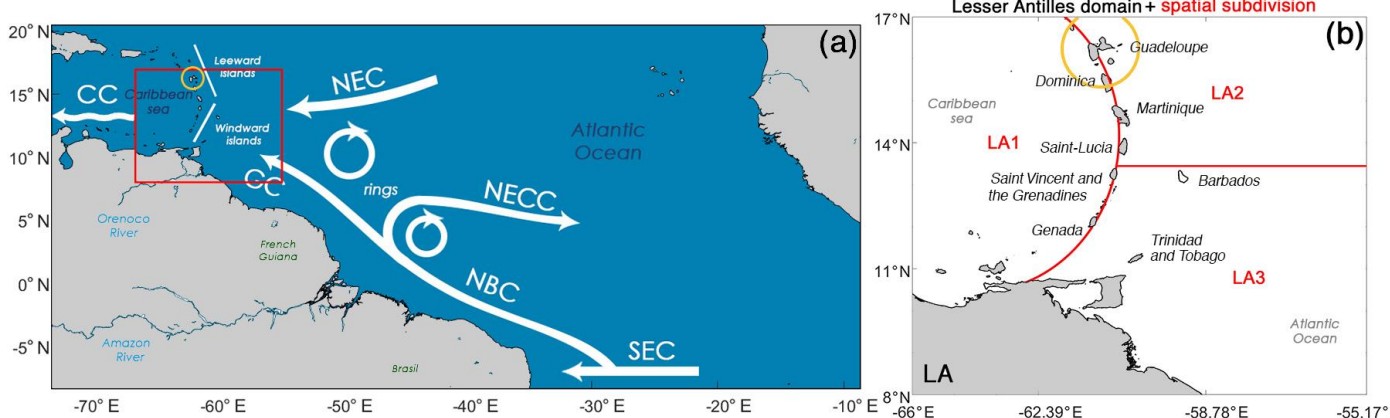


**Figure 1: (a) Main oceanic currents occurring and interacting in the central Atlantic and the Lesser Antilles regions; Caribbean Current (CC), North Equatorial current (NEC), North Brazil current (NBC), North equatorial Counter Current (NECC), South Equatorial current (SEC). Lesser Antilles domain (LA): the red rectangle corresponds to the study area (55-66° W, 8-17° N); (b) Spatial subdivision**
**of the study area into three sub-areas: LA1 (i.e. Caribbean Sea), LA2 (i.e. North Tropical Atlantic above Barbados (13.2° N)) and LA3 (i.e. North Tropical Atlantic below 13.2° N). The yellow circle corresponds to the 100 km offshore Guadeloupe area in which the satellite-based *Sargassum* abundance is analysed.**



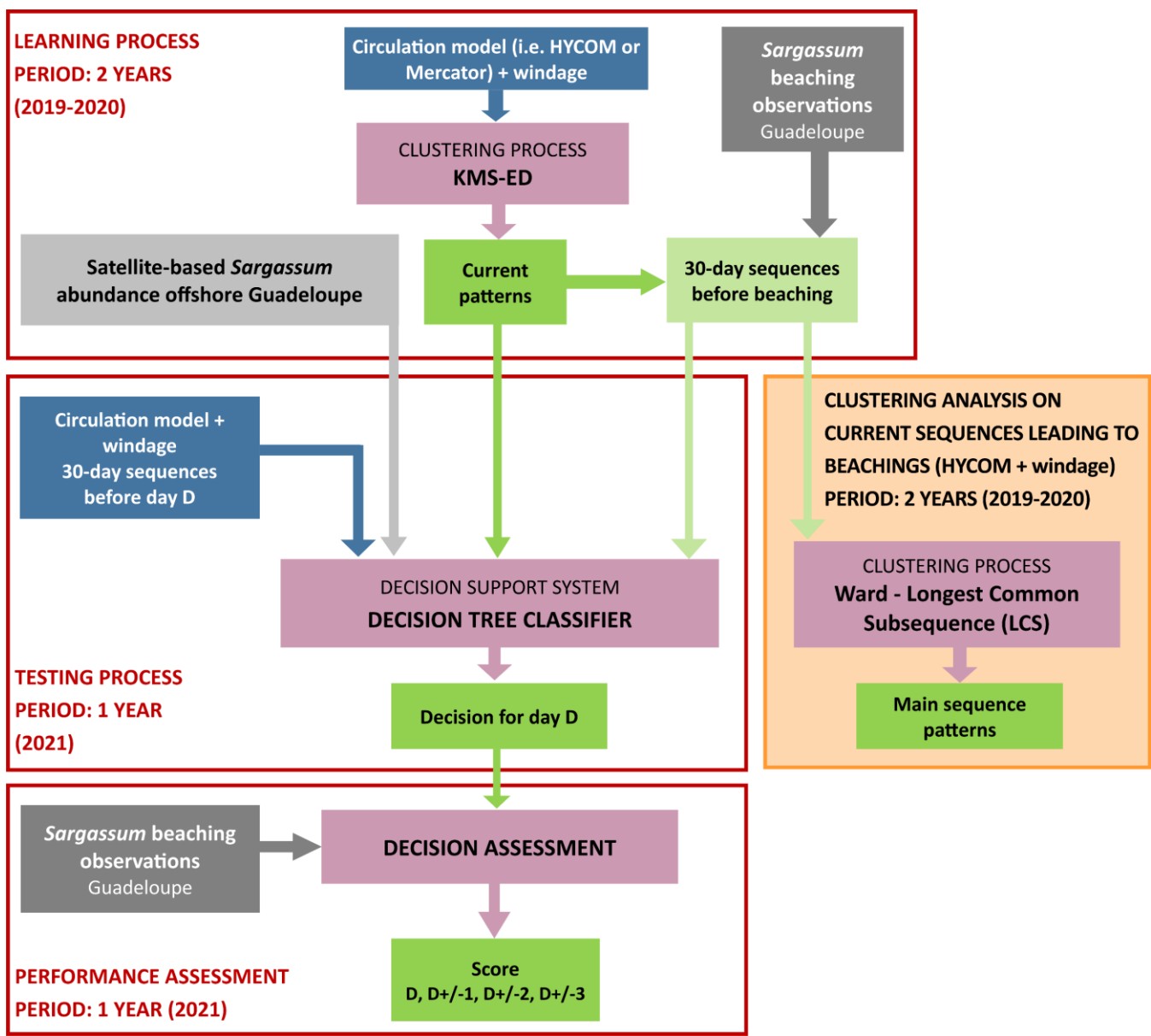

**Figure 2: A schematic showing the overall methodology.**


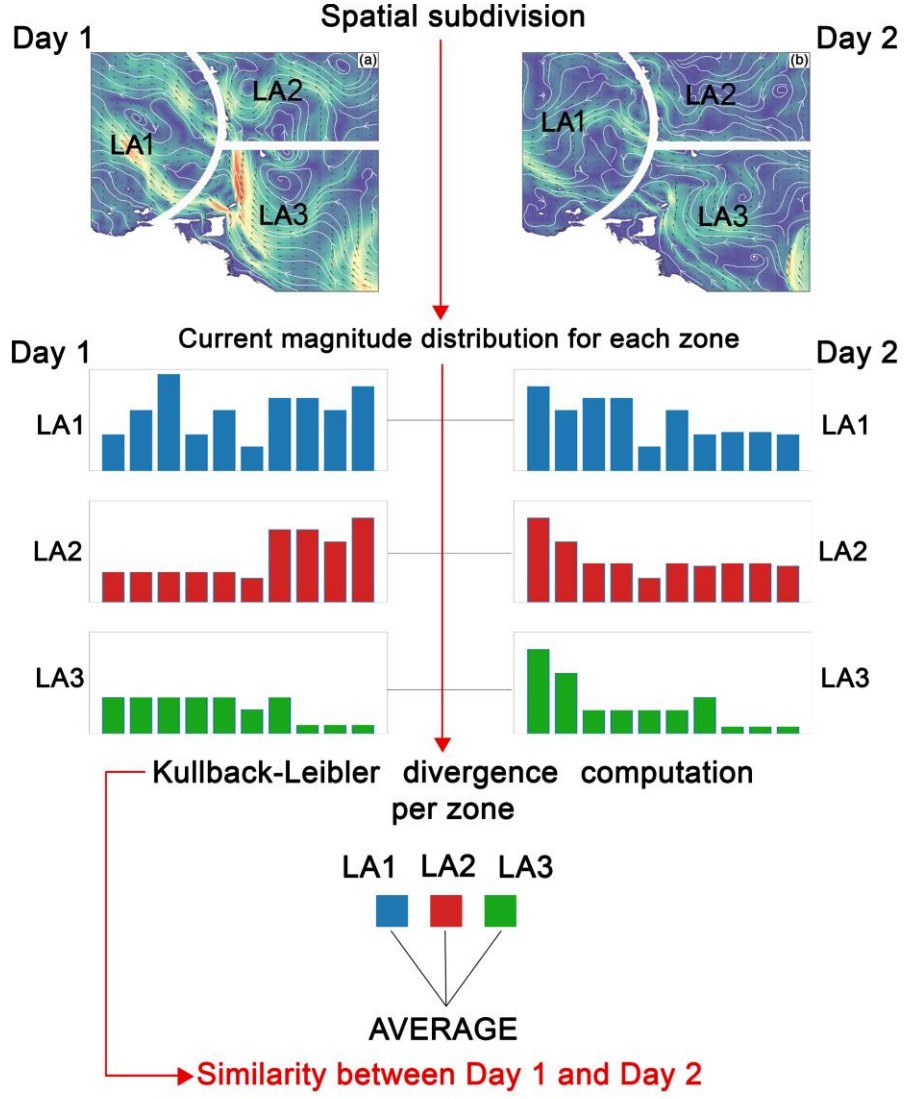

**Figure 3: The schematic of the Expert Distance process.**


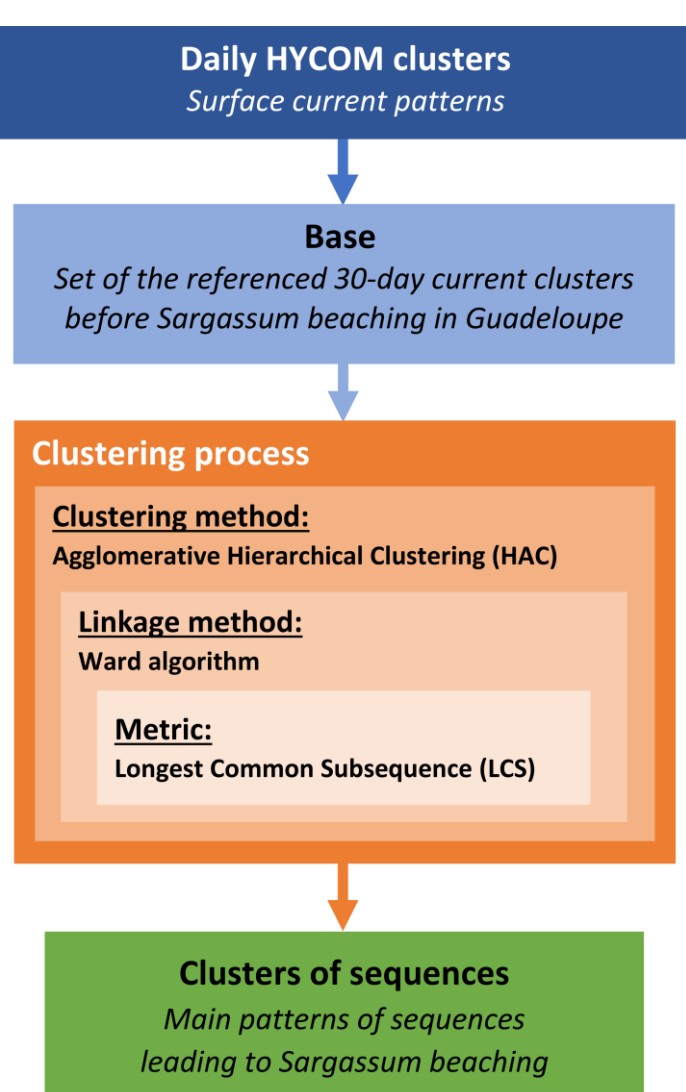

**Figure 4: The schematic of the clustering process used on the ocean current sequences leading to beachings.**


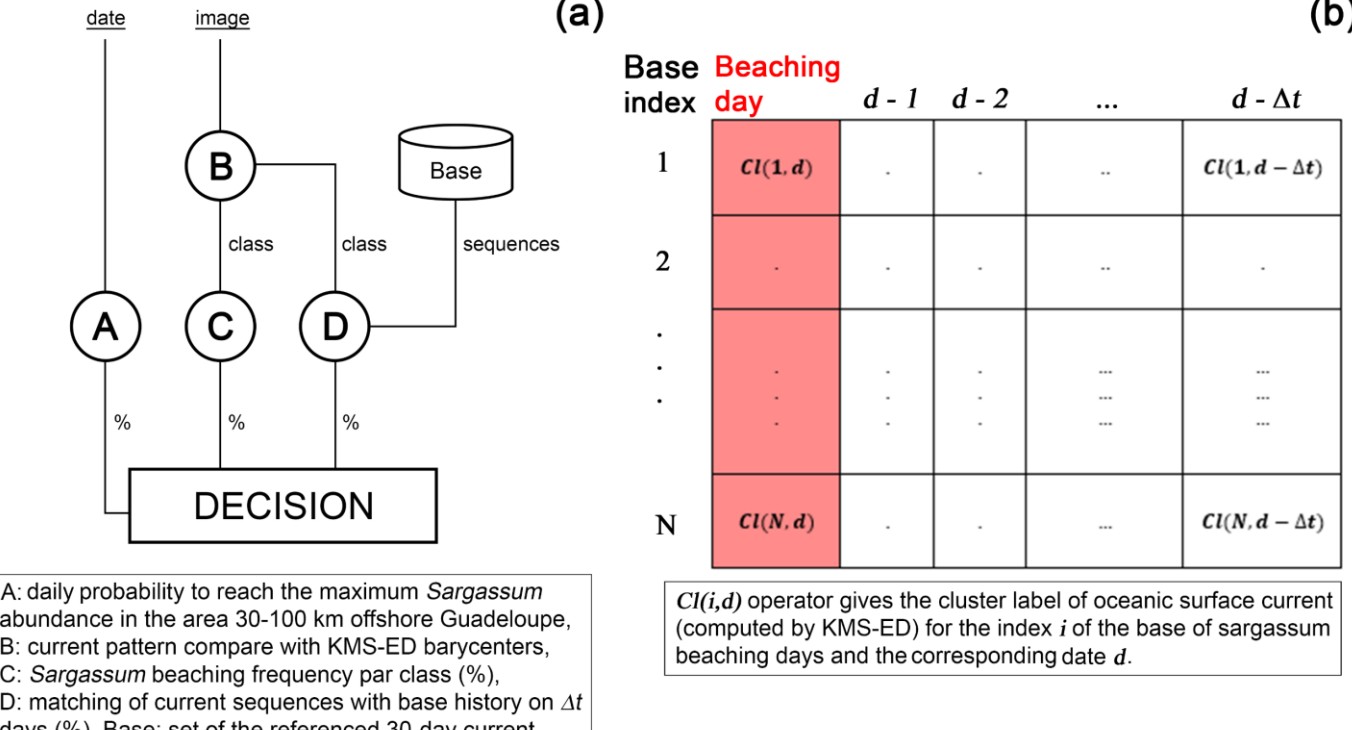

A: daily probability to reach the maximum *Sargassum* abundance in the area 30-100 km offshore Guadeloupe,
B: current pattern compare with KMS-ED barycenters,
C: *Sargassum* beaching frequency par class (%),
D: matching of current sequences with base history on $\Delta t$ days (%), Base: set of the referenced 30-day current sequences before *Sargassum* beaching.

$Cl(i,d)$ operator gives the cluster label of oceanic surface current (computed by KMS-ED) for the index $i$ of the base of sargassum beaching days and the corresponding date $d$.

**Figure 5: (a) Scheme of the decision tree classifier to predict *Sargassum* beaching probability. (b) Combination base of oceanic current clusters labels obtained by KMS-ED from each beaching day to $\Delta t$ days before.**


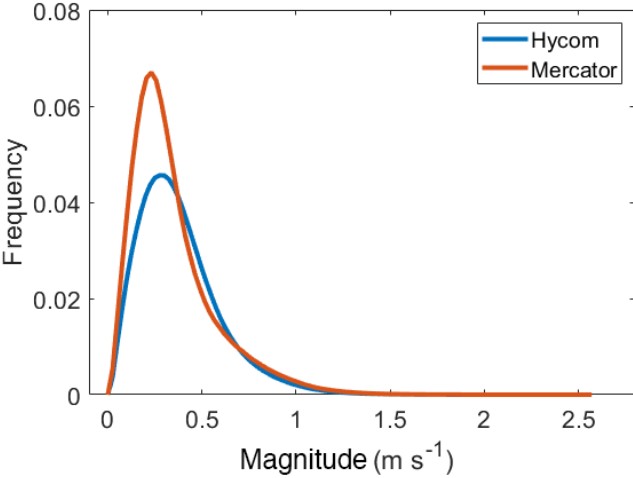

**Figure 6: Distributions of oceanic surface current magnitudes including windage for both models, HYCOM (blue) and Mercator (red) datasets.**


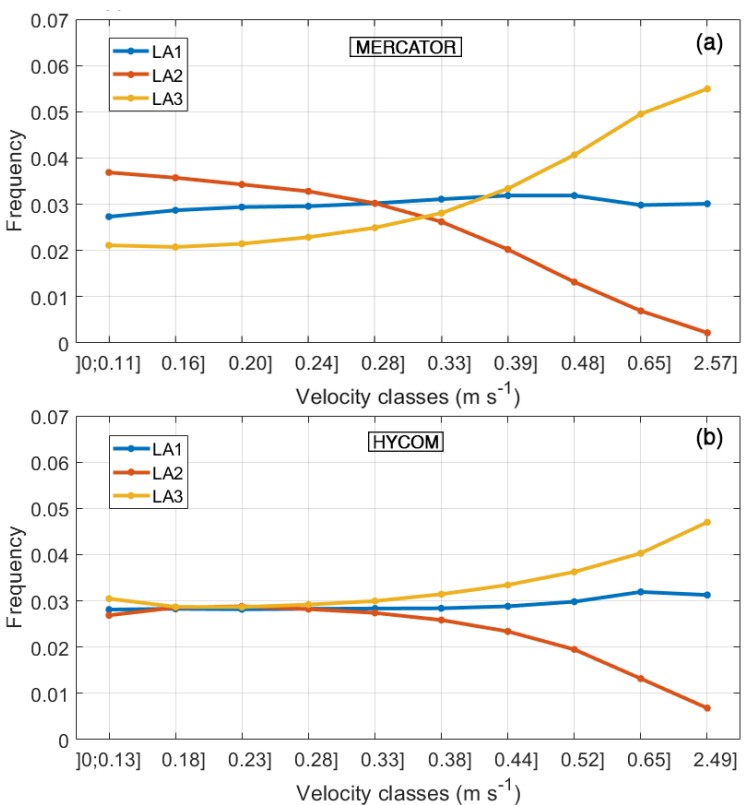

**Figure 7: Relative frequency distribution of current speeds for the three offshore sub-regions around the Lesser Antilles (2019-2020), LA1 (blue), LA2 (red), LA3 (yellow). (a) Mercator with ERA-5 windage and (b) HYCOM with ERA-5 windage.**


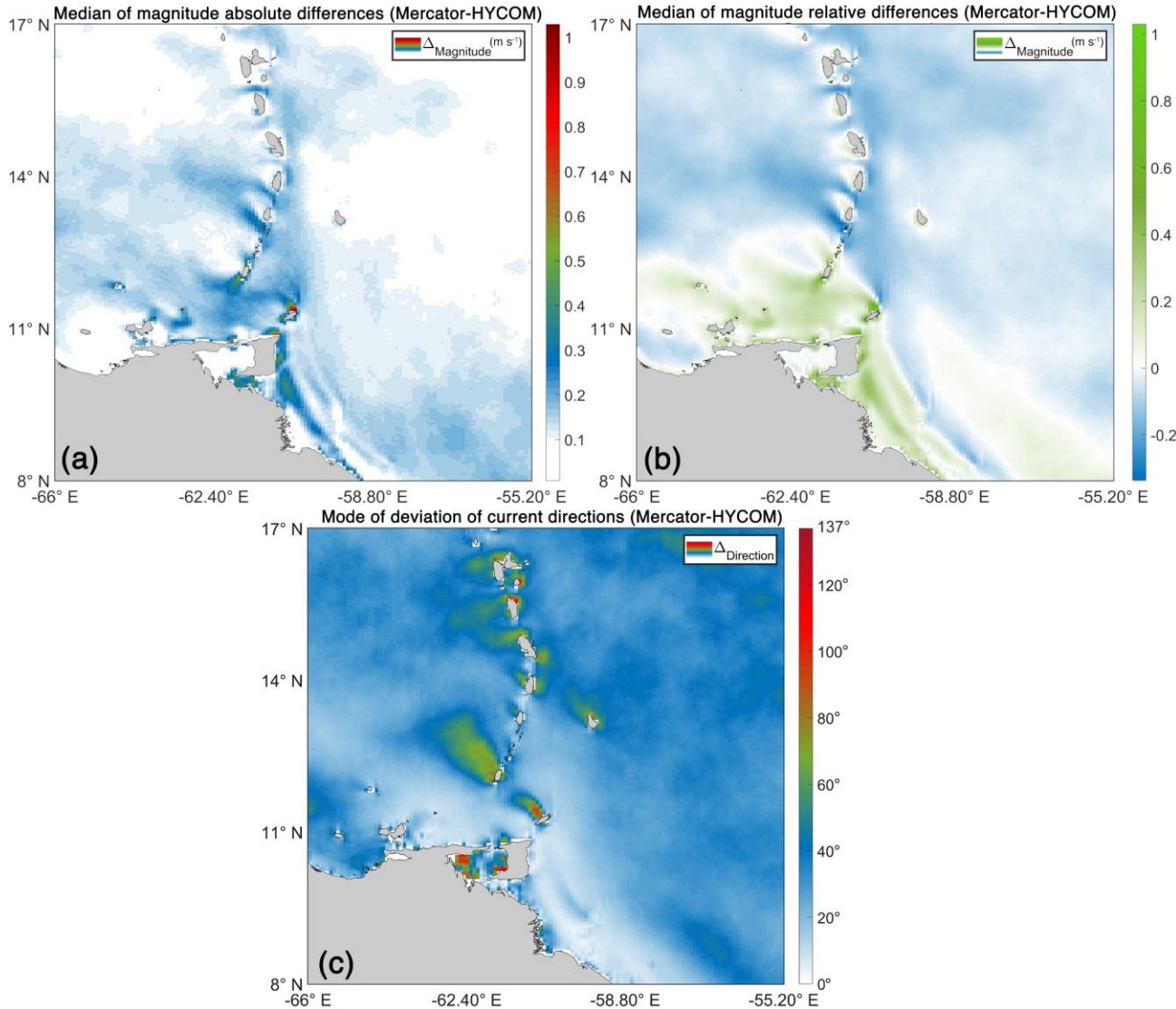

**Figure 8: Comparison between Mercator and HYCOM surface currents from 2019 to 2020 on the same 0.08° grid: (a) median of magnitude absolute differences (Mercator-HYCOM) in m s$^{-1}$ and (b) median of magnitude relative differences (Mercator-HYCOM) in m s$^{-1}$ and (c) mode of current direction differences (Mercator-HYCOM) in degree.**

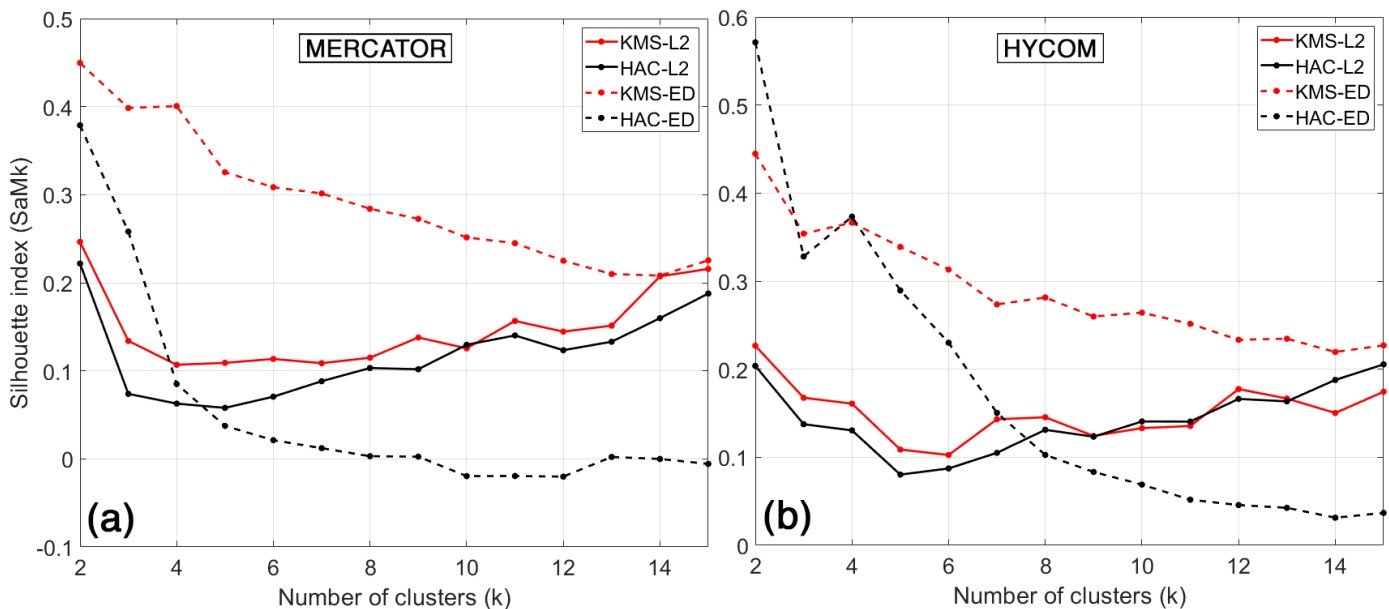

**Figure 9: Evolution of the SaMk silhouette index (by method) as a function of the number of clusters k, Mercator (a) and HYCOM (b): HAC method (black), KMS method (red), with L2 metric (solid line) and ED metric (dashed line).**

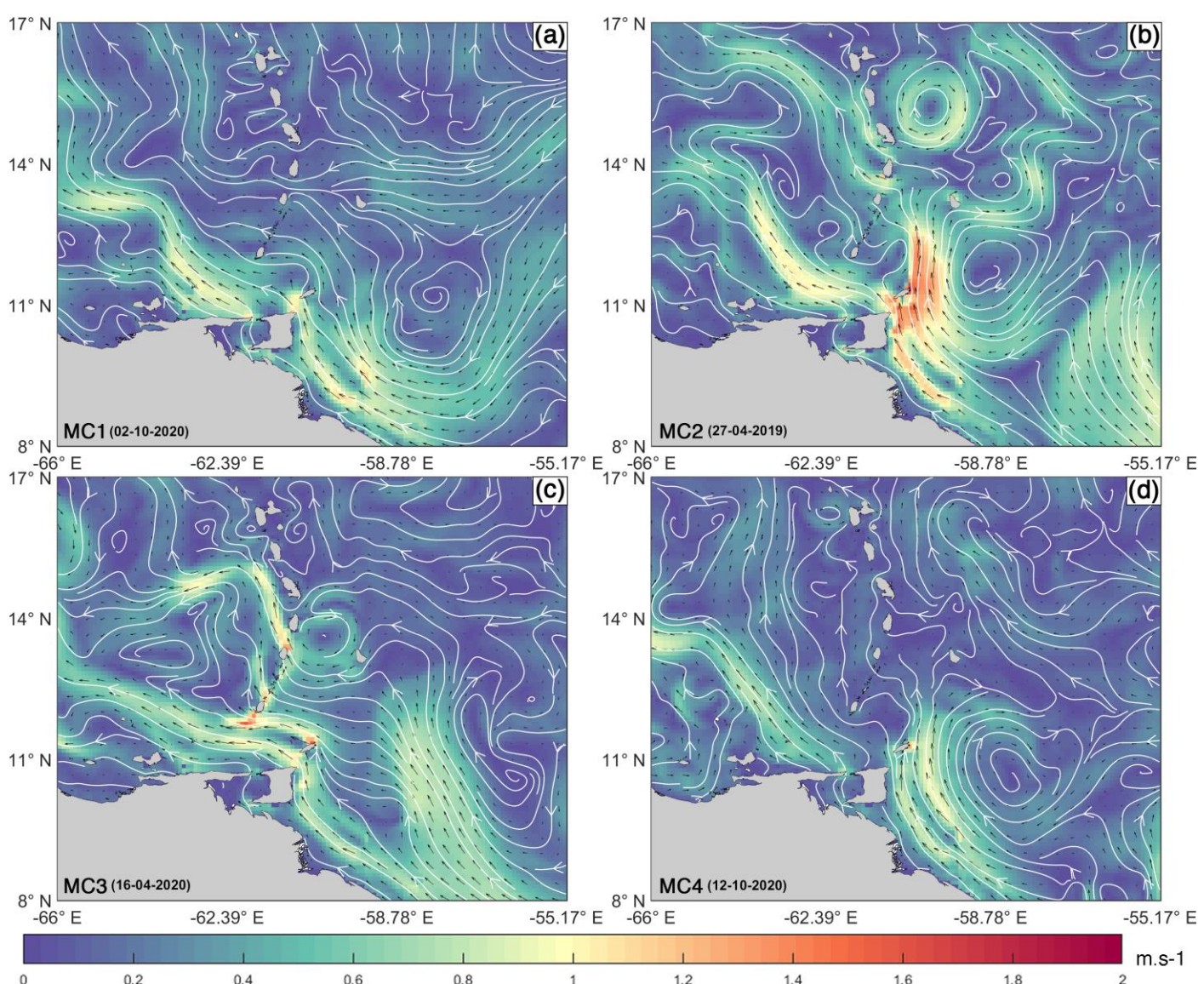

Figure 10: Representative elements of the clusters from Mercator current data combined with ERA-5 windage (KMS-ED method with k = 4): MC1 (day 02-10-2020) (a), MC2 (day 27-04-2019) (b), MC3 (day 16-04-2020) (c), MC4 (day 12-10-2020) (d).



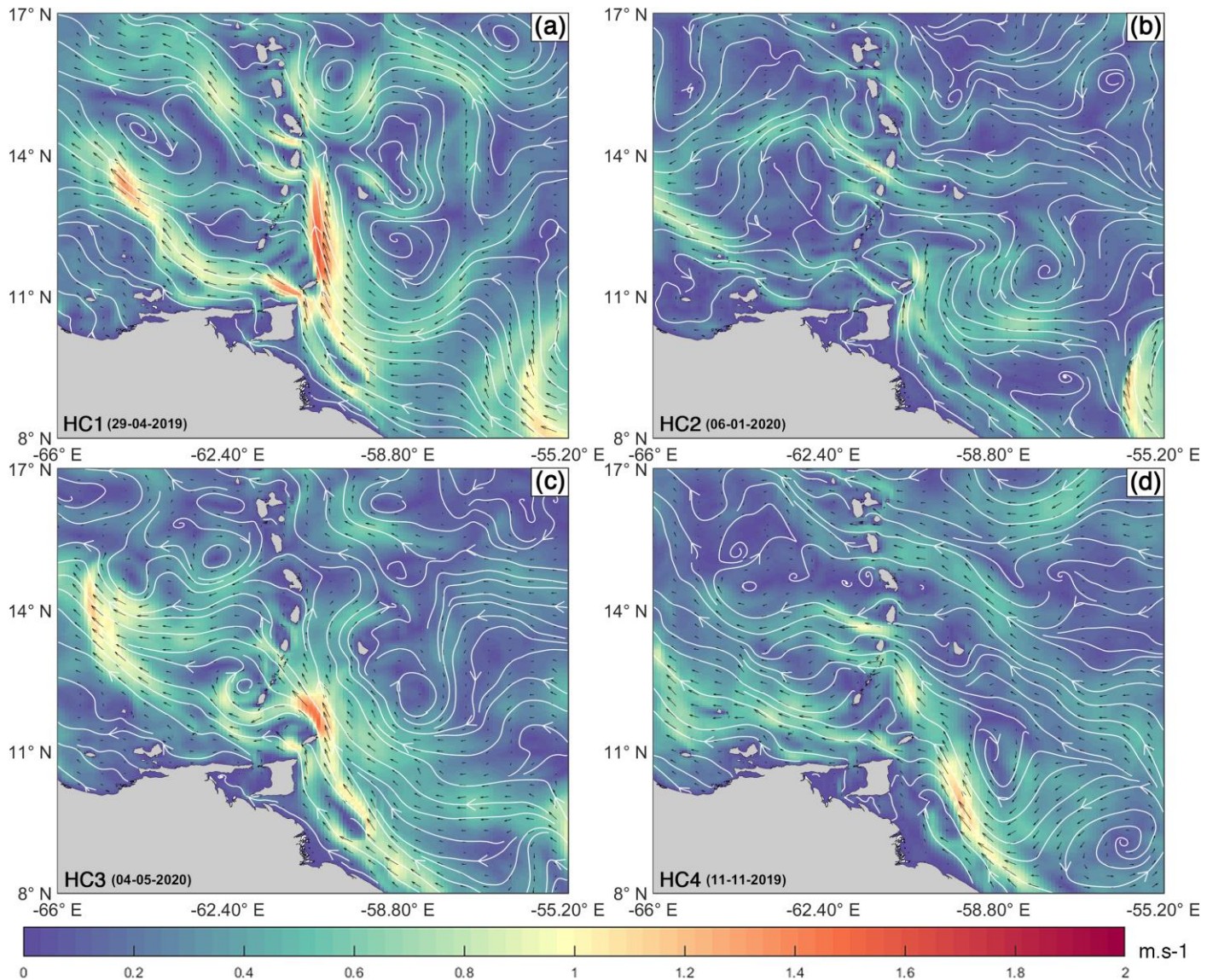

**Figure 11: Representative elements of the clusters from HYCOM current data combined with ERA-5 windage (KMS-ED method with k = 4): HC1 (day 29-04-2019) (a), HC2 (day 06-01-2020) (b), HC3 (day 04-05-2020) (c), HC4 (day 11-11-2019) (d). The HYCOM clusters numbering differs from the Mercator clusters numbering.**


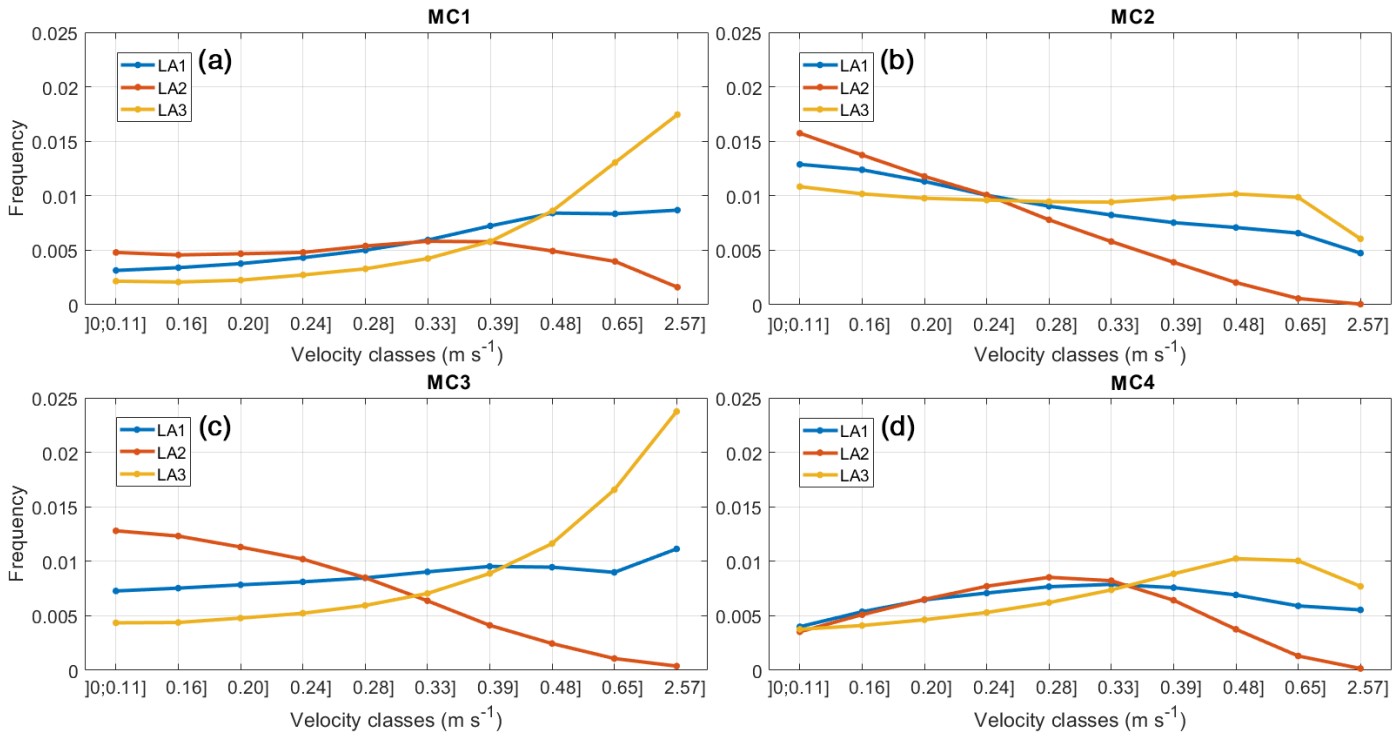

**Figure 12: Relative frequency distribution of current speeds for the three offshore sub-regions: MC1 (a), MC2 (b), MC3 (c) and MC4 (d). The representative elements were obtained after KMS-ED clustering for Mercator.**

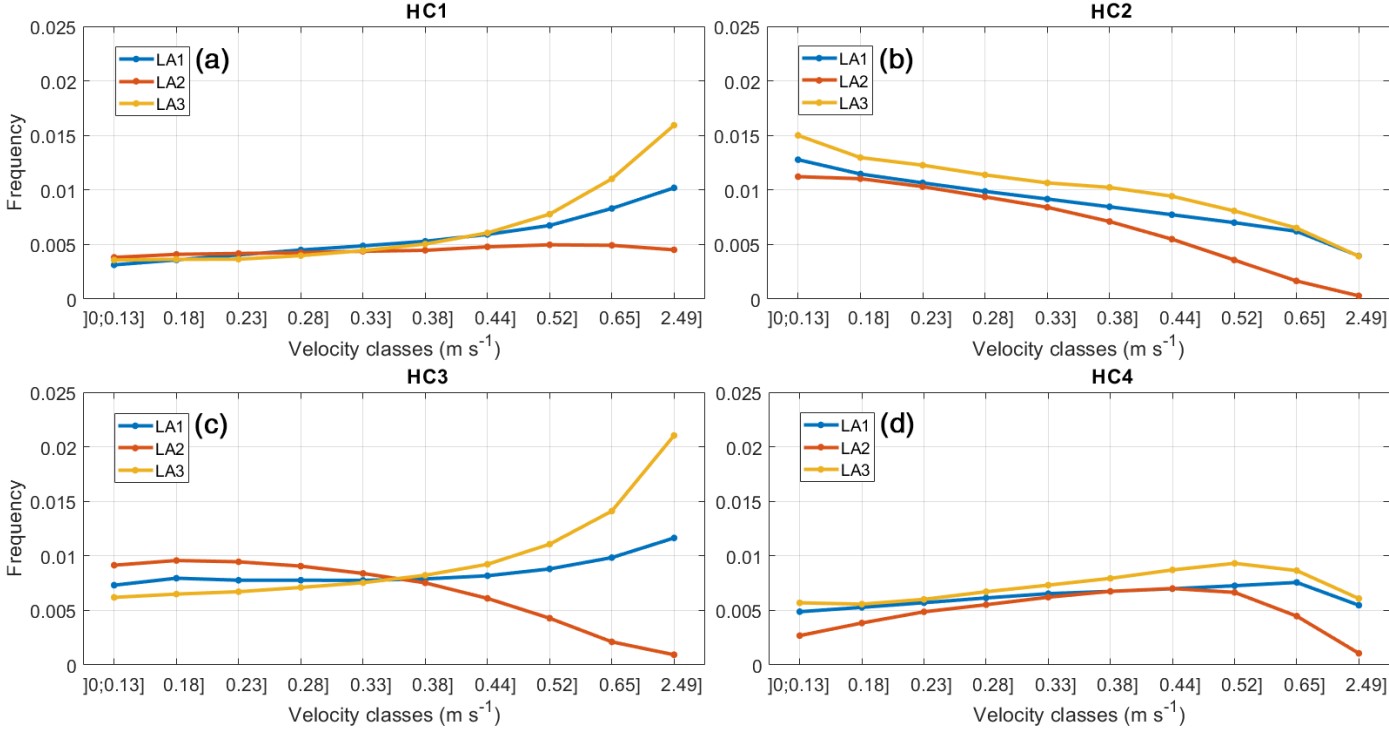


**Figure 13: Relative frequency distribution of current speeds for the three offshore sub-regions: HC1 (a), HC2 (b), HC3 (c) and HC4 (d). The representative elements were obtained after KMS-ED clustering for HYCOM.**

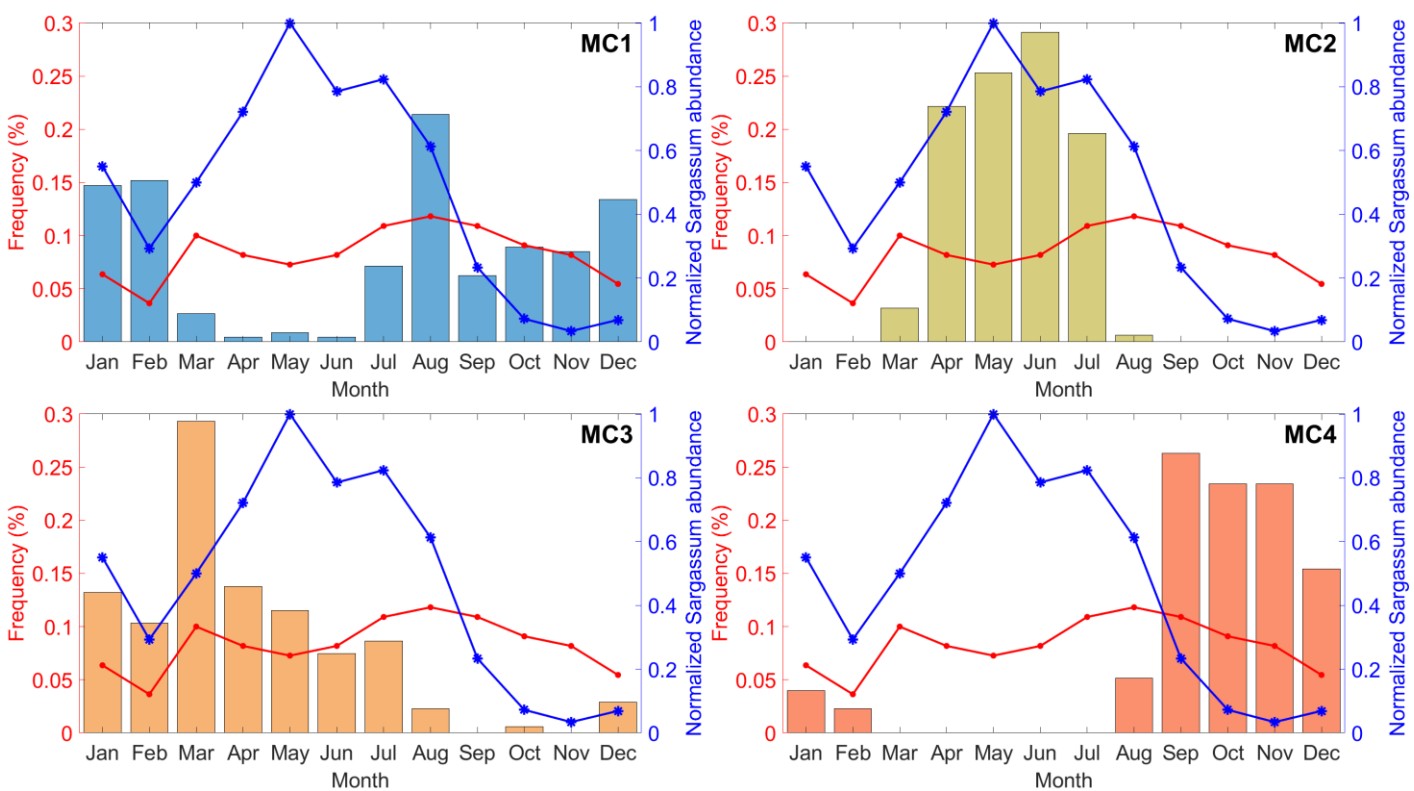

**Figure 14: Monthly distribution of cluster occurrence from Mercator outputs, from 2019 to 2020, in the Lesser Antilles (55-66°W, 8-17°N): MC1 (a), MC2 (b), MC3 (c) and MC4 (d). The red line shows the monthly distribution of *Sargassum* beachings on the coasts of Guadeloupe during the same period. The blue line indicates the monthly evolution of *Sargassum* abundance in the area 30-100 km offshore Guadeloupe normalized on the maximum value.**


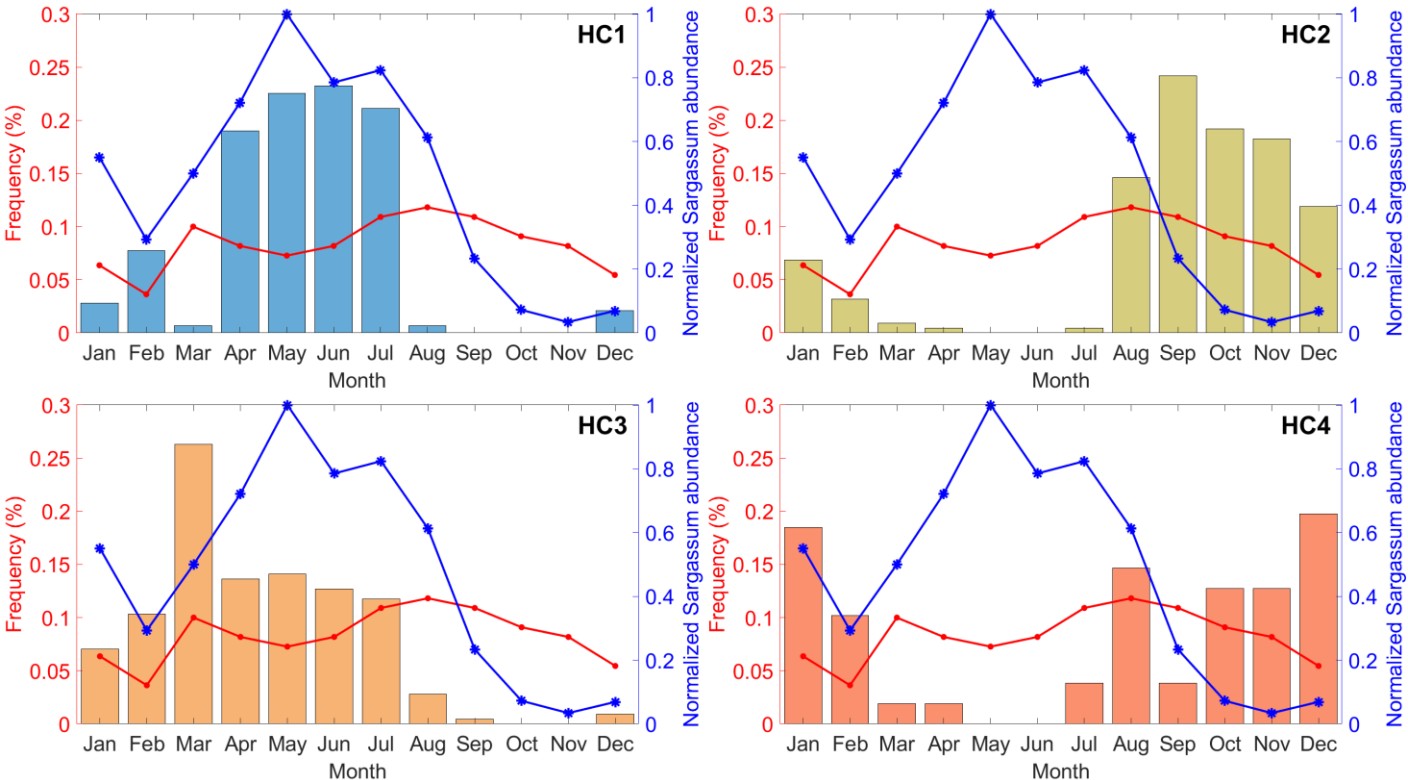

**Figure 15: Monthly distribution of cluster occurrence from HYCOM outputs, from 2019 to 2020, in the Lesser Antilles (55-66°W, 8-17°N): HC1 (a), HC2 (b), HC3 (c) and HC4 (d). The red line shows the monthly distribution of *Sargassum* beachings on the coasts of Guadeloupe during the same period. The blue line indicates the monthly evolution of *Sargassum* abundance in the area 30-100 km offshore Guadeloupe normalized on the maximum value.**


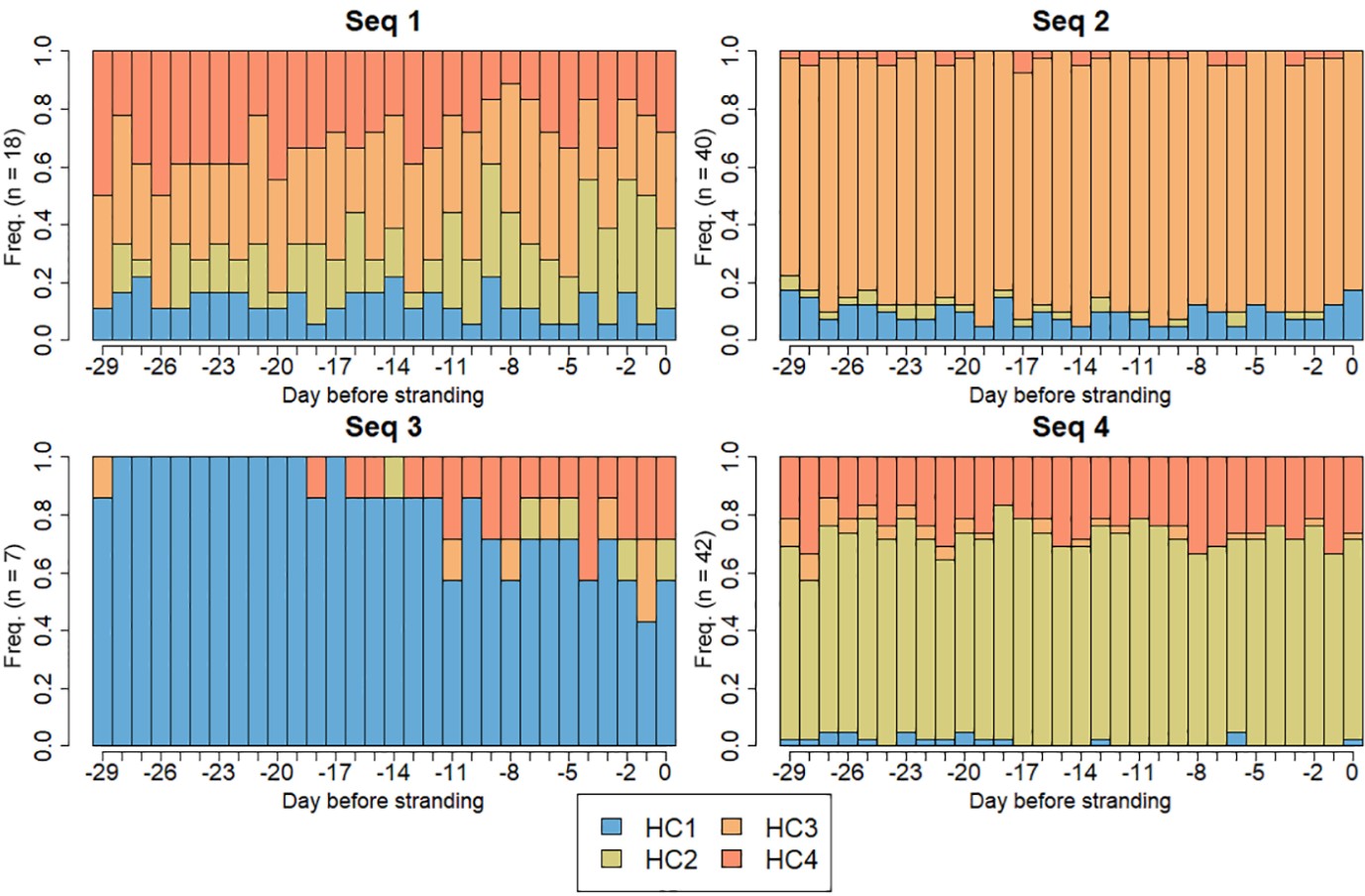

**Figure 16: Distribution of HYCOM current regime clusters (i.e. HC1 (in blue), HC2 (in green), HC3 (in orange), HC4 (in red)) in the 30-day sequence types (i.e. Seq1, Seq2, Seq3, Seq4).**


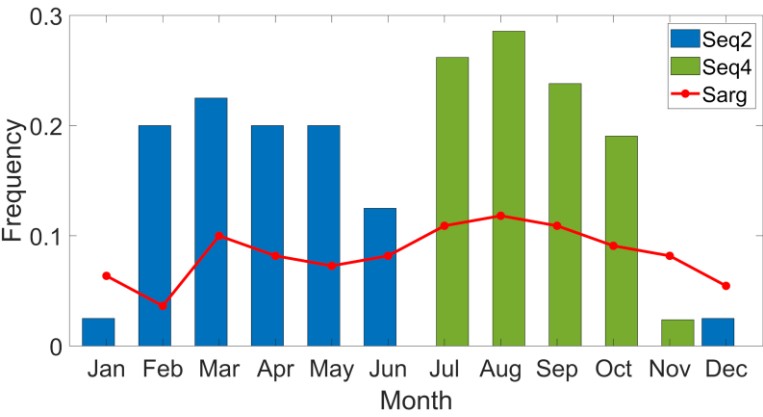

**Figure 17: Monthly distribution of the main observed current sequences leading to beaching: Seq2 (blue) and Seq4 (green). The red line represents the distribution of the observed beaching days.**


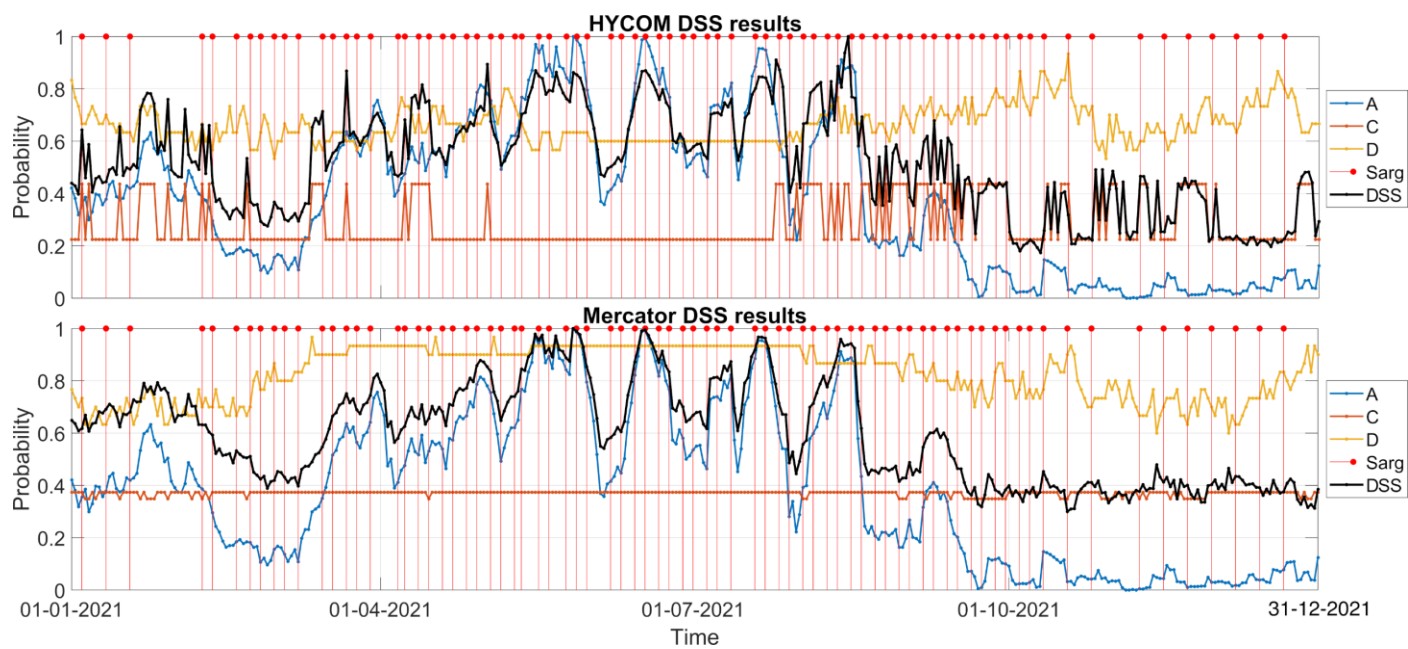

**Figure 18: Decision Support System (DSS) results: probability of beaching obtained per module. Daily probability to reach the maximum *Sargassum* abundance in the area 30-100 km offshore Guadeloupe for module A (blue line), beaching frequency per cluster for module C (orange line), match percentage for module D (yellow line), DSS Decision (black line). Days of observed beaching on Guadeloupe coasts**
**(red dots): HYCOM (a) and Mercator (b).**

| Deciles (D$_i$) | 0.1 | 0.2 | 0.3 | 0.4 | 0.5 | 0.6 | 0.7 | 0.8 | 0.9 | Max | Mean | Sigma |
|---|---|---|---|---|---|---|---|---|---|---|---|---|
| Mercator (m s$^{-1}$) | 0.11 | 0.16 | 0.20 | 0.24 | 0.28 | 0.32 | 0.39 | 0.48 | 0.65 | 2.57 | 0.33 | 0.22 |
| HYCOM (m s$^{-1}$) | 0.13 | 0.18 | 0.23 | 0.28 | 0.32 | 0.38 | 0.44 | 0.52 | 0.65 | 2.49 | 0.36 | 0.21 |

**Table 1: Boundaries of the histogram classes used to quantify surface currents velocity data with Sigma as Standard deviation.**

| Datasets | C1 | C2 | C3 | C4 |
|---|---|---|---|---|
| **MERCATOR** | **224** | 158 | 174 | 175 |
|  | (30.7%) | (21.6%) | (23.8%) | (23.9%) |
| **HYCOM** | 142 | **219** | 213 | 157 |
|  | (19. 4%) | (29.9%) | (29.1%) | (21.5%) |

**Table 2: Number of days corresponding to each cluster for MERCATOR and HYCOM datasets.**

| | | HYCOM | | | |
|---|---|---|---|---|---|
| | | C1 | C2 | C3 | C4 |
| | C1 | 8,3 % | 9,6 % | 7,1 % | **50,6 %** |
| **MERCATOR** | C2 | **60,4 %** | (-) | 12,4 % | 1,3 % |
| | C3 | 0,3 % | 4,8 % | **56,7 %** | 4,7 % |
| | C4 | (-) | **69,8 %** | 0,8 % | 3,1 % |

**Table 3: Correspondence table between the 4 clusters generated with MERCATOR and HYCOM datasets. The percentage expresses the proportion of common days between two clusters ((-) for 0%).**


| Datasets | C1 | C2 | C3 | C4 |
|----------|-----|-----|-----|-----|
| MERCATOR | 33 | 15 | 34 | 28 |
| HYCOM | 14 | 35 | 40 | 21 |

Table 4: Distribution of observed *Sargassum* beaching days (Guadeloupe coasts) in MERCATOR and HYCOM clusters.


| 30-day current sequence before beaching (HYCOM) | Seq1 | Seq2 | Seq3 | Seq4 |
|----------|-----|-----|-----|-----|
| n | 18 | 40 | 7 | 42 |
| % | 16.8 | 37.4 | 6.5 | 39.3 |

Table 5: Distribution of sequence clusters, with (n) corresponding to the number of sequences in each cluster and (%) corresponding to the ratio of the number of sequences on the total.



| Time range around D (day) | Datasets | True positive (recall %) | True negative (recall %) | Accuracy (ratio %) |
|---|---|---|---|---|
| 0 | **HYCOM** | **46 (59.0%)** | 151 (52.8%) | **197 (54.1%)** |
| | **Mercator** | **43 (55.1%)** | 141 (49.3%) | **184 (50.6%)** |
| +/- 1 | **HYCOM** | 52 (66.7%) | 175 (61.2%) | 227 (62.4%) |
| | **Mercator** | 47 (60.3%) | 151 (52.8%) | 198 (54.4%) |
| +/- 2 | **HYCOM** | 55 (70.5%) | 189 (66.1%) | 244 (67.0%) |
| | **Mercator** | 51 (65.4%) | 155 (54.2%) | 206 (56.6%) |
| +/- 3 | **HYCOM** | **57 (73.1%)** | 198 (69.2%) | **255 (70.1%)** |
| | **Mercator** | **51 (65.4%)** | 161 (56.3%) | **212 (58.2%)** |

**Table 6: Decision tree performance scores: "True positive/negative" respectively refer to the number of observed beaching/non-beaching days predicted by the decision system; "Recall" refers to the ratio in percentage between these respective numbers of days and the total number of tests (i.e. 364 days); "Accuracy" corresponds to the number of days with a true prediction and its ratio in percentage over the total number of tested days.**