# Peer review of "Clustering analysis of the *Sargassum* transport process: application to beaching prediction in the Lesser Antilles"

_Ocean Science, 2021_

## Referee Comment (RC2)

**Review of manuscript: Clustering analysis of the *Sargassum* transport process: application to stranding prediction in the Lesser Antilles by Bernard D *et al.**

**1. General comments**

The authors present a very interesting framework and method to better understand the ocean dynamics behind the strandings of *Sargassum* in the Lesser Antilles and to estimate their occurrence. The methodology presented is quite complex as well. A better explanation of the methodology is necessary, especially for the oceanographic audience of this journal to adequately follow and understand this interesting study. Section 2 I believe can be improved by making it easier for the reader to follow, especially the non-experts in these clustering methods. The technical details necessary for the reader to follow the study should be clearly described and the other details can be added as a section in supplementary material. A schematic of the method is given in fig. 2 for Section 2.7, but maybe a schematic for sections 2.5 and 2.6 could help too. In the discussion, I found that some comment on the impact (if any) of considering processes other than windage (e.g. presence of nutrients, sinking of *Sargassum*, waves?) could have on an even better understanding of the Sargassum strandings, was missing.

**2. Specific comments**

L23: "Strandings were also be observed in Africa (Széchy et al., 2012)." Why mention the occurrence of strandings in Africa? Any connection with the Caribbean strandings? Did the *Sargassum* strandings also cause natural hazards on the African coast?

L61: "MODIS AFAI satellite images", please define/describe

L66-68: Could be useful to include some references of the methodology here.

L69: A general definition of predictive modelling is missing in the introduction for the readers which do not know about this method and how it compares with a conventional forecast. For example could be included here (Line 69).

L75-76: "To optimize the final partitioning, an additional metric based on the Kullback Leiber divergence (Kulback and Leibler, 1951, Biabiany et al., 2020) will be included" : quite specific on the methodology, for readers not familiarised with

this method it could be hard to follow in this point in the introduction. More general details can be given, or this point can be moved to the methods section.

L82-83: "This ocean region corresponds to the CA and TA1 boxes in Johns (2020)", maybe say approximately corresponds, as not exactly the same. The LA3 region goes further south and LA2 and LA3 go until -55°E, whales region TA1 till -50°E. Most importantly, why choose the study regions to correspond to CA and TA1 boxes from Johns (2020)?

L96-96: From what I understand this dataset was not used before to simulate *Sargassum* trajectories, but was it used in any other Lagrangian study? Any validation studies done on the velocity outputs of this dataset?

L101: "Comparison between HYCOM and Mercator results" Do you mean the results from the *Sargassum* trajectories or a comparison of the velocity outputs of these datasets?

Section 2.3: Whats is the spatial and temporal resolution of the ERA-5 wind dataset?

L128: "Ward's method for HAC" Please explain and add reference.

L129-130: "with its own expertise on the input data" What do you mean by these? Please provide further explanations. Also, the new method name is not specified at all in section 2.5.1, and it will help for the reader to better follow the methodology. This section is only 5 lines long, more details on the process of the clustering methods could be given.

L132: "L2 clustering methods…" Please explain L2 in this context.

L133: "gatherings of different physical situations". What do you mean by this? Maybe give an example of physical situations for this particular study scenario. You refer to this in the next phrase as "biases". Is there then a tendency towards a specific physical situation?

L134: "spatial variability" : At what scales?

L139-L140: "The analyzed daily fields include a total of 14 279 meshes (4 282 meshes in LA1, 3 407 meshes in LA2 and 4 536 meshes in LA3). The remainder corresponds to land areas." What do you refer to here with meshes? The land areas

then correspond to *Sargassum* strandings? For clarity, these details could be described in a dataset section better, rather than in the middle of the methods description.

L141-142: "The second step was to group the information carried by the daily current velocity fields conditionally to the three given zones into histograms." More details on histograms, for example binning, velocity data from HYCOM and Mercator?

L158: "optimal matching methods" Please explain and add some references.

L158: "dividing the population" what do you refer to exactly here by population? Population of strandings or backward sequences?

L160-162: Please give further details (maybe as supplementary material?) and add more references.

L186-L187: "was experimented on the first 120 days…". Was experimented to…? Recall aim of doing these tests. Also why 120 days and during this period of time? Could results vary a lot if done during the northern hemisphere Summer months instead?

L190: Can maybe start section 3.1 giving some context on why this analysis is done.

L191: "90% of them remain below 0.65 m/s". For both models exactly same?

L193: Figure 3 distributions how are they calculated? With histograms? Kernel Density Estimator or something else applied to obtain this "smooth" distribution curves?

L194-L195: 5 times greater for both models?

L207-208: what are the implications of these differences?

L272-L273: "The monthly evolution of observed stranding days on the Guadeloupe coasts, the monthly evolution of Sargassum abundance over
the Central Atlantic region (SaWS, https://optics.marine.usf.edu/projects/
SaWS.html)" I imagine it should be: "Guadeloupe coasts and the monthly evolution…", to make clear you talking about two datasets. The observed stranding dataset is mentioned in the dataset section (section 2.4), but not the *Sargassum* abundance over the Central Atlantic region.

**3. Technical corrections**

Please write Sargassum in italics, like it is done in other studies like for example Johns *et al.*, (2020), as you are writing its scientific name, even if it is just the genus in this case.

L10: "including windage effect": gives the impression the HYCOM and Mercator datasets already include the windage effect, when you actually added separately. Please improve phrasing.

L20: "LA received…" to "The LA received…"

L23: "…were also be observed…" to "…were also observed…"

L46: Improve sentence, e.g. "… multi-year reanalysis of wind and current, and numerical models, both the role of subsurface nutrient supply and surface current transport were estimated."

L50: "Sargassum Watch System SaWS" to "Sargassum Watch System (SaWS)"

L83: "in Johns (2020)" *et al* missing.

L92: Please define the abbreviations HYCOM and NCODA (HYCOM defined in abstract but not in the main text)

L94: Please define 12Z fields.

L94-95: "u and v components" to "zonal (u) and meridional (v) velocity components"

L101-102: "Comparison…in the focused region" to "A comparison.. in the study region."

L107: "Sargassum raft transport", maybe trajectories instead of transport is more appropriate?

L112-113: "The region analyzed in the present work corresponds to the CA - TA1 region defined in Johns et al. (2020)" already mentioned in L82-83, is it necessary to repeat here?

L116-117: "This period includes 730 observational days with 110 days of observed strandings." , phrasing not clear do you mean that out of the total 730 days of data, only 110 days included observations of *Sargassum* strandings?

L137: "above Barbados island" to "above the island of Barbados"

L142: "The similarity of the most similar fields is estimated per pair.." Improve phrasing. What do you refer to exactly? Per pair of *Sargassum* meshes?

L148: "The SaMk index" to "The Silhouette (SaMk) index"

L151: Define all variables of equation 2!

L153-154: Improve phrasing.

L156: "January 2020" to "January 2019"

L165-L166: " surface currents with windage effects (Mercator, HYCOM and ERA-5)" to " surface currents (Mercator and HYCOM) with windage effects (ERA-5)"

L186-L187: "The proposed tree in Fig. 2…". Move to new line, to separate it from the phrase explaining the terms in equation (4)

L191: "do not exceed 2.57 m/s". Maybe better to say the maximum is 2.57 m/s, if not it sounds like 2.57 m/s is a key velocity value that should not be exceeded for some reason.

L193-L194: add at end to which model it each value corresponds to e.g. ".. for HYCOM and Mercator, respectively."

L205: "Globally, at sea, the current.." Is it necessary to specify at sea? What do you exactly mean with at sea here, open ocean?

L210: "into three magnitude groups of 45°" to "into three magnitude groups of 45° intervals"?

L215: Improve phrasing, gives the impression you used equation (1) to perform the clustering.

L244: "Table 3 shows results" to "Table 3 shows the results"

L297: "remain with probabilities" add probabilities of… Help the reader follow better your study, recalling details.

L317: Improve wording of Section 4.2 title, for example can simply remove "hazard"

L320: "retroflexion" to "retroflection"

L345 "The first peak of strandings, in March and seems.." to "The first peak of strandings, in March, seems.."

L373: Write as K-Means, and also in L217, write method in the same way.

**3.1. Figures and tables**

Figure 2: Describe BASE abbreviation as in L175.

Figures 4, 9 and 10: x-axis tick labels not clear, please improve.

Table 1: Header mean to Mean

Table 5: Caption mention what n and % refer to exactly.

---

## Author Comment (AC1)

Dear referee 1,

We thank you for the attention that you paid to this review and for your helpful comments and suggestions.

Firstly, in the introduction of your report, you mentioned the *"rather low performance of the classifier"*. Following this remark, we made some major changes to strengthen the evaluation of the decision tree classifier and to improve its recall scores. To strengthen the performance evaluation, the testing period was extended from the first four months of 2021 (i.e., from January 2021 to April 2021) to the full year of 2021 including seasonal variations of the offshore Sargassum abundance. To improve the recall score of the classifier, the module A producing the monthly probability of beaching was replaced by a new module based on satellite observations which produces the weekly probability to reach the maximum observed cumulative floating algae density in an area of 100 km radius offshore Guadeloupe. The performance evaluation of the classifier was also extended by adding three temporal uncertainty ranges around the decision day, respectively: +/-1 days, +/-2 days, +/-3 days. While the classifier may reproduce 61.5% of the observed beachings in 2021 with an accuracy lower than one day (this value reached 41.7% with the old module A and the limited testing period of four months), this recall score reaches 74.4% at +/-3 days accuracy.

Please find below our answers to your remarks (in bold). The proposed changes in the text are marked in red.

**1) The discussion of the appropriate time and space scale must be introduced in the methods to justify the choices made (30 days sequences, areas, monthly probability).**

Our answer:

We will justify our choices of time and space scale (30 days sequences, areas, monthly probability) in the methods section.

Areas are already described in the "2.5.2 Use of Expert Deviation in clustering algorithms" methods subsection, Lines 135-140:

*The LA study area was separated into three parts (Fig. 1b) based on the Sargassum rafts transport centers of action reported in the literature (Franks et al., 2016; Berline et al., 2020). To the west of LA, the first zone, LA1, is centered on the Caribbean Sea. To the east, the Atlantic zone has been split into two areas towards 13.5°N, just above Barbados island. To the south-east is the LA3 zone under the influence of the North Equatorial Recirculation Region (NERR) and its retroflection rings, while to the north-east is the LA2 zone, more representative of the North Equatorial Current. The analyzed daily fields include a total of 14 279 meshes (4 282 meshes in LA1, 3 407 meshes in LA2 and 4 536 meshes in LA3).*

Concerning the 30 days duration, the following sentences will be added in the methods, L154 (Section 2.6):

*"The 30 days duration corresponds to the empirical transport time of a passive particle moving from the main entrance location of Sargassum rafts in the Lesser Antilles area (i.e., in LA3 zone, 8°N; -55°E) to Guadeloupe (i.e., LA2 zone). Based on the mean current magnitude of 0.2 m s$^{-1}$ (average value over the LA zone, in HYCOM and in Mercator data) and the distance of 500 km between the main entrance location and the Guadeloupe coasts, 29 days are obtained for the transport. For simplicity, the duration of 30 days was selected instead of 29 days."*

To improve the decision support system, the stranding monthly probability (i.e., Module A in the decision tree) will be replaced by a weekly probability of Sargassum presence in an area of 100 km radius offshore Guadeloupe. This probability is based on the cumulative 7-day Floating Algae density (Wang and Hu, 2016) estimated in this area during the two years 2019 and 2020.

The following section will be added in the "Datasets and method" section:

*"**2.5 Satellite-based offshore abundance of Sargassum***

*Sargassum satellite observations were included in the present decision support system. To quantify the abundance of Sargassum in an area of 100 km radius offshore Guadeloupe, the 7-day Floating Algae (FA) density fields derived from the Alternative Floating Algae Index (Wang and Hu, 2016) were analyzed. As described by Trinanes*

*et al. (2021), the 7-day Floating Algae (FA) density fields are accumulated on 7 days and have a 0.1° resolution. Due to optical complexity in nearshore waters, the FA density fields are masked with missing values within 30 km from shoreline (Trinanes et al. 2021). The cumulative FA density values were summed in the area 30-100 km offshore Guadeloupe (Fig. 1) then weekly averaged during the two years 2019 and 2020."*

In the "Decision support system" section, the following sentence (L.168):

*"Module A takes as input the month of the selected day and returns the associated monthly probability (frequency) of stranding;"*

Will be replaced by:

*"Module A takes as input the week number of the selected day and returns the associated weekly probability to reach the maximum offshore abundance of Sargassum (based on observational FA density values during the two years 2019 and 2020)."*

[Figure]

*"Figure 1: (a) Main oceanic currents occurring and interacting in the central Atlantic and the Lesser Antilles regions; Caribbean Current (CC), North Equatorial current (NEC), North Brazil current (NBC), North equatorial Counter Current (NECC), South Equatorial current (SEC). Lesser Antilles domain (LA): the red rectangle corresponds to the study area (55-66° W, 8-17° N); (b) Spatial subdivision of the study area into three sub-areas: LA1 (i.e., Caribbean Sea), LA2 (i.e., North Tropical Atlantic above Barbados (13.2° N)) and LA3 (i.e., North Tropical Atlantic below 13.2° N). The yellow circle corresponds to the 100 km offshore Guadeloupe area in which the satellite-based Sargassum abundance is analysed."*

[Figure]

Figure 2: (a) Scheme of the decision tree classifier to predict Sargassum stranding probability. (b) Combination base of oceanic currents clusters labels obtained by KMS-ED from each stranding day to $\Delta t$ days before.

[Figure]

Figure 15: Decision Support System (DSS) results: probability of beaching obtained per module. Weekly probability to reach the maximum Sargassum abundance in the area 30-100 km offshore Guadeloupe for module A (blue line), stranding frequency per cluster for module C (orange line), match percentage for module D (yellow line), DSS Decision (black line). Day of observed beaching on Guadeloupe coasts (red dots): HYCOM (a) and Mercator (b).

**2) Strandings occurring in Guadeloupe are not affected by the dynamics of zones LA3 and LA1, but only by LA2. You should take it into account in the study.**

Our answer:

In the present study, the short-range transport from the LA2 zone is examined as well as the medium-range transport of Sargassum from the LA3 zone (i.e., the South of the Lesser Antilles Arc). Strandings occurring in Guadeloupe may be affected by dynamics of both zones LA2 and LA3.

**Although interesting, in its present form the overall approach needs to be better explained and the text is quite difficult to read. The language can be significantly improved and small typos removed.**

Our answer:

In the revised manuscript, we will try to clarify the text with language improvements and to better explain the overall approach. The following schematic will be added in the methods section.

[Figure]

*"Figure XX: The schematic of the adopted methodology."*

**Detailed comments**

The abstract

**L15 "small scale". Better use high resolution, as this is the crucial model configuration choice.**

Our answer:

Following your suggestion, *"small scale"* will be replaced by *"high resolution".*

**L20. Windward vs leeward. The only study citing this point is Marechal et al 2017.**

Our answer:

To correct this error, the following sentences L19-22:

*"During the periods 2011-2012, then 2014-2019, massive Sargassum strandings impacted most coasts of the Lesser Antilles (LA), mainly those facing east and southeast. LA received large amounts of algae on the windward Atlantic coastline, while leeward Caribbean coastal areas remained slightly affected (Franks et al., 2012, Gower et al., 2013, Johnson. et al., 2014, Hu et al., 2016, Wang and Hu, 2016, Maréchal et al., 2017)."*

will be changed to:

*"During the periods 2011-2012, then 2014-2019, massive Sargassum strandings impacted most coasts of the Lesser Antilles (LA), mainly those facing east and southeast (Franks et al., 2012, Gower et al., 2013, Johnson. et al., 2014, Hu et al., 2016, Wang and Hu, 2016). LA received large amounts of algae on the windward Atlantic coastline, while leeward Caribbean coastal areas remained slightly affected (Maréchal et al., 2017)."*

**L23 typo: also observed**

Following your suggestion, *"Strandings were also to be observed"* will be replaced by *"Strandings were also observed".*

**L41 wording: The volumes to be collected**

Following your suggestion, *"The volumes needed to be collected"* will be replaced by *"The volumes to be collected".*

**L55 There is also Jouanno et al 2020 (Env Res Letters) on the role of rivers.**

Our answer:

Following your suggestion, the reference "Jouanno et al. 2021" will be added here and in the references Section.

*Jouanno, J., Moquet, J.-S., Berline, L., Radenac, M.-H., Santini, W., Changeux, T., Thibaut, T., Podlejski W., Ménard, F., Martinez, J.-M., Aumont, O., Sheinbaum, J., Filizola N. and Moukandi N'Kaya G. D.: Evolution of the riverine nutrient export to the Tropical Atlantic over the last 15 years: is there a link with Sargassum proliferation?, Environ. Res. Lett., 16, 8 pp, https://doi.org/10.1088/1748-9326/abe11a, 2021.*

**L67 Unclear : The probability of a set of data…**

Our answer:

- Line 67: the sentence "*None of them used predictive modelling, including classifiers, to determine the probability of a set of data belonging to another set in order to discover repeatable patterns, allowing to produce a decision for risk prevention managers.*"

 will be replaced by:

*"None of them used predictive modelling based on a decision tree including current patterns and probabilities related to Sargassum strandings. This Sargassum beaching predictive tool based on repeatable current patterns would be useful for risk prevention managers."*

**L94 u and v: better zonal and meridional**

Following your suggestion, *"giving the u and v components"* will be replaced by *"giving the zonal (u) and meridional (v)"*.

**L91-L100 : You need to better describe the configuration of the reanalysis datasets you are using. In particular you should give the forcing fields (winds etc) and the data assimilated in each model. This is important as Mercator and Hycom models assimilate the same type of data (altimetry in particular), which largely explains their consistency in terms of large scale patterns (ie clusters).**

Our answer:

Line 91-100 the two paragraphs,

***2.1 HYCOM surface current dataset***

[revised manuscript text omitted]

will be replaced by

*"Surface wind data (at 1000 hPa) from the ERA-5 model were integrated with Mercator and HYCOM currents following this formula:"*

**L106 Berline et al 2020, not 2017**

Our answer:

We will fix this error in the revised manuscript.

*"Berline et al. (2017)"* will be replaced by *"Berline et al. (2020)".*

**L112. Check the year for Putman et al**

Our answer:

We will fix this error in the revised manuscript.

*"Putman et al. (2016)"* will be replaced by *"Putman et al. (2018)".*

**L121 Past tense is expected**

Our answer:

We will fix this error in the revised manuscript.

"we shall use" will be replaced by *"we used".*

**L125: 2.5.1: useless subsection**

Our answer:

Following your suggestion, the subsection "2.5.1 Clustering methods process" will be removed.

The paragraph *"Unsupervised learning methods such as Hierarchical Agglomerative Clustering (HAC) and K-means algorithms are used in the present study. Besides the measures and the classes of distance between objects*

*such as the Euclidean distance for K-means and the Ward's method for HAC, a new metric was also added (Biabiany et al. 2020). This metric integrates a set of knowledge about the dynamics of the data to be partitioned as well as their spatio-temporal properties. The result is an automated analysis with its own expertise on the input data."*

will be moved L125 after the sentence *"This method allowed significant improvement in clustering analysis dealing with climate data characterized by high spatio-temporal variability, such as precipitation (Biabiany et al., 2020)."*

**L132-133 Wording. Better : can lead to group different physical situations…**

Our answer:

We will fix this error in the revised manuscript.

*"can lead, within the same cluster, to gatherings of different physical situations"* will be replaced by *"can lead to group different physical situations within the same cluster."*

**L134. Unclear : " From L2"**

Our answer :

The sentences: *"The ED metric, which seems more suitable for this study, was used. L2 clustering methods can lead, within the same cluster, to gatherings of different physical situations (Biabiany et al., 2020).*

*To remove these biases linked with L2 clustering, the first step of the method used here is to consider the spatial variability in the dynamics of the analyzed daily surface currents from L2."*

will be replaced by

*"The ED metric, which seems more suitable for this study, was used. Clustering methods using euclidean distance (L2) can lead to group different physical situations within the same cluster (Biabiany et al., 2020)."*

**L139. Unclear terminology : Meshes. Better grid points. Need to be homogeneous throughout the text (grid cell at L205, grid points at L209)**

Our answer:

We will fix these errors in the revised manuscript.

L139-140: *"meshes"* will be replaced by *"grid points"*

L205: *"grid cell"* will be replaced by *"grid point".*

**The lines 141-150 on clustering should be grouped in a dedicated section.**

**The clarity and wording of this whole section must be improved. For instance " The similarity of the most similar fields (...) "Explanation should be given for one zone to avoid redundancy. A schematic would help. I do not clearly understand the algorithm for clustering. What is the role of the average divergence ?**

Our answer:

Line 141-150 the two paragraphs,

*"The second step was to group the information carried by the daily current velocity fields conditionally to the three given zones into histograms. The similarity of the most similar fields is estimated per pair and per zone based on the symmetrized Kullback-Leibler (KL) divergence computed from the histograms (Kullback and Leibler,*

*1951). This allows the entropy between two distributions to be expressed without having a priori reasoning concerning the probability distribution. The similarity between two histograms was quantified this way. The last step consisted in calculating the average of the divergence values for each zone. This allows to have a single value, named Expert Distance (ED) quantifying the similarity between the individuals of the database during clustering. The clustering results have been evaluated using the Silhouette Index (Rousseeuw, 1987).*

*The SaMk index defined in Biabiany et al. (2020) was used. This allows to express the quality of a clustering, by the average of the quality of each cluster, which is itself the average of the silhouette indices s(i) over the cluster elements. This index is defined as follows:"*

will be replaced by:

*"Clustering methods used in the present study were K-Means (KMS) and Agglomerative Hierarchical Clustering (HAC). Euclidean distance (L2) is usually computed to compare data in these algorithms. However, Biabiany et al. (2020) showed in a recent work that clustering results can be improved by using an Expert Distance (ED) to compare data. This ED is based on an empirical spatial subdivision and the use of Kullback-Leibler divergence, in order to quantify the similarity between two fields."*

To clarify this point, the following schematic will be added in the methods section.

[Figure]

*"Figure XX: The schematic of the Expert Distance process."*

**L152-162 (Section 2.6): wording and clarity should be improved. The word 'backward' is misleading here as there is no time integration in your analysis. You simply take the 30 days before one peculiar stranding event. What justifies the 30 days duration? Transport? Then is it consistent with the areas LA1, 2, 3?**

Our answer:

We will try to improve wording and clarity. The words *"the past stranding 30-day current backward sequences"* will be replaced by *"the 30-day sequences before beaching"*.

The following sentences will be added in the methods, L154 (Section 2.6):

*"The 30 days duration corresponds to the empirical transport time of a passive particle moving from the main entrance location of Sargassum rafts in the Lesser Antilles area (i.e., in LA3 zone, 8°N; -55°E) to Guadeloupe (i.e., LA2 zone). Based on the mean current magnitude of 0.2 m s$^{-1}$ (average value over the LA zone, in HYCOM and in Mercator data) and the distance of 500 km between the main entrance location and the Guadeloupe coasts, 29 days are obtained for the transport. For simplicity, the duration of 30 days was selected instead of 29 days."*

**L158-162. Unclear. "optimal matching methods" : which one ?**

**You compute a distance metric between the sequences of cluster numbers from previous section?**

Our answer:

The Longuest Common Subsequence (LCS) methods were used to compare the back-sequences. The sentence at L158 *"Dissimilarities between these backward sequences were calculated with optimal matching methods before dividing the population into several groups using a hierarchical classification (Larmarange et al., 2015)."*

can be replaced by: *"Dissimilarities between these backward sequences were calculated before dividing the population into several groups using a hierarchical classification (Larmarange et al., 2015)."*

**L161 Wald's or Ward ?**

Our answer: *We will fix this error in the revised manuscript. The right term is Ward.*

**L164 "At a given location" : which one ?**

Our answer: *This group of words has been deleted.*

**L168 : Why monthly? Are the stranding observations autocorrelated at this scale ?**

To improve the decision support system, the stranding monthly probability (i.e., Module A in the decision tree) will be replaced by a weekly probability of Sargassum presence in an area of 100 km radius offshore Guadeloupe. This probability is based on the cumulative 7-day Floating Algae density (Wang and Hu, 2016) estimated in this area during the two years 2019 and 2020.

The following section will be added in the "Datasets and method" section:

*"2.5 Satellite-based offshore abundance of Sargassum*

*Sargassum satellite observations were included in the present decision support system. To quantify the abundance of Sargassum in an area of 100 km radius offshore Guadeloupe, the 7-day Floating Algae (FA) density fields derived from the Alternative Floating Algae Index (Wang and Hu, 2016) were analyzed. As described by Trinanes et al. (2021), the 7-day Floating Algae (FA) density fields are accumulated on 7 days and have a 0.1° resolution. Due to optical complexity in nearshore waters, the FA density fields are masked with missing values within 30 km*

*from shoreline (Trinanes et al. 2021). The cumulative FA density values were summed in the area 30-100 km offshore Guadeloupe (Fig. 1) then weekly averaged during the two years 2019 and 2020."*

In the "Decision support system" section, the following sentence (L.168):

*"Module A takes as input the month of the selected day and returns the associated monthly probability (frequency) of stranding;"*

Will be replaced by:

*"Module A takes as input the week number of the selected day and returns the associated weekly probability to reach the maximum offshore abundance of Sargassum (based on observational FA density values during the two years 2019 and 2020)."*

**L170 L172 L176. Wording : "which " can be removed**

Our answer:  We will fix this error in the revised manuscript. *"which"* will be removed.

**L184 I understand you compute the average of P over an ensemble j pertaining to R. Use this notation then.**

Our answer: Following your remark, the equation (4) L184 will be replaced by:

"DECISION (i) = P(i) > MEAN( P(j) )

where j ∈ R, the set of past days (2019-2020),…"

**L191 Is it including windage ?**

Our answer: Yes it is including windage.

The caption of the Figure 3: "*Distributions of oceanic surface currents including windage for both models, HYCOM (blue) and Mercator (red) datasets.*"

will be replaced by:

"*Distributions of oceanic current magnitude including windage for both models, HYCOM (blue) and Mercator (red) datasets*"

**L191 "intensities ": better magnitude**

Our answer: Following your suggestion, *"intensities"* will be replaced by *"magnitude"*.

**L205 Mercator and Hycom outputs are not given on the same grid. Then how do you compare them? Should be explained in methods.**

Our answer:

As added above, the description of current data will be corrected. Mercator and HYCOM fields examined in the present study are given on the same 0.08-degree grid.

In "HYCOM surface current dataset" section, the sentences:

*"The HYCOM GLBy0.08 grid resolution is 0.08 degree in longitude and 0.04 degree in latitude. To perform the present study, the native HYCOM fields have been preliminarily interpolated on the Mercator uniform lon/lat 0.08-degree grid with a bilinear method."*

will be added.

**L205 : "At sea" : you mean offshore**

Our answer: Following your remark, *"At sea"* will be replaced by *"offshore".*

**L226 parangon**

Our answer: Following your remark, *"paragon"* has been replaced by *"parangon".*

**L241-242 should be in methods**

Our answer:

Following your suggestion, this part including matching formula (5) will be moved in method section before the section 2.6.

**L245 "most important": better highest**

Our answer: Following your suggestion *"most important"* has been replaced by *"highest".*

**L273 Where is the central Atlantic region used to quantify offshore abundance? Add it on the map.**

Our answer:

The offshore abundance from the satellite-based Sargassum Watch System (SaWS) will be replaced here by the monthly averaged offshore abundance of Sargassum (based on the floating algae density in the area 30-100 km offshore Guadeloupe).

Figs. 11 and 12 will be modified this way :

[Figure]

*Figure 11: Monthly distribution of cluster occurrence from Mercator outputs, from 2019 to 2020, in the Lesser Antilles (55-66°W, 8-17°N): MC1 (a), MC2 (b), MC3 (c) and MC4 (d). The red line shows the monthly distribution of Sargassum strandings on the coasts of Guadeloupe during the same period. The blue line indicates the monthly evolution of Sargassum abundance in the area 30-100 km offshore Guadeloupe normalized on the maximum value.*

[Figure]

*Figure 11: Monthly distribution of cluster occurrence from HYCOM outputs, from 2019 to 2020, in the Lesser Antilles (55-66°W, 8-17°N): HC1 (a), HC2 (b), HC3 (c) and HC4 (d). The red line shows the monthly distribution of Sargassum strandings on the coasts of Guadeloupe during the same period. The blue line indicates the monthly evolution of Sargassum abundance in the area 30-100 km offshore Guadeloupe normalized on the maximum value.*

Line 272, the sentences:

*"The monthly evolution of observed stranding days on the Guadeloupe coasts, the monthly evolution of Sargassum abundance over the Central Atlantic region (SaWS, https://optics.marine.usf.edu/projects/SaWS.html) were also analyzed on the focused period 2019-2020 (Figs. 11 and 12). During these two years, the amount of Sargassum over the Central Atlantic region increased significantly from February to July, then decreased from July to November."*

will be replaced by:

*"The monthly evolution of observed stranding days on the Guadeloupe coasts, the monthly evolution of Sargassum abundance in the area 30-100 km offshore Guadeloupe were also analyzed on the focused period 2019-2020 (Figs. 11 and 12). During these two years, the amount of Sargassum which may enhance the beaching risk in Guadeloupe increased significantly from February to May, then decreased from May to November."*

[Figure]

*"Figure 1: (a) Main oceanic currents occurring and interacting in the central Atlantic and the Lesser Antilles regions; Caribbean Current (CC), North Equatorial current (NEC), North Brazil current (NBC), North equatorial Counter Current (NECC), South Equatorial current (SEC). Lesser Antilles domain (LA): the red rectangle corresponds to the study area (55-66° W, 8-17° N); (b) Spatial subdivision of the study area into three sub-areas: LA1 (i.e., Caribbean Sea), LA2 (i.e., North Tropical Atlantic above Barbados (13.2° N)) and LA3 (i.e., North Tropical Atlantic below 13.2° N). The yellow circle corresponds to the 100 km offshore Guadeloupe area in which the satellite-based Sargassum abundance is analysed."*

**L276-281: Avoid redundancy.**

Our answer:

To avoid redundancy the sentence Line 279 : *"The pairs (MC1, HC2) and (MC3, HC3) include the greatest number of observed stranding days in Guadeloupe (Table 4)."* will be deleted.

**Discussion**

**L317 (Discussion) I suggest splitting this section. One for surface current and one for Sargassum stranding (L343-352).**

Our answer: According to your recommendations, this section will be splitted. A subsection will be added for *Sargassum* stranding.

**L334 North Current: You mean NEC?**

Our answer: Yes we mean NEC.

The sentence L344: "*The last identified factor is related to surface currents present in the North Atlantic region due to the North Equatorial Current and the associated gyre circulation.*"

will be replaced by: *"The last identified factor is related to the North Atlantic Gyre and the associated the North Equatorial Current. As the seasons change from winter to summer, the gyre shifts South by a few degrees in latitude."*

**L344 "Out of sync"?**

Our answer:

The sentence *"The monthly distribution of clusters and the distribution of observed strandings in Guadeloupe are out of sync."* will be removed.

**L345 remove "and**

Our answer:

Following your suggestion *"and"* will be removed here.

**L357 "independent variables": you mean explanatory**

Our answer:

Yes we mean explanatory: *"independent variables"* will be replaced by *"explanatory variables"*.

**L372 "ocean current 3D models": better ocean current reanalysis.**

Our answer:

We used analysis fields from HYCOM and Mercator models, not reanalysis. Following your suggestion, *"ocean current 3D models"* will be replaced by *"ocean current analysis"*.

**L404. This discussion of the appropriate time and space scale must be introduced earlier to justify the choices you made (30 days sequences, areas, monthly probability)**

Our answer:

As mentioned at the beginning of the document, we will justify our choices of time and space scale (30 days sequences, areas, monthly probability) in the methods section.

Areas are already described in the "2.5.2 Use of Expert Deviation in clustering algorithms" methods subsection, Lines 135-140:

*The LA study area was separated into three parts (Fig. 1b) based on the Sargassum rafts transport centers of action reported in the literature (Franks et al., 2016; Berline et al., 2020). To the west of LA, the first zone, LA1, is centered on the Caribbean Sea. To the east, the Atlantic zone has been split into two areas towards 13.5°N, just above Barbados island. To the south-east is the LA3 zone under the influence of the North Equatorial Recirculation Region (NERR) and its retroflection rings, while to the north-east is the LA2 zone, more representative of the North Equatorial Current. The analyzed daily fields include a total of 14 279 meshes (4 282 meshes in LA1, 3 407 meshes in LA2 and 4 536 meshes in LA3).*

Concerning the 30 days duration, the following sentences will be added in the methods, L154 (Section 2.6):

*"The 30 days duration corresponds to the empirical transport time of a passive particle moving from the main entrance location of Sargassum rafts in the Lesser Antilles area (i.e., in LA3 zone, 8°N; -55°E) to Guadeloupe (i.e., LA2 zone). Based on the mean current magnitude of 0.2 m s$^{-1}$ (average value over the LA zone, in HYCOM and in Mercator data) and the distance of 500 km between the main entrance location and the Guadeloupe coasts, 29 days are obtained for the transport. For simplicity, the duration of 30 days was selected instead of 29 days."*

Concerning the monthly probability, the following sentences will be added in the methods, L170 (Section 2.7), in Module A explanation:

*"The monthly period was selected to reduce possible biases associated with the uncertainties on the daily coastal observations. This sampling scale also allows comparisons with the monthly offshore Sargassum abundancy (SaWS, https://optics.marine.usf.edu/projects/SaWS.html) and with the monthly distribution of cluster occurrence."*

**Figures**

**Fig3 current magnitude**

Our answer: This figure will be corrected, "velocity" will be replaced by "current magnitude" on the x-axis.

[Figure]

The caption of the Figure 3: *"Distributions of oceanic surface currents including windage for both models, HYCOM (blue) and Mercator (red) datasets"*

will be replaced by

*"Distributions of oceanic surface current magnitudes including windage for both models, HYCOM (blue) and Mercator (red) datasets"*

**Fig5 Why not showing relative difference of magnitude, to see if Hycom is higher than Mercator for instance. What is the grid shown?**

Our answer:

Firstly, the grid shown is the same 0.08° grid for Mercator and HYCOM. Following your remarks, the Figure 5 will be modified including three panels: (a) Median of magnitude absolute differences (Mercator-HYCOM) (b), Median of magnitude relative differences (Mercator-HYCOM) and (c) Mode of current direction differences (Mercator-HYCOM).

[Figure]

The caption of the Figure 5: *"Comparison between Mercator and HYCOM surface currents from 2019 to 2020: current direction median differences in degree (a), current intensity median differences in m s-1 (b)."*

Will be replaced by

*"Comparison between Mercator and HYCOM surface currents from 2019 to 2020 on the same 0.08° grid: (a) median of magnitude absolute differences (Mercator-HYCOM) in m s⁻¹ and (b) median of magnitude relative differences (Mercator-HYCOM) in m s⁻¹ and (c) mode of current direction differences (Mercator-HYCOM) in degree."*

**Fig7 and 8. Mention Parangon as in the text. How is computed the stream function?**

Our answer:

Paragon has also been replaced by parangon. The stream function is calculated from the u and v components on the lon lat grid.

**For all figures showing clusters, in the tables and text: for clarity, you should rename the clusters from Mercator and Hycom to make similar patterns match. As HC1 is consistent with MC2, rename MC2 into MC1, etc.**

**This similarity of patterns is expected given the similarity of data assimilated into the two models.**

Our answer:

For greater clarity, we prefer keep this notation.

**Fig13 These are clusters of sequences. Mention it. Use same color as in figs 11-12**

Our answer:

The figure will be modified with same colors as figs 11-12.

The caption *"Figure 13: Distribution of current regimes over the 30-day stranding backward sequences: HC1 in green, HC2 in purple, HC3 in orange, and HC4 in yellow."*

Will be replaced by: *"Distribution of HYCOM current regime clusters (i.e., HC1 (in blue), HC2 (in green), HC3 (in orange), HC4 (in red)) in the 30-day sequence types (i.e., Seq1, Seq2, Seq3, Seq4)."*

**Fig 15. Time index: What is the corresponding date?**

Our answer: The corresponding dates will be added to the x-axis. The original figure will be modified with the new module A and the new testing period (i.e., all the year of 2021).

[Figure]

Figure 15: Decision Support System (DSS) results: probability of beaching obtained per module. Weekly probability to reach the maximum Sargassum abundance in the area 30-100 km offshore Guadeloupe for module A (blue line), stranding frequency per cluster for module C (orange line), match percentage for module D (yellow line), DSS Decision (black line). Day of observed beaching on Guadeloupe coasts (red dots): HYCOM (a) and Mercator (b).

**Table 6. Add recall.**

Our answer: The old table will be replaced by the following table including recall and time uncertainty ranges (i.e., +/-1 days, +/-2 days,…).

| Time range around D (day) | Datasets | TP (recall %) | TN (recall %) | FP (ratio %) | FN (ratio%) | Accuracy (ratio %) |
|---|---|---|---|---|---|---|
| **0** | **HYCOM** | 48 (61.5%) | 152 (53.1%) | 134 (36.8%) | 30 (8.2%) | 200 (54.9%) |
| | **Mercator** | 44 (56.4%) | 141 (49.3%) | 145 (39.8%) | 34 (9.3%) | 185 (50.8%) |
| **+/- 1** | **HYCOM** | 53 (67.9%) | 170 (59.4%) | (-) | (-) | (-) |
| | **Mercator** | 47 (60.3%) | 142 (49.6%) | (-) | (-) | (-) |
| **+/- 2** | **HYCOM** | 54 (69.2%) | 184 (64.3%) | (-) | (-) | (-) |
| | **Mercator** | 47 (60.3%) | 146 (51%) | (-) | (-) | (-) |
| **+/- 3** | **HYCOM** | 58 (74.4%) | 193 (67.5%) | (-) | (-) | (-) |
| | **Mercator** | 47 (60.3%) | 150 (52.4%) | (-) | (-) | (-) |

Table 6: Decision tree performance (with TP: True Positive, TN: True Negative, FP: False Positive, FN: False Negative, (-): same as above).

---

## Author Comment (AC2)

Dear Dr Nathan Putman, we thank you for your helpful comments and suggestions.

Your two last remarks dealing with the integration of *Sargassum* abundance in the decision support system and your suggested reference *"Trinanes et al. (2021)"* were very useful to strengthen and improve the predictive model. The predictive model originally based on circulation/wind/past-beachings was modified with a new module based on satellite observations which produces the weekly probability to reach the maximum observed cumulative floating algae density in an area of 100 km radius offshore Guadeloupe.

**1. Throughout the text:** *"Sargassum"* **is the genus name of the pelagic, brown algae discussed. Accordingly, it should be italicized wherever used.**

Our answer:

Following your suggestion, *"Sargassum"* will be italicized in the revised manuscript.

**2. Lines 79-80: Citing Putman et al. 2018 (already cited elsewhere) would be appropriate here as they model the % of *Sargassum* that follows these routes.**

Our answer:

Following your suggestion "Putman et al. 2018" will be cited here.

After the sentence (L79):*"The North Equatorial Current (NEC), the Guiana Current (GC), the eddies and the retroflection front of the North Brazil Current (NBC) are the main contributors of this transport.".*

The following sentence will be added:

*"Putman et al. (2018) modeled the percentage of Sargassum that follow these routes."*

**3. Lines 91-96: I am confused what HYCOM output you are using. What is reported here (GOMu0.04/expt_90.1m000 version) appears to only extend from latitude 18N to 32N and is thus outside of the area of this study. Can you please clarify? Did you run your own HYCOM at 1/25 degree resolution? The Global Analysis of HYCOM uses a grid of 0.04 degree longitude and 0.08 degree latitude, is this what you actually used?**

Our answer:

This was a mistake, we will correct it in the revised manuscript. The right version of HYCOM output used here is the HYCOM GLBy0.08 which has a grid resolution of 0.08 degree in longitude and 0.04 degree in latitude. To perform the present study, the native HYCOM fields have been preliminarily interpolated on the Mercator uniform lon/lat 0.08-degree grid with a bilinear method.

Line 91-96 :

*"**2.1 HYCOM surface current dataset**

*"Fine scale surface current data from the 1/25-degree HYCOM + NCODA Gulf of Mexico analysis model (GOMu0.04/expt_90.1m000 version, Hogan et al, 2014; Helber et al., 2013; Cummings and Smedstad, 2013; Cummings, 2005) between 1st January 2019 (i.e., available data starting date) and 31 December 2020 were analyzed. Daily 12Z fields giving the u and v components of the current at 50 cm depth were used. These fine*

*resolution current data were not used in previous studies dealing with Sargassum hazard (Putman et al., 2018; Johns et al., 2020)."*

will be replaced by:

**"2.1 HYCOM surface current dataset**

*Daily 12Z surface current components from the 41-layer HYCOM + NCODA global 1/12-degree analysis (HYCOM GLBy0.08 version), were examined. The HYCOM surface forcing including 10-m wind velocities are extracted from Climate Forecast System Version 2 (CFSv2). The Navy Coupled Ocean Data Assimilation (NCODA) system is used to assimilate available observational data: satellite altimeter sea surface height, satellite and in-situ sea surface temperature, temperature vertical profiles and salinity vertical profiles (Cummings, 2005; Cummings and Smedstad, 2013; Helber et al., 2013). The Bathymetry used is the GEBCO8 (Becker et al., 2009) with 30 arc second of resolution. The HYCOM GLBy0.08 grid resolution is 0.08 degree in longitude and 0.04 degree in latitude. To perform the present study, the native HYCOM fields have been preliminarily interpolated on the Mercator uniform lon/lat 0.08-degree grid with a bilinear method.".*

**4. Line 101-103: See also, Putman NF & He R (2013) Tracking the long-distance dispersal of marine organisms: sensitivity to ocean model resolution. Journal of the Royal Society Interface, 10:20120979**

Our answer: We thank you for this suggestion.

**5. Line 104: I am confused, what is the basis for assuming the "optimal factors of Cw = 0.01"? Surely this is not the case based on data from Johns et al. 2020, which showed no evidence that a windage factor of 1% was appropriate for *Sargassum*. They simply picked the "reasonable" value that has been used in the earlier publication Putman et al. 2018. The value of 1% was chosen by Putman et al. 2018 to test the sensitivity of model predictions to windage and did not claim that it was optimal (or even somewhat correct). Work since that point has been conducted which seems to suggest that the situation is somewhat more complicated, see Putman et al. 2020 (already cited elsewhere) and Johnson, D.R., Franks, J.S., Oxenford, H.A. and Cox, S.A.L., 2020. Pelagic *Sargassum* Prediction and Marine Connectivity in the Tropical Atlantic. Gulf and Caribbean Research, 31(1), pp.GCFI20-GCFI30. Whether the best windage value is 0, 0.5%, 1%, 3% or something else likely depends on the oceanographic region and the ocean circulation model and wind product used.**

Our answer:

We agree with this remark, the part Lines 104-108:

"*Surface wind influences the transport of floating seaweed rafts, with an optimal factor of $C_w$ = 0.01, which corresponds to the drag coefficient or windage, following Johns et al. (2020). A first clustering (KMS-L2) on Mercator analysis without windage had been proposed by Bernard et al. (2019). Berline et al. (2017), Putman et al. (2018) and Johns et al. (2020) have shown that the windage improves the Lagrangian simulations of Sargassum rafts transport in the Caribbean region. The windage was included in the present surface current clustering.*"

will be replaced by:

*"Surface wind influences the transport of floating seaweed rafts and a drag or windage coefficient must be added to the surface currents. The value of $C_w$ = 0.01 was used by Putman et al. (2018), Johns et al. (2020) and Berline et al. (2020). The use of other windage values should be investigated in a further study."*

**6. Line 112: I think that "Putman et al. (2016)" should be "Putman et al. (2018)"**

Our answer:

"Putman et al. (2016)" will be corrected to *"Putman et al. (2018)"*

**7. Line 206: change to "current speed differences are relatively small…"**

Our answer:

Line 206: *"current speed differences are small"*

will be replaced by

*"current speed differences are relatively small"*

**8. Line 334: change to "…due to the North Equatorial Current…"**

Our answer:

Line 334: *"The last identified factor is related to surface currents present in the North Atlantic region due to the North Current and the associated gyre circulation."*

will be replaced by

"*The last identified factor is related to surface currents present in the North Atlantic region due to the North Equatorial Current and the associated gyre circulation.*"

**9. Lines 360-361: Another issue may be that ocean current patterns may be highly important for "non-beaching" events (e.g., the currents are directed so that material doesn't reach the island), but for Sargassum to beach there needs to be Sargassum present. Thus, currents might be in a state to transport material to the island, but if there is no Sargassum present, there can be no beaching. Am I correct that this predictive model is based only on circulation/wind and not Sargassum abundance/coverage/distribution?**

Our answer :

Following your suggestion,

To improve the predictive model originally based on circulation/wind/past-beachings the module A producing the monthly probability of beaching was replaced by a new module based on satellite observations which produces the weekly probability to reach the maximum observed cumulative floating algae density in an area of 100 km radius offshore Guadeloupe.

Moreover, to strengthen the performance evaluation, the testing period was extended from the first four months of 2021 (i.e., from January 2021 to April 2021) to the full year of 2021 including seasonal variations of the offshore *Sargassum* abundance.

The performance evaluation of the classifier was also extended by adding three temporal uncertainty ranges around the decision day, respectively: +/-1 day, +/-2 days, +/-3 days. While the classifier may reproduce 61.5% of the observed beachings in 2021 with an accuracy lower than one day (this value reached 41.7% with the old module A and the limited testing period of four months), this recall score reaches 74.4% at +/-3 days accuracy.

The following section will be added in the "Datasets and method" section:

*"2.5 Satellite-based offshore abundance of Sargassum*

*Sargassum satellite observations were included in the present decision support system. To quantify the abundance of Sargassum in an area of 100 km radius offshore Guadeloupe, the 7-day Floating Algae (FA) density fields*

*derived from the Alternative Floating Algae Index (Wang and Hu, 2016) were analyzed. As described by Trinanes et al. (2021), the 7-day Floating Algae (FA) density fields are accumulated on 7 days and have a 0.1° resolution. Due to optical complexity in nearshore waters, the FA density fields are masked with missing values within 30 km from shoreline (Trinanes et al. 2021). The cumulative FA density values were summed in the area 30-100 km offshore Guadeloupe (Fig. 1) then weekly averaged during the two years 2019 and 2020.”*

In the "Decision support system" section, the following sentence (L.168):

*"Module A takes as input the month of the selected day and returns the associated monthly probability (frequency) of stranding;"*

Will be replaced by:

*"Module A takes as input the week number of the selected day and returns the associated weekly probability to reach the maximum offshore abundance of Sargassum (based on observational FA density values during the two years 2019 and 2020)."*

The figs 1, 2, 11, 12 and 15 will be modified as follows.

[Figure]

*"Figure 1: (a) Main oceanic currents occurring and interacting in the central Atlantic and the Lesser Antilles regions; Caribbean Current (CC), North Equatorial current (NEC), North Brazil current (NBC), North equatorial Counter Current (NECC), South Equatorial current (SEC). Lesser Antilles domain (LA): the red rectangle corresponds to the study area (55-66° W, 8-17° N); (b) Spatial subdivision of the study area into three sub-areas: LA1 (i.e., Caribbean Sea), LA2 (i.e., North Tropical Atlantic above Barbados (13.2° N)) and LA3 (i.e., North Tropical Atlantic below 13.2° N). The yellow circle corresponds to the 100 km offshore Guadeloupe area in which the satellite-based Sargassum abundance is analysed.”*

[Figure]

Figure 2: (a) Scheme of the decision tree classifier to predict *Sargassum* stranding probability. (b) Combination base of oceanic currents clusters labels obtained by KMS-ED from each stranding day to Δt days before.

[Figure]

*Figure 11: Monthly distribution of cluster occurrence from Mercator outputs, from 2019 to 2020, in the Lesser Antilles (55-66°W, 8-17°N): MC1 (a), MC2 (b), MC3 (c) and MC4 (d). The red line shows the monthly distribution*

[Figure]

*Figure 12: Monthly distribution of cluster occurrence from HYCOM outputs, from 2019 to 2020, in the Lesser Antilles (55-66°W, 8-17°N): HC1 (a), HC2 (b), HC3 (c) and HC4 (d). The red line shows the monthly distribution of Sargassum strandings on the coasts of Guadeloupe during the same period. The blue line indicates the monthly evolution of Sargassum abundance in the area 30-100 km offshore Guadeloupe normalized on the maximum value.*

[Figure]

Figure 15: Decision Support System (DSS) results: probability of beaching obtained per module. Weekly probability to reach the maximum *Sargassum* abundance in the area 30-100 km offshore Guadeloupe for module A (blue line), stranding frequency per cluster for module C (orange line), match percentage for module D (yellow

line), DSS Decision (black line). Day of observed beaching on Guadeloupe coasts (red dots): HYCOM (a) and Mercator (b).

Table 6 will be modified as follows.

| Time range around D (day) | Datasets | TP (recall %) | TN (recall %) | FP (ratio %) | FN (ratio%) | Accuracy (ratio %) |
|---|---|---|---|---|---|---|
| **0** | **HYCOM** | 48 (61.5%) | 152 (53.1%) | 134 (36.8%) | 30 (8.2%) | 200 (54.9%) |
| | **Mercator** | 44 (56.4%) | 141 (49.3%) | 145 (39.8%) | 34 (9.3%) | 185 (50.8%) |
| **+/- 1** | **HYCOM** | 53 (67.9%) | 170 (59.4%) | (-) | (-) | (-) |
| | **Mercator** | 47 (60.3%) | 142 (49.6%) | (-) | (-) | (-) |
| **+/- 2** | **HYCOM** | 54 (69.2%) | 184 (64.3%) | (-) | (-) | (-) |
| | **Mercator** | 47 (60.3%) | 146 (51%) | (-) | (-) | (-) |
| **+/- 3** | **HYCOM** | 58 (74.4%) | 193 (67.5%) | (-) | (-) | (-) |
| | **Mercator** | 47 (60.3%) | 150 (52.4%) | (-) | (-) | (-) |

Table 6: Decision tree performance (with TP: True Positive, TN: True Negative, FP: False Positive, FN: False Negative, (-): same as above).

**10. Lines 400-406: You may wish to draw reader's attention to the fact that there is considerable interest in monitoring and predicting coastal inundation by _Sargassum_. For instance, you may note how your smaller-scale study's goals might enhance the region-wide efforts such as the _Sargassum_ Inundation Reports (SIR) discussed here:**

**Trinanes J, Putman NF, Goni G, Hu C, Wang M (2021) Monitoring pelagic _Sargassum_ inundation potential for coastal communities. Journal of Operational Oceanography 14, in press (published online).**

Our answer:

Thank you for this useful suggestion. "Trinanes et al. (2021)" will be added to the references.

The following part will be added in the Introduction section (Line 65):

_"Trinanes et al. (2021) presented the Sargassum Inundation Reports (SIR), a product based on satellite observations to weekly predict Sargassum coastal inundation potential throughout the Caribbean Sea region, the Gulf of Mexico, and extending to the east coast of Florida and the Bahamas. As described by Trinanes et al. (2021), the SIR algorithm uses the Floating Algae density values within 50 km of each coastal pixel to predict three inundation potential levels (low, medium, and high). This algorithm does not include ocean currents, winds, and waves which may modify the movement of Sargassum._

The following sentence will be added in the Conclusion Line 406:

*"As the Sargassum Inundation Reports (Trinanes et al. (2021)), the present small-scale Sargassum beaching predictive model may contribute to the region-wide efforts to help coastal communities managing this hazard."*

---

## Author Comment (AC3)

Dear referee 2,

We thank you sincerely for your comments which helped us to improve the quality of the paper.

Firstly, we would like to draw your attention on some major changes we proposed to strengthen the evaluation of the decision tree classifier and to improve its recall scores. To strengthen the performance evaluation, the testing period was extended from the first four months of 2021 (i.e., from January 2021 to April 2021) to the full year of 2021 including seasonal variations of the offshore Sargassum abundance. To improve the recall score of the classifier, the module A producing the monthly probability of beaching was replaced by a new module based on satellite observations which produces the weekly probability to reach the maximum observed cumulative floating algae density in an area of 100 km radius offshore Guadeloupe. The performance evaluation of the classifier was also extended by adding three temporal uncertainty ranges around the decision day, respectively: +/-1 days, +/-2 days, +/-3 days. While the classifier may reproduce 61.5% of the observed beachings in 2021 with an accuracy lower than one day (this value reached 41.7% with the old module A and the limited testing period of four months), this recall score reaches 74.4% at +/-3 days accuracy.

Please find below our answers to your remarks (in bold). The proposed changes in the text are marked in red.

1. General comments

**The authors present a very interesting framework and method to better understand the ocean dynamics behind the strandings of Sargassum in the Lesser Antilles and to estimate their occurrence. The methodology presented is quite complex as well.**

**A better explanation of the methodology is necessary, especially for the oceanographic audience of this journal to adequately follow and understand this interesting study.**

**Section 2 I believe can be improved by making it easier for the reader to follow, especially the non-experts in these clustering methods. The technical details necessary for the reader to follow the study should be clearly described and the other details can be added as a section in supplementary material.**

Our answer: In the revised manuscript, we will try to clarify the text with language improvements and to better explain the overall approach. The following schematic will be added in the methods section to summarize the overall methodology.

[Figure]

*"Figure XX: The schematic of the adopted methodology."*

**A schematic of the method is given in fig. 2 for Section 2.7, but maybe a schematic for sections 2.5 and 2.6 could help too.**

Our answer: Following your suggestion, the two schematics, below, will be added in sections 2.5 and 2.6, respectively.

In Section 2.5:

[Figure]

*"Figure XX: The schematic of the Expert Distance process."*

In Section 2.6:

[Figure]

*"Figure XX: The schematic of the clustering process on the current sequences leading to Sargassum beachings."*

**In the discussion, I found that some comment on the impact (if any) of considering processes other than windage (e.g. presence of nutrients, sinking of Sargassum, waves?) could have on an even better understanding of the Sargassum strandings, was missing.**

Our answer:

Following your suggestion we will add this sentence at the end of the discussion (L365, end of section 4.3):

*"The present study does not take into account the effects of other factors (e.g., presence of nutrient, sinking of Sargassum and waves) which would allow a more realistic understanding of the Sargassum strandings."*

2. Specific comments

**L23: "Strandings were also be observed in Africa (Széchy et al., 2012)." Why mention the occurrence of strandings in Africa? Any connection with the Caribbean strandings? Did the Sargassum strandings also cause natural hazards on the African coast?**

Our answer:

This sentence will be removed.

**L61: "MODIS AFAI satellite images", please define/describe**

Our answer:

*"A combination of MODIS AFAI satellite images"* will be replaced by:

*"A combination of satellite-based Alternative Floating Algae Index (AFAI, Wang and Hu, 2016) fields"*

**L66-68: Could be useful to include some references of the methodology here.**

Our answer:

The sentence "*None of them used predictive modelling, including classifiers, to determine the probability of a set of data belonging to another set in order to discover repeatable patterns, allowing to produce a decision for risk prevention managers.*" Will be modified as below:

*"None of them used predictive modelling (Geisser, 1993; Kuhn and Johnson, 2013) including classifiers (Friedl and Brodley, 1997), to determine the probability of a set of data belonging to another set in order to discover repeatable patterns, allowing to produce a decision for risk prevention managers."*

These references will be included to improve the understanding.

*Friedl, M. A., and Brodley, C. E.: Decision tree classification of land cover from remotely sensed data, Remote Sensing of Environment, vol. 61, Issue 3, pp. 399-409, ISSN:0034-4257, https://doi.org/10.1016/S0034-4257(97)00049-7, 1997.*

*Geisser, S.: Predictive Inference: An Introduction, Chapman & Hall, ISBN: 978-0-412-03471-8, 1993.*

*Kuhn, M., and Johnson, K.: Applied predictive modeling. New York, Springer. isbn:978-1-4614-6848-6, doi:10.1007/978-1-4614-6849-3, 2013.*

**L69: A general definition of predictive modelling is missing in the introduction for the readers which do not know about this method and how it compares with a conventional forecast. For example could be included here (Line 69).**

Our answer:

The following sentences will be added here L69:

*"Predictive modelling refers to mathematical and computational methods for predicting future events on the analysis of the statistical patterns in the input dataset (Geisser, 1993; Friedl and Brodley, 1997; Kuhn and Johnson, 2013). Compared to other conventional forecast, predictive modelling methods requiring low computational costs are characterized by their flexibility, and their intuitive simplicity (Friedl and Brodley, 1997)."*

**L75-76: "To optimize the final partitioning, an additional metric based on the Kullback Leiber divergence (Kulback and Leibler, 1951, Biabiany et al., 2020) will be included" : quite specific on the methodology, for readers not familiarised with this method it could be hard to follow in this point in the introduction. More general details can be given, or this point can be moved to the methods section.**

Our answer:

Following your remark, this sentence will be removed. The Expert Distance process with the Kullback Leiber divergence will be explained in the method section (2.5).

**L82-83: "This ocean region corresponds to the CA and TA1 boxes in Johns (2020)", maybe say approximately corresponds, as not exactly the same. The LA3 region goes further south and LA2 and LA3 go until -55°E, whales region TA1 till -50°E. Most importantly, why choose the study regions to correspond to CA and TA1 boxes from Johns (2020)?**

 Our answer:

The study regions were not choose to correspond to CA and TA1 boxes from Johns (2020). The sentence *"This ocean region corresponds to the CA and TA1 boxes in Johns (2020)."* will be removed.

**L96-96: From what I understand this dataset was not used before to simulate Sargassum trajectories, but was it used in any other Lagrangian study? Any validation studies done on the velocity outputs of this dataset?**

Our answer:

Firstly, there was a mistake about the HYCOM output resolution, we will correct it in the revised manuscript. The right version of HYCOM output used here is the HYCOM GLBy0.08 which has a grid resolution of 0.08 degree in longitude and 0.04 degree in latitude. To perform the present study, the native HYCOM fields have been preliminarily interpolated on the Mercator uniform lon/lat 0.08-degree grid with a bilinear method.

[revised manuscript text omitted]

**L101: "Comparison between HYCOM and Mercator results" Do you mean the results from the Sargassum trajectories or a comparison of the velocity outputs of these datasets?**

Our answer: Because of the previous changes in HYCOM and Mercator outputs description, the sentence *"Comparison between HYCOM and Mercator results would help to better understand the effects of spatial resolution on surface current patterns in the focused region"* will be removed in the revised manuscript.

**Section 2.3: What is the spatial and temporal resolution of the ERA-5 wind dataset?**

Our answer:

The ERA-5 wind dataset has a spatial resolution of 31 km and hourly fields are available.

Line 108, the part *"Surface wind data (at 1000 hPa) from the ERA-5 model for the time period 2019 to 2020 were integrated with Mercator currents following this formula:"*
will be replaced by

*"The daily 12h (UTC) surface wind data (at 1000 hPa) from the 31-km scale ERA-5 model were integrated with Mercator and HYCOM currents following this formula:"*

**L128: "Ward's method for HAC" Please explain and add reference.**

Our answer:

The sentence *"Besides the measures and the classes of distance between objects such as the Euclidean distance for K-means and the Ward's method for HAC, a new metric was also added (Biabiany et al. 2020)"*

*Will be modified as below*

*"Besides the measures and the classes of distance between objects such as the Euclidean distance for K-means and the Ward's method* which allows to identify homogeneous subsets of data (Ward, 1963), *a new metric was also added (Biabiany et al. 2020)"*

This reference will be added

*J. H. Ward Jr. (1963) Hierarchical Grouping to Optimize an Objective Function, Journal of the American Statistical Association, 58:301, 236-244, DOI:10.1080/01621459.1963.10500845.*

**L129-130: "with its own expertise on the input data" What do you mean by these? Please provide further explanations. Also, the new method name is not specified at all in section 2.5.1, and it will help for the reader to better follow the methodology. This section is only 5 lines long, more details on the process of the clustering methods could be given.**

Our answer:

The sentence "*The result is an automated analysis with its own expertise on the input data.*" Will be removed.

Moreover to help the reader better following the methodology the section "2.5.1 Clustering methods process" will be removed.

The paragraph (L126-129) *"Unsupervised learning methods such as Hierarchical Agglomerative Clustering (HAC) and K-means algorithms are used in the present study. Besides the measures and the classes of distance between objects such as the Euclidean distance for K-means and the Ward's method for HAC, a new metric was also added (Biabiany et al. 2020). This metric integrates a set of knowledge about the dynamics of the data to be partitioned as well as their spatio-temporal properties."*

will be moved L125 after the sentence *"This method allowed significant improvement in clustering analysis dealing with climate data characterized by high spatio-temporal variability, such as precipitation (Biabiany et al., 2020)."*

**L132: "L2 clustering methods…" Please explain L2 in this context.**

The sentences: *"The ED metric, which seems more suitable for this study, was used. L2 clustering methods can lead, within the same cluster, to gatherings of different physical situations (Biabiany et al., 2020).*

*To remove these biases linked with L2 clustering, the first step of the method used here is to consider the spatial variability in the dynamics of the analyzed daily surface currents from L2."*

will be replaced by

*"The ED metric, which seems more suitable for this study, was used. Clustering methods using euclidean distance (L2) can lead to group different physical situations within the same cluster (Biabiany et al., 2020)."*

**L133: "gatherings of different physical situations". What do you mean by this? Maybe give an example of physical situations for this particular study scenario.**

Our answer:

We will clarify this part in the revised manuscript.

The part *"can lead, within the same cluster, to gatherings of different physical situations"* will be replaced by *"can lead to group different physical situations within the same cluster."*

**You refer to this in the next phrase as "biases". Is there then a tendency towards a specific physical situation?**

Our answer:

There is no trend towards a specific physical situation. These side effects have been described in Biabiany et al. (2020).

As stated above, the sentence (L133)  will be removed in the revised manuscript.

**L134: "spatial variability" : At what scales?**

Our answer: As stated above, the sentence (L133)  will be removed in the revised manuscript.

**L139-L140: "The analyzed daily fields include a total of 14 279 meshes (4 282 meshes in LA1, 3 407 meshes in LA2 and 4 536 meshes in LA3). The remainder corresponds to land areas." What do you refer to here with meshes?**

Our answer:

L139-140: *"meshes"* will be replaced by *"grid points"*

**The land areas then correspond to Sargassum strandings? For clarity, these details could be described in a dataset section better, rather than in the middle of the methods description.**

Our answer:

The land areas do not correspond to Sargassum strandings, they correspond to areas over land (e.g., islands).

To clarify this point, the sentence *"The remainder corresponds to land areas"* will be replaced by *"The remainder corresponds to areas over land (e.g., islands)."*

**L141-142: "The second step was to group the information carried by the daily current velocity fields conditionally to the three given zones into histograms." More details on histograms, for example binning, velocity data from HYCOM and Mercator?**

Our answer:

Histogram bins are given in Table 1 and Fig. 4. The values correspond to deciles from HYCOM and Mercator datasets.

The sentence *"The second step was to group the information carried by the daily current velocity fields conditionally to the three given zones into histograms." Will be modified as follows*

*"The second step was to group the information carried by the daily current velocity fields conditionally to the three given zones into histograms (Table 1, Fig.4)."*

**L158: "optimal matching methods" Please explain and add some references.**

Our answer:

The words *"optimal matching methods"* will be removed.

The sentence at L158 *"Dissimilarities between these backward sequences were calculated with optimal matching methods before dividing the population into several groups using a hierarchical classification (Larmarange et al., 2015)."*

will be replaced by:

*"Dissimilarities between these backward sequences were calculated before dividing the sequences dataset into several groups using a hierarchical classification (Larmarange et al., 2015)."*

**L158: "dividing the population" what do you refer to exactly here by population? Population of strandings or backward sequences?**

Our answer :

To clarify this point, the words *"dividing the population"* will be replaced by *"dividing the sequences dataset"*

**L160-162: Please give further details (maybe as supplementary material?) and add more references**.

Our answer :

*"Wald's algorithm"* will be corrected to *"Ward algorithm"* already described (Line 128) as a *"method which allows to identify homogeneous subsets of data (Ward, 1963)"*.

Ward, J. H. Jr.: Hierarchical Grouping to Optimize an Objective Function, Journal of the American Statistical Association, 58:301, 236-244, DOI: 10.1080/01621459.1963.10500845, 1963.

**L186-L187: "was experimented on the first 120 days…". Was experimented to…? Recall aim of doing these tests. Also why 120 days and during this period of time? Could results vary a lot if done during the northern hemisphere Summer months instead?**

Our answer:

To strengthen the performance evaluation, the testing period was extended from the first four months of 2021 (i.e., from January 2021 to April 2021) to the full year of 2021 including seasonal variations of the offshore Sargassum abundance.

The sentence *"The proposed tree in Fig. 2 was experimented on the first 120 days of the year 2021, from 1 st January 2021 to 30 April 2021, i.e., 120 tests."* Will be replaced by

*"The proposed tree in Fig. 2 was tested on the full year of 2021 (from 1ˢᵗ January to 31 December), i.e., 360 tests."*

The original Fig. 2 will be replaced by the following figure:

[Figure]

Figure 2: (a) Scheme of the decision tree classifier to predict Sargassum stranding probability. (b) Combination base of oceanic currents clusters labels obtained by KMS-ED from each stranding day to Δt days before.

The Decision support system results figure will be replaced by the following figure:

[Figure]

Figure 15: Decision Support System (DSS) results: probability of beaching obtained per module. Weekly probability to reach the maximum Sargassum abundance in the area 30-100 km offshore Guadeloupe for module A (blue line), stranding frequency per cluster for module C (orange line), match percentage for module D (yellow line), DSS Decision (black line). Day of observed beaching on Guadeloupe coasts (red dots): HYCOM (a) and Mercator (b).

**L190: Can maybe start section 3.1 giving some context on why this analysis is done**.

Our answer:

The following sentence will be added L190 at the beginning of the section 3.1:

*"In view of the lack of study dealing with surface current patterns in the Lesser Antilles area, this preliminary analysis is presented here."*

**L191: "90% of them remain below 0.65 m/s". For both models exactly same?**

Our answer :

The sentence "*For both models HYCOM and Mercator, the velocity intensities do not exceed 2.57 m s-1 and 90% of them remain below 0.65 m s$^{-1}$*"

*will be replaced by*

*"For both models HYCOM and Mercator, the velocity intensities do not exceed 2.57 m s$^{-1}$ and 90% of them remain below 0.65 m s$^{-1}$ (the respective 90th centile values are respectively 0.6515 m.s$^{-1}$ and 0.6458 m.s$^{-1}$ for HYCOM and Mercator )."*

**L193: Figure 3 distributions how are they calculated? With histograms? Kernel Density Estimator or something else applied to obtain this "smooth" distribution curves?**

Our answer :

After the sentence (L193): *"Figure 3 shows skewed distributions with skewness equal to 1.31 and 1.21."*

The following sentence will be added *"A normal kernel was used to obtain these distributions."*

**L194-L195: 5 times greater for both models?**

Our answer :

The sentence *"There are extreme values indicating surface current speeds with 195 deviations 5 times greater than the standard deviation."* Will be removed.

**L207-208: what are the implications of these differences?**

Our answer:

Firstly, the original Fig. 5 will be replaced by the following figure:

[Figure]

Figure 5: Comparison between Mercator and HYCOM surface currents from 2019 to 2020 on the same 0.08° grid: (a) median of magnitude absolute differences (Mercator-HYCOM) in m s⁻¹ and (b) median of magnitude relative differences (Mercator-HYCOM) in m s⁻¹ and (c) mode of current direction differences (Mercator-HYCOM) in degree.

The sentence *"The largest differences, above 0.5 m s-1 are observed in the South part of the LA arc, around Trinidad and Tobago."* Will be replaced by the following sentence:

*"In the South part of the LA arc, around Trinidad and Tobago, Mercator current magnitudes are globally higher than HYCOM current magnitudes. Thus, Mercator surface currents might induce higher Sargassum influx from the Western Central Atlantic to the Caribbean Sea in this area."*

**L272-L273: "The monthly evolution of observed stranding days on the Guadeloupe coasts, the monthly evolution of Sargassum abundance over the Central Atlantic region (SaWS, https://optics.marine.usf.edu/projects/SaWS.html)" I imagine it should be: "Guadeloupe coasts and the monthly evolution…", to make clear you talking about two datasets. The observed stranding dataset is mentioned in the dataset section (section 2.4), but not the Sargassum abundance over the Central Atlantic region.**

Our answer:

Line 272, the sentences:

*"The monthly evolution of observed stranding days on the Guadeloupe coasts, the monthly evolution of Sargassum abundance over the Central Atlantic region (SaWS, https://optics.marine.usf.edu/projects/SaWS.html) were also analyzed on the focused period 2019-2020 (Figs. 11 and 12). During these two years, the amount of Sargassum over the Central Atlantic region increased significantly from February to July, then decreased from July to November."*

will be replaced by:

*"The monthly evolution of observed stranding days on the Guadeloupe coasts, the monthly evolution of Sargassum abundance in the area 30-100 km offshore Guadeloupe were also analyzed on the focused period 2019-2020 (Figs. 11 and 12). During these two years, the amount of Sargassum which may enhance the beaching risk in Guadeloupe increased significantly from February to May, then decreased from May to November."*

Figs. 11 and 12 will be replaced by the following figures:

[Figure]

*Figure 11: Monthly distribution of cluster occurrence from Mercator outputs, from 2019 to 2020, in the Lesser Antilles (55-66°W, 8-17°N): MC1 (a), MC2 (b), MC3 (c) and MC4 (d). The red line shows the monthly distribution of Sargassum strandings on the coasts of Guadeloupe during the same period. The blue line indicates the monthly evolution of Sargassum abundance in the area 30-100 km offshore Guadeloupe normalized on the maximum value.*

[Figure]

*Figure 12: Monthly distribution of cluster occurrence from HYCOM outputs, from 2019 to 2020, in the Lesser Antilles (55-66°W, 8-17°N): HC1 (a), HC2 (b), HC3 (c) and HC4 (d). The red line shows the monthly distribution of Sargassum strandings on the coasts of Guadeloupe during the same period. The blue line indicates* *the monthly evolution of Sargassum abundance in the area 30-100 km offshore Guadeloupe normalized on the maximum value.*

3. Technical corrections

**Please write Sargassum in italics, like it is done in other studies like for example Johns et al., (2020), as you are writing its scientific name, and even if it is just the genus in this case.**

Our answer: Following your suggestion, *"Sargassum"* will be italicized in the revised manuscript.

**L10: "including windage effect": gives the impression the HYCOM and Mercator datasets already include the windage effect, when you actually added separately. Please improve phrasing.**

Our answer:

To clarify this point, the sentence:

*"The input surface currents including windage effect were derived from the Mercator model and the Hybrid Coordinate Ocean Model (HYCOM)."*

will be replaced by

*"The input surface currents were derived from the Mercator model and the Hybrid Coordinate Ocean Model (HYCOM) outputs in which we integrated the windage effect."*

**L20: "LA received…" to "The LA received…"**

Our answer: *"LA received"* will be replaced by *"The LA received"*

**L23: "…were also be observed…" to "…were also observed…"**

Our answer: *"were also be observed"* will be replaced by *"were also observed"*

**L46: Improve sentence, e.g. "… multi-year reanalysis of wind and current, and numerical models, both the role of subsurface nutrient supply and surface current transport were estimated."**

Our answer:

As you suggested the part:

*"...multi-year reanalysis of wind and current, numerical models estimated both the role of subsurface nutrient supply and surface current transport."*

will be replaced by

*"...multi-year reanalysis of wind and current, and numerical models, both the role of subsurface nutrient supply and surface current transport were estimated."*

**L50: "Sargassum Watch System SaWS" to "*Sargassum* Watch System (SaWS)"**

Our answer: *"Sargassum Watch System SaWS"* will be replaced by *"Sargassum Watch System (SaWS)"*

**L83: "in Johns (2020)" et al. missing.**

Our answer: *"in Johns (2020)"* will be replaced by *" in Johns et al. (2020)"*

**L92: Please define the abbreviations HYCOM and NCODA (HYCOM defined in abstract but not in the main text)**

Our answer: The pa

*2.1 HYCOM surface current dataset*

"*Fine scale surface current data from the 1/25-degree HYCOM + NCODA Gulf of Mexico analysis model (GOMu0.04/expt_90.1m000 version, Hogan et al, 2014; Helber et al., 2013; Cummings and Smedstad, 2013; Cummings, 2005) between 1st January 2019 (i.e., available data starting date) and 31 December 2020 were analyzed. Daily 12Z fields giving the u and v components of the current at 50 cm depth were used. These fine resolution current data were not used in previous studies dealing with Sargassum hazard (Putman et al., 2018; Johns et al., 2020).*

will be replaced by:

*"2.1 HYCOM surface current dataset*

*Daily 12Z surface current components from the 41-layer* Hybrid Coordinate Ocean Model (HYCOM) *global 1/12-degree analysis (HYCOM GLBy0.08 version), were examined. The HYCOM surface forcing including 10-m wind velocities are extracted from Climate Forecast System Version 2 (CFSv2).* The Navy Coupled Ocean Data Assimilation (NCODA) *system …*

**L94: Please define 12Z fields.**

Our answer:

*"12Z"* will be replaced by *"12 UTC (i.e., Coordinated Universal Time) "*

**L94-95: "u and v components" to "zonal (u) and meridional (v) velocity components"**

Our answer: *"u and v components"* will be replaced by *"zonal (u) and meridional (v) velocity components"*

**L101-102: "Comparison…in the focused region" to "A comparison.. in the study region."**

Our answer:

This sentence *"Comparison between HYCOM and Mercator results would help to better understand the effects of spatial resolution on surface current patterns in the focused region."* Will be removed in the revised manuscript.

**L107: "Sargassum raft transport", maybe trajectories instead of transport is more appropriate?**

Our answer:

Following your suggestion the part "*Lagrangian simulations of Sargassum rafts transport in the Caribbean region.*"

Will be replaced by

*"Lagrangian simulations of Sargassum raft trajectories in the Caribbean region."*

**L112-113: "The region analyzed in the present work corresponds to the CA - TA1 region defined in Johns et al. (2020)" already mentioned in L82-83, is it necessary to repeat here?**

Our answer:

This sentence will be removed.

**L116-117: "This period includes 730 observational days with 110 days of observed strandings." , phrasing not clear do you mean that out of the total 730 days of data, only 110 days included observations of Sargassum strandings?**

Our answer:

During the two years 2019-2020, only 110 days of observed beachings in Guadeloupe have been recorded.

To clarify this point,

The following sentence:

*This period includes 730 observational days with 110 days of observed strandings."*

Will be replaced by

*"During this period of 730 days, only 110 days of observed beaching have been recorded (i.e., 30 days in 2019 and 80 days in 2020). During the year of 2021, 78 days of beaching were observed in Guadeloupe."*

**L137: "above Barbados island" to "above the island of Barbados"**

Our answer: *"above Barbados island"* will be replaced by *"above the island of Barbados"*

**L142: "The similarity of the most similar fields is estimated per pair.." Improve phrasing. What do you refer to exactly? Per pair of Sargassum meshes?**

Our answer:

Line 141-150 the two paragraphs,

*"The second step was to group the information carried by the daily current velocity fields conditionally to the three given zones into histograms. The similarity of the most similar fields is estimated per pair and per zone based on the symmetrized Kullback-Leibler (KL) divergence computed from the histograms (Kullback and Leibler, 1951). This allows the entropy between two distributions to be expressed without having a priori reasoning concerning the probability distribution. The similarity between two histograms was quantified this way. The last step consisted in calculating the average of the divergence values for each zone. This allows to have a single value, named Expert Distance (ED) quantifying the similarity between the individuals of the database during clustering.*

*The clustering results have been evaluated using the Silhouette Index (Rousseeuw, 1987). The SaMk index defined in Biabiany et al. (2020) was used. This allows to express the quality of a clustering, by the average of*

*the quality of each cluster, which is itself the average of the silhouette indices s(i) over the cluster elements. This index is defined as follows:"*

will be replaced by:

*"Clustering methods used in the present study were K-Means (KMS) and Agglomerative Hierarchical Clustering (HAC). Euclidean distance (L2) is usually computed to compare data in these algorithms. However, Biabiany et al. (2020) showed in a recent work that clustering results can be improved by using an Expert Distance (ED) to compare data. This ED is based on an empirical spatial subdivision and the use of Kullback-Leibler divergence, in order to quantify the similarity between two fields."*

To clarify this part, the following schematic will be added in the methods section.

[Figure]

*"Figure XX: The schematic of the Expert Distance process."*

**L148: "The SaMk index" to "The Silhouette (SaMk) index"**

Our answer: *"The SaMk index"* will be replaced by *"The Silhouette (SaMk) index"*

**L151: Define all variables of equation 2!**

Our answer:

This sentence will be added below the equation (2) :

where *k* is the number of clusters, *Cj* the set of days from the cluster *j*, *i* a day form *Cj* and *s(i)* the silhouette index (Rousseeuw, 1987) value of day *i*.

**L153-154: Improve phrasing.**

The sentence: *"To better understand current regime dynamics which may lead to Sargassum strandings on the coasts of Guadeloupe, the past stranding 30-day current backward sequences were analyzed."*

Will be replaced by

*"To better understand current dynamics which may lead to Sargassum beaching in Guadeloupe, we analyzed the 30-day current sequences before beaching."*

**L156: "January 2020" to "January 2019"**

 Our answer: *"January 2020"* will be replaced by *"January 2019"*

**L165-L166: " surface currents with windage effects (Mercator, HYCOM and ERA-5)" to " surface currents (Mercator and HYCOM) with windage effects (ERA-5)"**

Our answer: " *surface currents with windage effects (Mercator, HYCOM and ERA-5)*" will be replaced by *" surface currents (Mercator and HYCOM) with windage effects (ERA-5)"*

**L186-L187: "The proposed tree in Fig. 2…". Move to new line, to separate it from the phrase explaining the terms in equation (4)**

Our answer: The sentence starting with *"The proposed tree in Fig. 2…"* will be moved to new line.

**L191: "do not exceed 2.57 m/s". Maybe better to say the maximum is 2.57 m/s, if not it sounds like 2.57 m/s is a key velocity value that should not be exceeded for some reason.**

Our answer: " *For both models HYCOM and Mercator, the velocity magnitudes do not exceed 2.57 m s$^{-1}$ and 90% of them remain below 0.65 m s$^{-1}$.*"

Will be replaced by

*"For both models HYCOM and Mercator, the maximum surface velocity is 2.57 m s$^{-1}$ and 90% of them remain below 0.65 m s$^{-1}$."*

**L193-L194: add at end to which model it each value corresponds to e.g. ".. for HYCOM and Mercator, respectively."**

Our answer: *"Figure 3 shows skewed distributions with skewness equal to 1.31 and 1.21*"

Will be replaced by

*"Figure 3 shows skewed distributions with skewness equal to 1.31 and 1.21 for HYCOM and Mercator, respectively."*

**L205: "Globally, at sea, the current.." Is it necessary to specify at sea? What do you exactly mean with at sea here, open ocean?**

Our answer: *"at sea"* will be replaced by *"at open ocean"*

**L210: "into three magnitude groups of 45°" to "into three magnitude groups of 45° intervals"?**

Our answer: *"into three magnitude groups of 45°"* will be replaced by *"into three magnitude groups of 45° intervals"*

**L215: Improve phrasing, gives the impression you used equation (1) to perform the clustering.**

Our answer: the part *"according to equation (1)"* will be removed.

**L244: "Table 3 shows results" to "Table 3 shows the results"**

Our answer: *"Table 3 shows results"* will be replaced by *"Table 3 shows the results"*

**L297: "remain with probabilities" add probabilities of… Help the reader follow better your study, recalling details**

Our answer:

This part will be modified with new results produced by our improved version of Decision Support System. Please find below the modified Table 6 which includes recalling details.

| Time range around D (day) | Datasets | TP (recall %) | TN (recall %) | FP (ratio %) | FN (ratio%) | Accuracy (ratio %) |
|---|---|---|---|---|---|---|
| **0** | **HYCOM** | 48 (61.5%) | 152 (53.1%) | 134 (36.8%) | 30 (8.2%) | 200 (54.9%) |
| | **Mercator** | 44 (56.4%) | 141 (49.3%) | 145 (39.8%) | 34 (9.3%) | 185 (50.8%) |
| **+/- 1** | **HYCOM** | 53 (67.9%) | 170 (59.4%) | (-) | (-) | (-) |
| | **Mercator** | 47 (60.3%) | 142 (49.6%) | (-) | (-) | (-) |
| **+/- 2** | **HYCOM** | 54 (69.2%) | 184 (64.3%) | (-) | (-) | (-) |
| | **Mercator** | 47 (60.3%) | 146 (51%) | (-) | (-) | (-) |
| **+/- 3** | **HYCOM** | 58 (74.4%) | 193 (67.5%) | (-) | (-) | (-) |
| | **Mercator** | 47 (60.3%) | 150 (52.4%) | (-) | (-) | (-) |

Table 6: Decision tree performance (with TP: True Positive, TN: True Negative, FP: False Positive, FN: False Negative, (-): same as above).

**L317: Improve wording of Section 4.2 title, for example can simply remove "hazard"**

Our answer: "Hazard" will be removed. The new Section 4.2 title will be: *"4.2. Surface current analysis applied to Sargassum"*

**L320: "retroflexion" to "retroflection"**

Our answer: "retroflexion" will be replaced by *"retroflection"*

**L345 "The first peak of strandings, in March and seems.." to "The first peak of strandings, in March, seems.."**

Our answer: *"The first peak of strandings, in March and seems.."* will be replaced by *"The first peak of strandings, in March, seems.."*

**L373: Write as K-Means, and also in L217, write method in the same way.**

Our answer: *"k-mean"* will be replaced by *"K-Means"* (L217 and L373)

4. Figures and tables

**Figure 2: Describe BASE abbreviation as in L175.**

Our answer:

*"BASE"* is not an abbreviation, to avoid misunderstanding, the term will be written in the normal case (i.e., Base or base). Base will be explained both in the Fig. 2a and in the caption.

[Figure]

*Figure 2: (a) Scheme of the decision tree classifier to predict Sargassum stranding probability. (b) Combination base of oceanic currents clusters labels obtained by KMS-ED from each stranding day to Δt days before. Base is the set of 30-day current sequences before Sargassum beaching.*

**Figures 4, 9 and 10: x-axis tick labels not clear, please improve.**

Our answer: The x-axis tick labels will be improved as below.

Figures 4, 9 and 10 will be replaced by the following figures.

[Figure]

Figure 4: Relative frequency distribution of current speeds for the three offshore sub-regions around the Lesser Antilles (2019-2020), LA1 (blue), LA2 (red), LA3 (yellow). (a) Mercator with ERA-5 windage and (b) HYCOM with ERA-5 windage.

[Figure]

Figure 9: Relative frequency distribution of current speeds for the three offshore sub-regions: MC1 (a), MC2 (b), MC3 (c) and MC4 (d). The representative elements were obtained after KMS-ED clustering for Mercator.

[Figure]

Figure 10: Relative frequency distribution of current speeds for the three offshore sub-regions: HC1 (a), HC2 (b), HC3 (c) and HC4 (d). The representative elements were obtained after KMS-ED clustering for HYCOM.

**Table 1: Header mean to Mean**

Our answer: *"mean"* will be replaced by *"Mean"*

| Deciles (Dᵢ) | 0.1 | 0.2 | 0.3 | 0.4 | 0.5 | 0.6 | 0.7 | 0.8 | 0.9 | Max | Mean | Sigma |
|---|---|---|---|---|---|---|---|---|---|---|---|---|
| Mercator (m s⁻¹) | 0.11 | 0.16 | 0.20 | 0.24 | 0.28 | 0.32 | 0.39 | 0.48 | 0.65 | 2.57 | 0.33 | 0.22 |
| HYCOM (m s⁻¹) | 0.13 | 0.18 | 0.23 | 0.28 | 0.32 | 0.38 | 0.44 | 0.52 | 0.65 | 2.49 | 0.36 | 0.21 |

Table 1: Boundaries of the histogram classes used to quantify surface currents velocity data with Sigma as Standard deviation.

**Table 5: Caption mention what n and % refer to exactly.**

Our answer:

Table 5 will be modified as shown below:

| 30-day current sequence before beaching (HYCOM) | Seq1 | Seq2 | Seq3 | Seq4 |
|---|---|---|---|---|
| Number (n) | 18 | 40 | 7 | 42 |
| Ratio (%) | 16.8 | 37.4 | 6.5 | 39.3 |

The caption will be modified as below:

*Table 5: Distribution of backward sequence clusters, with (n) corresponding to the number of sequences in each cluster and (%) corresponding to the ratio of the number of sequences on the total.*

---

## Author Response (AR1)

**Dear Editor,**

The manuscript has been largely modified following suggestions and comments from the two referees and from community. Please, find below our answers (in blue) to each comment (in black) and the changes made (in red).

**Answers to RC1**

1) The discussion of the appropriate time and space scale must be introduced in the methods to justify the choices made (30 days sequences, areas, monthly probability).

We justified our choices of time and space scale (30 days sequences, areas, monthly probability) in the methods section.

Areas were described, L163 (Section 2.6):

*"The LA study area was separated into three parts (Fig. 1b, Fig. 3) based on the Sargassum rafts transport centers of action reported in the literature (Franks et al., 2016; Berline et al., 2020). To the west of LA, the first zone, LA1, is centered on the Caribbean Sea. To the east, the Atlantic zone was split into two areas towards 13.5°N, just above the island of Barbados. To the south-east is the LA3 zone under the influence of the North Equatorial Recirculation Region (NERR) and its retroflection rings, while to the north-east is the LA2 zone, more representative of the North Equatorial Current. The analyzed daily fields include a total of 14 279 grid points (4 282 grid points in LA1, 3 407 grid points in LA2 and 4 536 grid points in LA3). The remainder corresponds to areas over land (e.g., islands)."*

Concerning the 30 days duration, the following sentences were added in the methods, L183 (Section 2.7):

*"The 30 days duration corresponds to the empirical transport time of a passive particle moving from the main entrance location of Sargassum rafts in the Lesser Antilles area (i.e. in LA3 zone, 8° N; -55° E) to Guadeloupe (i.e. LA2 zone). Based on the mean current magnitude of 0.2 m s-1 (average value over the LA zone, in HYCOM and in Mercator data) and the distance of 500 km between the main entrance location and the Guadeloupe coasts, 29 days are obtained for the transport. For simplicity, the duration of 30 days was selected instead of 29 days."*

To improve the decision support system, the stranding monthly probability (i.e., Module A in the decision tree) was replaced by a daily probability of Sargassum presence in an area of 100 km radius offshore Guadeloupe. This probability is based on the cumulative 7-day Floating Algae density (Wang and Hu, 2016) estimated in this area during the two years 2019 and 2020.

The following section will be added in the "Datasets and method" section (L145):

*"2.5 Satellite-based offshore abundance of Sargassum*

*Sargassum satellite observations were included in the present decision support system. To quantify the abundance of Sargassum in an area of 100 km radius offshore Guadeloupe, the 7-day Floating Algae (FA) density fields derived from the Alternative Floating Algae Index (Wang and Hu, 2016) were analyzed. As described by Trinanes et al. (2021), the 7-day Floating Algae (FA) density fields are accumulated on 7 days and have a 0.1° resolution. Due to optical complexity in nearshore waters, the FA density fields are masked with missing values within 30 km from shoreline (Trinanes et al. 2021). The cumulative FA density values were added up in the area 30-100 km offshore Guadeloupe (Fig. 1) then averaged over the two years 2019 and 2020 for each day."*

In the "Decision support system" section, the following sentence (L.168):

*"Module A takes as input the month of the selected day and returns the associated monthly probability (frequency) of stranding;"*

was replaced L199 (section 2.8) by:

*"Module A takes as input the week number of the selected day and returns the associated daily probability to reach the maximum offshore abundance of Sargassum (based on observational FA density values during the two years 2019 and 2020);"*

The following figures were modified

[Figure]

Figure 1: (a) Main oceanic currents occurring and interacting in the central Atlantic and the Lesser Antilles regions; Caribbean Current (CC), North Equatorial current (NEC), North Brazil current (NBC), North equatorial Counter Current (NECC), South Equatorial current (SEC). Lesser Antilles domain (LA): the red rectangle corresponds to the study area (55-66° W, 8-17° N); (b) Spatial subdivision of the study area into three sub-areas: LA1 (i.e. Caribbean Sea), LA2 (i.e. North Tropical Atlantic above Barbados (13.2° N)) and LA3 (i.e. North Tropical Atlantic below 13.2° N). The yellow circle corresponds to the 100 km offshore Guadeloupe area in which the satellite-based Sargassum abundance is analysed.

[Figure]

Figure 5: (a) Scheme of the decision tree classifier to predict Sargassum beaching probability. (b) Combination base of oceanic current clusters labels obtained by KMS-ED from each beaching day to $\Delta t$ days before.

[Figure]

Figure 18: Decision Support System (DSS) results: probability of beaching obtained per module. Daily probability to reach the maximum *Sargassum* abundance in the area 30-100 km offshore Guadeloupe for module A (blue line), beaching frequency per cluster for module C (orange line), match percentage for module D (yellow line), DSS Decision (black line). Days of observed beaching on Guadeloupe coasts (red dots): HYCOM (a) and Mercator (b).

**2) Strandings occurring in Guadeloupe are not affected by the dynamics of zones LA3 and LA1, but only by LA2. You should take it into account in the study.**

In the present study, the short-range transport from the LA2 zone is examined as well as the medium-range transport of Sargassum from the LA3 zone (i.e., the South of the Lesser Antilles Arc). Beachings occurring in Guadeloupe may be affected by dynamics of both zones LA2 and LA3.

**Although interesting, in its present form the overall approach needs to be better explained and the text is quite difficult to read. The language can be significantly improved and small typos removed.**

In the revised manuscript, we tried to clarify the text with language improvements and to better explain the overall approach. The following schematic was added in Section 2 "Datasets and methods".

[Figure]

Figure 2: The schematic of the overall methodology.

**Detailed comments**

**The abstract**

**L15 "small scale". Better use high resolution, as this is the crucial model configuration choice.**

Following your suggestion, *"small scale"* was replaced by *"high resolution"*.

**L20. Windward vs leeward. The only study citing this point is Marechal et al 2017.**

To correct this error, the following sentences L19-22:

*"During the periods 2011-2012, then 2014-2019, massive Sargassum strandings impacted most coasts of the Lesser Antilles (LA), mainly those facing east and southeast. LA received large amounts of algae on the windward Atlantic coastline, while leeward Caribbean coastal areas remained slightly affected (Franks et al., 2012, Gower et al., 2013, Johnson. et al., 2014, Hu et al., 2016, Wang and Hu, 2016, Maréchal et al., 2017)."*

was changed to (L20):

*"During the periods 2011-2012, then 2014-2019, massive Sargassum beachings impacted most coasts of the Lesser Antilles (LA), mainly those facing east and southeast (Franks et al., 2012, Gower et al., 2013, Johnson. et al., 2014, Hu et al., 2016, Wang and Hu, 2016). The LA received large amounts of*

*algae on the windward Atlantic coastline, while leeward Caribbean coastal areas remained slightly affected (Maréchal et al., 2017)."*

**L23 typo: also observed**

This sentence was removed.

**L41 wording: The volumes to be collected**

Following your suggestion, *"The volumes needed to be collected"* was replaced by *"The volumes to be collected"*.

**L55 There is also Jouanno et al 2020 (Env Res Letters) on the role of rivers.**

Following your suggestion, the reference "Jouanno et al. 2021b" was added here and in the references Section.

*Jouanno, J., Moquet, J.-S., Berline, L., Radenac, M.-H., Santini, W., Changeux, T., Thibaut, T., Podlejski W., Ménard, F., Martinez, J.-M., Aumont, O., Sheinbaum, J., Filizola N. and Moukandi N'Kaya G. D.: Evolution of the riverine nutrient export to the Tropical Atlantic over the last 15 years: is there a link with Sargassum proliferation?, Environ. Res. Lett., 16, 8 pp, https://doi.org/10.1088/1748-9326/abe11a, 2021b.*

**L67 Unclear : The probability of a set of data…**

- Line 67: the sentence *"None of them used predictive modelling, including classifiers, to determine the probability of a set of data belonging to another set in order to discover repeatable patterns, allowing to produce a decision for risk prevention managers."*

was replaced by L74:

*None of them used predictive modelling (Geisser, 1993; Kuhn and Johnson, 2013) including classifiers (Friedl and Brodley, 1997) to determine the probability of repeatable patterns in a dataset so as to produce a decision for risk prevention managers."*

**L94 u and v: better zonal and meridional**

This part was removed.

**L91-L100 : You need to better describe the configuration of the reanalysis datasets you are using. In particular you should give the forcing fields (winds etc) and the data assimilated in each model. This is important as Mercator and Hycom models assimilate the same type of data (altimetry in particular), which largely explains their consistency in terms of large scale patterns (ie clusters).**

Line 91-100 the two paragraphs,

*2.1 HYCOM surface current dataset*

[revised manuscript text omitted]

**L139. Unclear terminology : Meshes. Better grid points. Need to be homogeneous throughout the text (grid cell at L205, grid points at L209)**

This error was fixed in the revised manuscript.

In the revised manuscript *"meshes"* was replaced by *"grid points"* and *"grid cell"* was replaced by *"grid point"*.

**The lines 141-150 on clustering should be grouped in a dedicated section.**

**The clarity and wording of this whole section must be improved. For instance " The similarity of the most similar fields (...) "Explanation should be given for one zone to avoid redundancy. A schematic would help. I do not clearly understand the algorithm for clustering. What is the role of the average divergence ?**

This part was replaced (L153-162) by:

*"Unsupervised learning methods such as Hierarchical Agglomerative Clustering (HAC) and K-means algorithms were used in the present study. Besides the measures and the classes of distance between objects such as the Euclidean distance for K-means and the Ward method which allows to identify homogeneous subsets of data (Ward, 1963), a new metric was also added. The Expert Distance (ED) which integrates image analysis within unsupervised learning methods (Clustering) was used. This method allowed significant improvement in clustering analysis dealing with climate data characterized by high spatio-temporal variability, such as precipitation (Biabiany et al., 2020). Clustering methods using Euclidean distance (L2) can lead to group different physical situations within the same cluster (Biabiany et al., 2020). The ED metric integrates a set of knowledge about the dynamics of the data to be partitioned as well as their spatio-temporal properties.*

*This ED is based on an empirical spatial subdivision and the use of Kullback-Leibler divergence, in order to quantify the similarity between two fields. Figure 3 shows the schematic of the Expert Distance process adopted here."*

To clarify this point, the following schematic will be added in the methods section.

[Figure]

Figure 3: The schematic of the Expert Distance process.

**L152-162 (Section 2.6): wording and clarity should be improved. The word 'backward' is misleading here as there is no time integration in your analysis. You simply take the 30 days before one peculiar stranding event. What justifies the 30 days duration? Transport? Then is it consistent with the areas LA1, 2, 3?**

We tried to improve wording and clarity. The words *"the past stranding 30-day current backward sequences"* was replaced by *"the 30-day sequences before beaching"*.

The following sentences will be added in the methods, L183 (Section 2.7):

*"The 30 days duration corresponds to the empirical transport time of a passive particle moving from the main entrance location of Sargassum rafts in the Lesser Antilles area (i.e. in LA3 zone, 8° N; -55° E) to Guadeloupe (i.e. LA2 zone). Based on the mean current magnitude of 0.2 m s-1 (average value over the LA zone, in HYCOM and in Mercator data) and the distance of 500 km between the main entrance location and the Guadeloupe coasts, 29 days are obtained for the transport. For simplicity, the duration of 30 days was selected instead of 29 days."*

**L158-162. Unclear. "optimal matching methods" : which one ?**

**You compute a distance metric between the sequences of cluster numbers from previous section?**

The Longuest Common Subsequence (LCS) methods were used to compare the back-sequences.

The sentence at L158 *"Dissimilarities between these backward sequences were calculated with optimal matching methods before dividing the population into several groups using a hierarchical classification (Larmarange et al., 2015)."*

was replaced by: *"Dissimilarities between these sequences were calculated before dividing the sequences dataset into several groups using a hierarchical classification (Larmarange et al., 2015). The Longest Common Subsequence (LCS) method was used to compute the distances between the sequences (Elzinga and Struder, 2015; Studer and Ritschard, 2016)."*(L190)

**L161 Wald's or Ward ?**

This error was fixed in the revised manuscript.

*Wald's* was replaced by *Ward.*

**L164 "At a given location" : which one ?**

This group of words was deleted.

**L168 : Why monthly? Are the stranding observations autocorrelated at this scale ?**

To improve the decision support system, the stranding monthly probability (i.e., Module A in the decision tree) was replaced by a daily probability of Sargassum presence in an area of 100 km radius offshore Guadeloupe. This probability is based on the cumulative 7-day Floating Algae density (Wang and Hu, 2016) estimated in this area during the two years 2019 and 2020.

The following section will be added in the "Datasets and method" section:

*2.5 Satellite-based offshore abundance of Sargassum*

*Sargassum satellite observations were included in the present decision support system. To quantify the abundance of Sargassum in an area of 100 km radius offshore Guadeloupe, the 7-day Floating Algae (FA) density fields derived from the Alternative Floating Algae Index (Wang and Hu, 2016) were analyzed. As described by Trinanes et al. (2021), the 7-day Floating Algae (FA) density fields are accumulated on 7 days and have a 0.1° resolution. Due to optical complexity in nearshore waters, the FA density fields are masked with missing values within 30 km from shoreline (Trinanes et al. 2021).*

*The cumulative FA density values were added up in the area 30-100 km offshore Guadeloupe (Fig. 1) then averaged over the two years 2019 and 2020 for each day.*

In the "Decision support system" section, the following sentence (L.168):

*"Module A takes as input the month of the selected day and returns the associated monthly probability (frequency) of stranding;"*

was replaced L199 (section 2.8) by:

*"Module A takes as input the week number of the selected day and returns the associated daily probability to reach the maximum offshore abundance of Sargassum (based on observational FA density values during the two years 2019 and 2020);"*

**L170 L172 L176. Wording : "which " can be removed**

This error was fixed in the revised manuscript.

 *"which"* was removed.

**L184 I understand you compute the average of P over an ensemble j pertaining to R. Use this notation then.**

Following your remark, the equation (4) L184 will be replaced by:

"DECISION (i) = P(i) > MEAN( P(j) )

where j ∈ R, the set of past days (2019-2020),…"

**L191 Is it including windage ?**

Yes it is including windage.

The caption of the Figure 3: "*Distributions of oceanic surface currents including windage for both models, HYCOM (blue) and Mercator (red) datasets.*"

was replaced by:

*Figure 6: Distributions of oceanic surface current magnitudes including windage for both models, HYCOM (blue) and Mercator (red) datasets.*

**L191 "intensities ": better magnitude**

Following your suggestion, *"intensities"* was replaced by *"magnitude"*.

**L205 Mercator and Hycom outputs are not given on the same grid. Then how do you compare them?**

**Should be explained in methods.**

The description of current data was corrected. Mercator and HYCOM fields examined in the present study are given on the same 0.08-degree grid.

In "HYCOM surface current dataset" section, the sentences:

*"The HYCOM GLBy0.08 grid resolution is 0.08 degree in longitude and 0.04 degree in latitude. To perform the present study, the native HYCOM fields were preliminarily interpolated on the Mercator uniform lon/lat 0.08-degree grid with a bilinear method."*

were added L115.

**L205 :  "At sea" : you mean offshore**

Following your remark, *"At sea"* was replaced by *"at open ocean"*.

**L226 parangon**

*"paragon"* was replaced by *"parangon"*.

**L241-242 should be in methods**

Following your suggestion, this part including matching formula was moved in method section (L178).

**L245 "most important": better highest**

*"most important"* was replaced by *"highest"*.

**L273 Where is the central Atlantic region used to quantify offshore abundance? Add it on the map.**

The offshore abundance from the satellite-based Sargassum Watch System (SaWS) was replaced here by the monthly averaged offshore abundance of Sargassum (based on the floating algae density in the area 30-100 km offshore Guadeloupe).

The corresponding figures were modified this way:

[Figure]

Figure 14: Monthly distribution of cluster occurrence from Mercator outputs, from 2019 to 2020, in the Lesser Antilles (55-66°W, 8-17°N): MC1 (a), MC2 (b), MC3 (c) and MC4 (d). The red line shows the monthly distribution of *Sargassum* beachings on the coasts of Guadeloupe during the same period. The blue line indicates the monthly evolution of *Sargassum* abundance in the area 30-100 km offshore Guadeloupe normalized on the maximum value.

[Figure]

Figure 15: Monthly distribution of cluster occurrence from HYCOM outputs, from 2019 to 2020, in the Lesser Antilles (55-66°W, 8-17°N): HC1 (a), HC2 (b), HC3 (c) and HC4 (d). The red line shows the monthly distribution of Sargassum beachings on the coasts of Guadeloupe during the same period. The blue line indicates the monthly evolution of Sargassum abundance in the area 30-100 km offshore Guadeloupe normalized on the maximum value.

Line 272, the sentences:

*"The monthly evolution of observed stranding days on the Guadeloupe coasts, the monthly evolution of Sargassum abundance over the Central Atlantic region (SaWS, https://optics.marine.usf.edu/projects/SaWS.html) were also analyzed on the focused period 2019-2020 (Figs. 11 and 12). During these two years, the amount of Sargassum over the Central Atlantic region increased significantly from February to July, then decreased from July to November."*

were replaced by:

*"The monthly evolution of observed stranding days on the Guadeloupe coasts, the monthly evolution of Sargassum abundance in the area 30-100 km offshore Guadeloupe were also analyzed on the focused period 2019-2020 (Figs. 14 and 15). During these two years, the amount of Sargassum likely to enhance the beaching risk in Guadeloupe increased significantly from February to May, then decreased from May to November."* (L304)

**L276-281: Avoid redundancy.**

To avoid redundancy the sentence: *"The pairs (MC1, HC2) and (MC3, HC3) include the greatest number of observed stranding days in Guadeloupe (Table 4)."* was deleted.

**Discussion**

**L317 (Discussion) I suggest splitting this section. One for surface current and one for Sargassum stranding (L343-352).**

According to your recommendations, this section was splitted.

**L334 North Current: You mean NEC?**

Yes we mean NEC.

The sentence L344: "*The last identified factor is related to surface currents present in the North Atlantic region due to the North Equatorial Current and the associated gyre circulation.*"

was replaced by: *"The last identified factor is related to the North Atlantic Gyre and the associated North Equatorial Current."* L370

**L344 "Out of sync"?**

The sentence *"The monthly distribution of clusters and the distribution of observed strandings in Guadeloupe are out of sync."* Was removed.

**L345 remove "and**

Following your suggestion *"and"* was removed here.

**L357 "independent variables": you mean explanatory**

Yes we mean explanatory.

*"independent variables"* was replaced by *"explanatory variables"*.

**L372 "ocean current 3D models": better ocean current reanalysis.**

We used analysis fields from HYCOM and Mercator models, not reanalysis.

Following your suggestion, *"ocean current 3D models"* was by *"ocean current analysis"*.

**L404. This discussion of the appropriate time and space scale must be introduced earlier to justify the choices you made (30 days sequences, areas, monthly probability)**

We justified our choices of time and space scale (30 days sequences, areas, monthly probability) in the methods section.

Areas were described, L163 (Section 2.6):

*"The LA study area was separated into three parts (Fig. 1b, Fig. 3) based on the Sargassum rafts transport centers of action reported in the literature (Franks et al., 2016; Berline et al., 2020). To the west of LA, the first zone, LA1, is centered on the Caribbean Sea. To the east, the Atlantic zone was split into two areas towards 13.5°N, just above the island of Barbados. To the south-east is the LA3 zone under the influence of the North Equatorial Recirculation Region (NERR) and its retroflection rings, while to the north-east is the LA2 zone, more representative of the North Equatorial Current. The analyzed daily fields include a total of 14 279 grid points (4 282 grid points in LA1, 3 407 grid points in LA2 and 4 536 grid points in LA3). The remainder corresponds to areas over land (e.g., islands)."*

Concerning the 30 days duration, the following sentences were added in the methods, L183 (Section 2.7):

*"The 30 days duration corresponds to the empirical transport time of a passive particle moving from the main entrance location of Sargassum rafts in the Lesser Antilles area (i.e. in LA3 zone, 8° N; -55° E) to Guadeloupe (i.e. LA2 zone). Based on the mean current magnitude of 0.2 m s-1 (average value over the LA zone, in HYCOM and in Mercator data) and the distance of 500 km between the main*

*entrance location and the Guadeloupe coasts, 29 days are obtained for the transport. For simplicity, the duration of 30 days was selected instead of 29 days.”*

To improve the decision support system, the stranding monthly probability (i.e., Module A in the decision tree) was replaced by a daily probability of Sargassum presence in an area of 100 km radius offshore Guadeloupe. This probability is based on the cumulative 7-day Floating Algae density (Wang and Hu, 2016) estimated in this area during the two years 2019 and 2020.

The following section will be added in the "Datasets and method" section (L145):

*"**2.5 Satellite-based offshore abundance of Sargassum***

*Sargassum satellite observations were included in the present decision support system. To quantify the abundance of Sargassum in an area of 100 km radius offshore Guadeloupe, the 7-day Floating Algae (FA) density fields derived from the Alternative Floating Algae Index (Wang and Hu, 2016) were analyzed. As described by Trinanes et al. (2021), the 7-day Floating Algae (FA) density fields are accumulated on 7 days and have a 0.1° resolution. Due to optical complexity in nearshore waters, the FA density fields are masked with missing values within 30 km from shoreline (Trinanes et al. 2021). The cumulative FA density values were added up in the area 30-100 km offshore Guadeloupe (Fig. 1) then averaged over the two years 2019 and 2020 for each day."*

In the "Decision support system" section, the following sentence (L.168):

*"Module A takes as input the month of the selected day and returns the associated monthly probability (frequency) of stranding;"*

was replaced L199 (section 2.8) by:

*"Module A takes as input the week number of the selected day and returns the associated daily probability to reach the maximum offshore abundance of Sargassum (based on observational FA density values during the two years 2019 and 2020);"*

**Figures**

**Fig3 current magnitude**

This figure was corrected.

"velocity" was replaced by "current magnitude" on the x-axis.

[Figure]

The caption of the Figure 3: *"Distributions of oceanic surface currents including windage for both models, HYCOM (blue) and Mercator (red) datasets"*

was replaced by

*"Figure 6: Distributions of oceanic surface current magnitudes including windage for both models, HYCOM (blue) and Mercator (red) datasets."*

**Fig5 Why not showing relative difference of magnitude, to see if Hycom is higher than Mercator for instance. What is the grid shown?**

Following your remarks, the Figure 5 was modified including three panels: (a) Median of magnitude absolute differences (Mercator-HYCOM) (b), Median of magnitude relative differences (Mercator-HYCOM) and (c) Mode of current direction differences (Mercator-HYCOM).

[Figure]

The caption of the Figure 5: *"Comparison between Mercator and HYCOM surface currents from 2019 to 2020: current direction median differences in degree (a), current intensity median differences in m s-1 (b)."*

was replaced by

*Figure 8: Comparison between Mercator and HYCOM surface currents from 2019 to 2020 on the same 0.08° grid: (a) median of magnitude absolute differences (Mercator-HYCOM) in m s-1 and (b) median of magnitude relative differences (Mercator-HYCOM) in m s-1 and (c) mode of current direction differences (Mercator-HYCOM) in degree.*

**Fig7 and 8. Mention Parangon as in the text. How is computed the stream function?**

Paragon was replaced by parangon.

The stream function is calculated from the u and v components on the lon lat grid.

**For all figures showing clusters, in the tables and text: for clarity, you should rename the clusters from Mercator and Hycom to make similar patterns match. As HC1 is consistent with MC2, rename MC2 into MC1, etc.**

**This similarity of patterns is expected given the similarity of data assimilated into the two models.**

For greater clarity, we prefer keep this notation.

**Fig13 These are clusters of sequences. Mention it. Use same color as in figs 11-12**

The figure was modified with same colors as figs 11-12.

[Figure]

The caption *"Figure 13: Distribution of current regimes over the 30-day stranding backward sequences: HC1 in green, HC2 in purple, HC3 in orange, and HC4 in yellow."*

was be replaced by: *"Figure 16: Distribution of HYCOM current regime clusters (i.e. HC1 (in blue), HC2 (in green), HC3 (in orange), HC4 (in red)) in the 30-day sequence types (i.e. Seq1, Seq2, Seq3, Seq4)."*

**Fig 15. Time index: What is the corresponding date?**

The corresponding dates were added to the x-axis. The original figure was modified with the new module A and the new testing period (i.e., all the year of 2021).

[Figure]

Figure 18: Decision Support System (DSS) results: probability of beaching obtained per module. Daily probability to reach the maximum Sargassum abundance in the area 30-100 km offshore Guadeloupe for module A (blue line), beaching frequency per cluster for module C (orange line), match percentage for module D (yellow line), DSS Decision (black line). Days of observed beaching on Guadeloupe coasts (red dots): HYCOM (a) and Mercator (b).

**Table 6. Add recall.**

The old table will be replaced by the following table including recall and time uncertainty ranges (i.e., +/-1 days, +/-2 days,…).

| Time range around D (day) | Datasets | True positive (recall %) | True negative (recall %) | Accuracy (ratio %) |
|---|---|---|---|---|
| **0** | **HYCOM** | **46 (59.0%)** | 151 (52.8%) | **197 (54.1%)** |
| | **Mercator** | **43 (55.1%)** | 141 (49.3%) | **184 (50.6%)** |
| **+/- 1** | **HYCOM** | 52 (66.7%) | 175 (61.2%) | 227 (62.4%) |
| | **Mercator** | 47 (60.3%) | 151 (52.8%) | 198 (54.4%) |
| **+/- 2** | **HYCOM** | 55 (70.5%) | 189 (66.1%) | 244 (67.0%) |
| | **Mercator** | 51 (65.4%) | 155 (54.2%) | 206 (56.6%) |
| **+/- 3** | **HYCOM** | **57 (73.1%)** | 198 (69.2%) | **255 (70.1%)** |
| | **Mercator** | **51 (65.4%)** | 161 (56.3%) | **212 (58.2%)** |

Table 6: Decision tree performance scores.

**Answers to RC2**

**1. General comments**

**The authors present a very interesting framework and method to better understand the ocean dynamics behind the strandings of Sargassum in the Lesser Antilles and to estimate their occurrence. The methodology presented is quite complex as well.**

**A better explanation of the methodology is necessary, especially for the oceanographic audience of this journal to adequately follow and understand this interesting study.**

**Section 2 I believe can be improved by making it easier for the reader to follow, especially the non-experts in these clustering methods. The technical details necessary for the reader to follow the study should be clearly described and the other details can be added as a section in supplementary material.**

In the revised manuscript, we tried to clarify the text with language improvements and to better explain the overall approach.

The following schematic was added in the methods section to summarize the overall methodology.

[Figure]

Figure 2: The schematic of the overall methodology.

**A schematic of the method is given in fig. 2 for Section 2.7, but maybe a schematic for sections 2.5 and 2.6 could help too.**

Following your suggestion, the two schematics, below, were added respectively.

In Section 2.6:

[Figure]

Figure 3: The schematic of the Expert Distance process.

In Section 2.7:

[Figure]

Figure 4: The schematic of the clustering process on the current sequences leading to beachings.

**In the discussion, I found that some comment on the impact (if any) of considering processes other than windage (e.g. presence of nutrients, sinking of Sargassum, waves?) could have on an even better understanding of the Sargassum strandings, was missing.**

Following your suggestion, we added this sentence at the end of the discussion (L410, end of section 4.3):

*"The present study does not take into account the effects of other factors (e.g., presence of nutrient, sinking of algae and waves) which would allow a more realistic understanding of the Sargassum beachings."*

**2. Specific comments**

**L23: "Strandings were also be observed in Africa (Széchy et al., 2012)." Why mention the occurrence of strandings in Africa? Any connection with the Caribbean strandings? Did the Sargassum strandings also cause natural hazards on the African coast?**

This sentence was removed.

**L61: "MODIS AFAI satellite images", please define/describe**

*"A combination of MODIS AFAI satellite images"* was replaced by (L63):

*"A combination of satellite-based Alternative Floating Algae Index (AFAI, Wang and Hu, 2016) fields"*

**L66-68: Could be useful to include some references of the methodology here.**

The sentence "*None of them used predictive modelling, including classifiers, to determine the probability of a set of data belonging to another set in order to discover repeatable patterns, allowing to produce a decision for risk prevention managers.*" was changed to (L74):

*"None of them used predictive modelling (Geisser, 1993; Kuhn and Johnson, 2013) including classifiers (Friedl and Brodley, 1997) to determine the probability of repeatable patterns in a dataset so as to produce a decision for risk prevention managers."*

These references were included to improve the understanding.

*Friedl, M. A., and Brodley, C. E.: Decision tree classification of land cover from remotely sensed data, Remote Sensing of Environment, vol. 61, Issue 3, pp. 399-409, ISSN:0034-4257, https://doi.org/10.1016/S0034-4257(97)00049-7, 1997.*

*Geisser, S.: Predictive Inference: An Introduction, Chapman & Hall, ISBN: 978-0-412-03471-8, 1993.*

*Kuhn, M., and Johnson, K.: Applied predictive modeling. New York, Springer. isbn:978-1-4614-6848-6, doi:10.1007/978-1-4614-6849-3, 2013.*

**L69: A general definition of predictive modelling is missing in the introduction for the readers which do not know about this method and how it compares with a conventional forecast. For example could be included here (Line 69).**

The following sentences were added here L76:

*"Predictive modelling refers to mathematical and computational methods of predicting future events based on the analysis of the repeatable patterns in the input dataset (Geisser, 1993; Friedl and Brodley, 1997; Kuhn and Johnson, 2013). Compared to other conventional forecast, predictive modelling methods requiring low computational costs are characterized by their flexibility, and their intuitive simplicity (Friedl and Brodley, 1997)."*

**L75-76: "To optimize the final partitioning, an additional metric based on the Kullback Leiber divergence (Kulback and Leibler, 1951, Biabiany et al., 2020) will be included" : quite specific on the methodology, for readers not familiarised with this method it could be hard to follow in this point in the introduction. More general details can be given, or this point can be moved to the methods section.**

Following your remark, this sentence was removed. The Expert Distance process with the Kullback Leiber divergence was explained in the method section 2.6.

**L82-83: "This ocean region corresponds to the CA and TA1 boxes in Johns (2020)", maybe say approximately corresponds, as not exactly the same. The LA3 region goes further south and LA2 and LA3 go until -55°E, whales region TA1 till -50°E. Most importantly, why choose the study regions to correspond to CA and TA1 boxes from Johns (2020)?**

The study regions were not chosen to correspond to CA and TA1 boxes from Johns (2020).

The sentence *"This ocean region corresponds to the CA and TA1 boxes in Johns (2020)."* Was removed.

**L96-96: From what I understand this dataset was not used before to simulate Sargassum trajectories, but was it used in any other Lagrangian study? Any validation studies done on the velocity outputs of this dataset?**

Firstly, there was a mistake about the HYCOM output resolution, we will correct it in the revised manuscript. The right version of HYCOM output used here is the HYCOM GLBy0.08 which has a grid resolution of 0.08 degree in longitude and 0.04 degree in latitude. To perform the present study, the native HYCOM fields have been preliminarily interpolated on the Mercator uniform lon/lat 0.08-degree grid with a bilinear method.

[revised manuscript text omitted]

**L101: "Comparison between HYCOM and Mercator results" Do you mean the results from the Sargassum trajectories or a comparison of the velocity outputs of these datasets?**

The sentence *"Comparison between HYCOM and Mercator results would help to better understand the effects of spatial resolution on surface current patterns in the focused region"* was removed in the revised manuscript.

**Section 2.3: What is the spatial and temporal resolution of the ERA-5 wind dataset?**

The ERA-5 wind dataset has a spatial resolution of 31 km and hourly fields are available.

Line 108, the part *"Surface wind data (at 1000 hPa) from the ERA-5 model for the time period 2019 to 2020 were integrated with Mercator currents following this formula:"*
was replaced (L133) by:

*"The daily 12 UTC surface wind data (at 1000 hPa) from the 31-km scale ERA-5 model were integrated with Mercator and HYCOM currents following this formula:"*

**L128: "Ward's method for HAC" Please explain and add reference.**

The sentence *"Besides the measures and the classes of distance between objects such as the Euclidean distance for K-means and the Ward's method for HAC, a new metric was also added (Biabiany et al. 2020)"*

was modified like below (L154)

*"Besides the measures and the classes of distance between objects such as the Euclidean distance for K-means and the Ward method which allows to identify homogeneous subsets of data (Ward, 1963), a new metric was also added."*

This reference was added

*J. H. Ward Jr. (1963) Hierarchical Grouping to Optimize an Objective Function, Journal of the American Statistical Association, 58:301, 236-244, DOI:10.1080/01621459.1963.10500845.*

**L129-130: "with its own expertise on the input data" What do you mean by these? Please provide further explanations. Also, the new method name is not specified at all in section 2.5.1, and it will help for the reader to better follow the methodology. This section is only 5 lines long, more details on the process of the clustering methods could be given.**

The sentence "*The result is an automated analysis with its own expertise on the input data.*" was removed.

Moreover to help the reader better following the methodology the section "2.5.1 Clustering methods process" was removed.

The paragraph *"Unsupervised learning methods such as Hierarchical Agglomerative Clustering (HAC) and K-means algorithms are used in the present study. Besides the measures and the classes of distance between objects such as the Euclidean distance for K-means and the Ward's method for HAC, a new metric was also added (Biabiany et al. 2020). This metric integrates a set of knowledge about the dynamics of the data to be partitioned as well as their spatio-temporal properties. The result is an automated analysis with its own expertise on the input data."*

Was changed to (L125): *"Unsupervised learning methods such as Hierarchical Agglomerative Clustering (HAC) and K-means algorithms were used in the present study. Besides the measures and the classes of distance between objects such as the Euclidean distance for K-means and the Ward method which allows to identify homogeneous subsets of data (Ward, 1963), a new metric was also added. The Expert Distance (ED) which integrates image analysis within unsupervised learning methods (Clustering) was used. This method allowed significant improvement in clustering analysis dealing with climate data characterized by high spatio-temporal variability, such as precipitation (Biabiany et al., 2020). Clustering methods using Euclidean distance (L2) can lead to group different physical situations within the same cluster (Biabiany et al., 2020). The ED metric integrates a set of knowledge about the dynamics of the data to be partitioned as well as their spatio-temporal properties."*

**L132: "L2 clustering methods…" Please explain L2 in this context.**

The sentences: "The ED metric, which seems more suitable for this study, was used. L2 clustering methods can lead, within the same cluster, to gatherings of different physical situations (Biabiany et al., 2020).

To remove these biases linked with L2 clustering, the first step of the method used here is to consider the spatial variability in the dynamics of the analyzed daily surface currents from L2."

were replaced by

*"Clustering methods using Euclidean distance (L2) can lead to group different physical situations within the same cluster (Biabiany et al., 2020)." (L158)*

**L133: "gatherings of different physical situations". What do you mean by this? Maybe give an example of physical situations for this particular study scenario.**

We clarified this part in the revised manuscript.

*"can lead, within the same cluster, to gatherings of different physical situations"* was replaced by *"can lead to group different physical situations within the same cluster."* (L158)

**You refer to this in the next phrase as "biases". Is there then a tendency towards a specific physical situation?**

There is no trend towards a specific physical situation. These side effects have been described in Biabiany et al. (2020).

As stated above, the sentence (L133) *"To remove these biases linked with L2 clustering, the first step of the method used here is to consider the spatial variability in the dynamics of the analyzed daily surface currents from L2."* was removed in the revised manuscript.

**L134: "spatial variability" : At what scales?**

This sentence was removed in the revised manuscript.

**L139-L140: "The analyzed daily fields include a total of 14 279 meshes (4 282 meshes in LA1, 3 407 meshes in LA2 and 4 536 meshes in LA3). The remainder corresponds to land areas." What do you refer to here with meshes?**

*"meshes"* were replaced by *"grid points"*

**The land areas then correspond to Sargassum strandings? For clarity, these details could be described in a dataset section better, rather than in the middle of the methods description.**

The land areas do not correspond to Sargassum strandings, they correspond to areas over land (e.g., islands).

To clarify this point, the sentence *"The remainder corresponds to land areas"* was replaced by *"The remainder corresponds to areas over land (e.g., islands)."*

**L141-142: "The second step was to group the information carried by the daily current velocity fields conditionally to the three given zones into histograms." More details on histograms, for example binning, velocity data from HYCOM and Mercator?**

Histogram bins are given in Table 1 and Fig. 7. The values correspond to deciles from HYCOM and Mercator datasets. This sentence was removed.

**L158: "optimal matching methods" Please explain and add some references.**

The words *"optimal matching methods"* was removed.

The sentence at L158 *"Dissimilarities between these backward sequences were calculated with optimal matching methods before dividing the population into several groups using a hierarchical classification (Larmarange et al., 2015)."*

was replaced (L190) by:

*Dissimilarities between these sequences were calculated before dividing the sequences dataset into several groups using a hierarchical classification (Larmarange et al., 2015).*

**L158: "dividing the population" what do you refer to exactly here by population? Population of strandings or backward sequences?**

To clarify this point, the words *"dividing the population"* were replaced by *"dividing the sequences dataset"*

**L160-162: Please give further details (maybe as supplementary material?) and add more references.**

*"Wald's algorithm"* was corrected to *"Ward algorithm".* It is already described (Line 155) as a *"method which allows to identify homogeneous subsets of data (Ward, 1963)".*

Ward, J. H. Jr.: Hierarchical Grouping to Optimize an Objective Function, Journal of the American Statistical Association, 58:301, 236-244, DOI: 10.1080/01621459.1963.10500845, 1963.

**L186-L187: "was experimented on the first 120 days…". Was experimented to…? Recall aim of doing these tests. Also why 120 days and during this period of time? Could results vary a lot if done during the northern hemisphere Summer months instead?**

To strengthen the performance evaluation, the testing period was extended from the first four months of 2021 (i.e., from January 2021 to April 2021) to the full year of 2021 including seasonal variations of the Sargassum offshore abundance.

The sentence *"The proposed tree in Fig. 2 was experimented on the first 120 days of the year 2021, from 1 st January 2021 to 30 April 2021, i.e., 120 tests."* was replaced by (L218)

*The proposed tree in Fig. 5 was tested on the full year of 2021 except 31 December 2021 including missing data, i.e. in total 364 tests.*

The original Fig. 2 will be replaced by the following figure:

[Figure]

Figure 5: (a) Scheme of the decision tree classifier to predict *Sargassum* beaching probability. (b) Combination base of oceanic current clusters labels obtained by KMS-ED from each beaching day to $\Delta t$ days before.

The Decision support system results figure will be replaced by the following figure:

[Figure]

Figure 18: Decision Support System (DSS) results: probability of beaching obtained per module. Daily probability to reach the maximum *Sargassum* abundance in the area 30-100 km offshore Guadeloupe for module A (blue line), beaching frequency per cluster for module C (orange line), match percentage for module D (yellow line), DSS Decision (black line). Days of observed beaching on Guadeloupe coasts (red dots): HYCOM (a) and Mercator (b).

**L190: Can maybe start section 3.1 giving some context on why this analysis is done.**

The following sentence was added L223 at the beginning of the section 3.1:

*"In view of the lack of study dealing with surface current patterns in the Lesser Antilles area, this preliminary analysis is presented here"*

**L191: "90% of them remain below 0.65 m/s". For both models exactly same?**

The sentence "*For both models HYCOM and Mercator, the velocity intensities do not exceed 2.57 m s-1 and 90% of them remain below 0.65 m s⁻¹"*

*was replaced by (L224)*

*"For both models HYCOM and Mercator, the maximum surface velocity is 2.57 m s-1 and 90% of them remain below 0.65 m s-1 (the respective 90th centile values are respectively 0.6515 m s-1 and 0.6458 m s-1 for HYCOM and Mercator)."*

**L193: Figure 3 distributions how are they calculated? With histograms? Kernel Density Estimator or something else applied to obtain this "smooth" distribution curves?**

After the sentence *"Figure 3 shows skewed distributions with skewness equal to 1.31 and 1.21."*

The following sentence was added (L229): "*A normal kernel was used to obtain these distributions."*

**L194-L195: 5 times greater for both models?**

The sentence *"There are extreme values indicating surface current speeds with 195 deviations 5 times greater than the standard deviation."* Was removed.

**L207-208: what are the implications of these differences?**

Firstly, the original figure was replaced by the following figure:

[Figure]

Figure 8: Comparison between Mercator and HYCOM surface currents from 2019 to 2020 on the same 0.08° grid: (a) median of magnitude absolute differences (Mercator-HYCOM) in m s$^{-1}$ and (b) median of magnitude relative differences (Mercator-HYCOM) in m s$^{-1}$ and (c) mode of current direction differences (Mercator-HYCOM) in degree.

The sentence *"The largest differences, above 0.5 m s-1 are observed in the South part of the LA arc, around Trinidad and Tobago."* was replaced L241 by the following sentence:

*"In the South part of the LA arc, around Trinidad and Tobago, Mercator current magnitudes are globally higher than HYCOM current magnitudes. Thus, Mercator surface currents might induce higher Sargassum influx from the Western Central Atlantic to the Caribbean Sea in this area."*

**L272-L273: "The monthly evolution of observed stranding days on the Guadeloupe coasts, the monthly evolution of Sargassum abundance over the Central Atlantic region (SaWS, https://optics.marine.usf.edu/projects/SaWS.html)" I imagine it should be: "Guadeloupe coasts and the monthly evolution…", to make clear you talking about two datasets. The observed stranding dataset is mentioned in the dataset section (section 2.4), but not the Sargassum abundance over the Central Atlantic region.**

Line 272, the sentences:

*"The monthly evolution of observed stranding days on the Guadeloupe coasts, the monthly evolution of Sargassum abundance over the Central Atlantic region (SaWS, https://optics.marine.usf.edu/projects/SaWS.html) were also analyzed on the focused period 2019-2020 (Figs. 11 and 12). During these two years, the amount of Sargassum over the Central Atlantic region increased significantly from February to July, then decreased from July to November."*

were replaced L304 by:

*"The monthly evolution of observed stranding days on the Guadeloupe coasts, the monthly evolution of Sargassum abundance in the area 30-100 km offshore Guadeloupe were also analyzed on the focused period 2019-2020 (Figs. 11 and 12). During these two years, the amount of Sargassum which may enhance the beaching risk in Guadeloupe increased significantly from February to May, then decreased from May to November."*

Figs. 11 and 12 were replaced by the following figures:

[Figure]

Figure 14: Monthly distribution of cluster occurrence from Mercator outputs, from 2019 to 2020, in the Lesser Antilles (55-66°W, 8-17°N): MC1 (a), MC2 (b), MC3 (c) and MC4 (d). The red line shows the monthly distribution of *Sargassum* beachings on the coasts of Guadeloupe during the same period. The blue line indicates the monthly evolution of *Sargassum* abundance in the area 30-100 km offshore Guadeloupe normalized on the maximum value.

[Figure]

Figure 15: Monthly distribution of cluster occurrence from HYCOM outputs, from 2019 to 2020, in the Lesser Antilles (55-66°W, 8-17°N): HC1 (a), HC2 (b), HC3 (c) and HC4 (d). The red line shows the monthly distribution of *Sargassum* beachings on the coasts of Guadeloupe during the same period. The blue line indicates the monthly evolution of *Sargassum* abundance in the area 30-100 km offshore Guadeloupe normalized on the maximum value.

**3. Technical corrections**

**Please write Sargassum in italics, like it is done in other studies like for example Johns et al., (2020), as you are writing its scientific name, and even if it is just the genus in this case.**

Following your suggestion, *"Sargassum"* was italicized in the revised manuscript.

**L10: "including windage effect": gives the impression the HYCOM and Mercator datasets already include the windage effect, when you actually added separately. Please improve phrasing.**

To clarify this point, the sentence:

*"The input surface currents including windage effect were derived from the Mercator model and the Hybrid Coordinate Ocean Model (HYCOM)."*

was replaced by (L9)

*"The input surface currents were derived from the Mercator model and the Hybrid Coordinate Ocean Model (HYCOM) outputs in which we integrated the windage effect."*

**L20: "LA received…" to "The LA received…"**

*"LA received"* will be replaced by *"The LA received"*

**L23: "…were also be observed…" to "…were also observed…"**

This sentence was removed.

**L46: Improve sentence, e.g. "… multi-year reanalysis of wind and current, and numerical models, both the role of subsurface nutrient supply and surface current transport were estimated."**

As you suggested the part:

*"...multi-year reanalysis of wind and current, numerical models estimated both the role of subsurface nutrient supply and surface current transport."*

Was replaced by

*"...multi-year reanalysis of wind and current, and numerical models, both the role of subsurface nutrient supply and surface current transport were estimated."*

**L50: "Sargassum Watch System SaWS" to "*Sargassum* Watch System (SaWS)"**

*"Sargassum Watch System SaWS"* was replaced by *"Sargassum Watch System (SaWS)"*

**L83: "in Johns (2020)" et al. missing.**

This sentence was removed.

**L92: Please define the abbreviations HYCOM and NCODA (HYCOM defined in abstract but not in the main text)**

The part:

*2.1 HYCOM surface current dataset*

*"Fine scale surface current data from the 1/25-degree HYCOM + NCODA Gulf of Mexico analysis model (GOMu0.04/expt_90.1m000 version, Hogan et al, 2014; Helber et al., 2013; Cummings and Smedstad, 2013; Cummings, 2005) between 1st January 2019 (i.e., available data starting date) and 31 December 2020 were analyzed. Daily 12Z fields giving the u and v components of the current at 50 cm depth were used. These fine resolution current data were not used in previous studies dealing with Sargassum hazard (Putman et al., 2018; Johns et al., 2020).*

was replaced by:

*2.1 HYCOM surface current dataset*

*Daily 12 UTC (i.e. Coordinated Universal Time) surface current components from the 41-layer Hybrid Coordinate Ocean Model (HYCOM) global 1/12-degree analysis (HYCOM GLBy0.08 version), were examined. The HYCOM surface forcing including 10-m wind velocities are extracted from Climate Forecast System Version 2 (CFSv2). The Navy Coupled Ocean Data Assimilation (NCODA) system is used to assimilate available observational data: satellite altimeter sea surface height, satellite and in-situ sea surface temperature, temperature vertical profiles and salinity vertical profiles (Cummings, 2005; Cummings and Smedstad, 2013; Helber et al., 2013). The Bathymetry used is the GEBCO8 (Becker et al., 2009) with 30 arc second of resolution. The HYCOM GLBy0.08 grid resolution is 0.08 degree in longitude and 0.04 degree in latitude. To perform the present study, the native HYCOM fields were preliminarily interpolated on the Mercator uniform lon/lat 0.08-degree grid with a bilinear method. Putman et al. (2018) and Johns et al. (2020) used a previous version of HYCOM model including uniform lon/lat 0.08° scale grid to successfully simulate Sargassum trajectories.*

**L94: Please define 12Z fields.**

*"12Z"* was replaced by *"12 UTC (i.e., Coordinated Universal Time) "*

**L94-95: "u and v components" to "zonal (u) and meridional (v) velocity components"**

This part was removed.

**L101-102: "Comparison…in the focused region" to "A comparison.. in the study region."**

This sentence *"Comparison between HYCOM and Mercator results would help to better understand the effects of spatial resolution on surface current patterns in the focused region."* was removed in the revised manuscript.

**L107: "Sargassum raft transport", maybe trajectories instead of transport is more appropriate?**

Following your suggestion the part *"Lagrangian simulations of Sargassum rafts transport in the Caribbean region."*

was replaced by

*"Lagrangian simulations of Sargassum raft trajectories in the Caribbean region."*

**L112-113: "The region analyzed in the present work corresponds to the CA - TA1 region defined in Johns et al. (2020)" already mentioned in L82-83, is it necessary to repeat here?**

This sentence was removed.

**L116-117: "This period includes 730 observational days with 110 days of observed strandings." , phrasing not clear do you mean that out of the total 730 days of data, only 110 days included observations of Sargassum strandings?**

During the two years 2019-2020, only 110 days of observed beachings in Guadeloupe have been recorded.

To clarify this point,

The following sentence:

*This period includes 730 observational days with 110 days of observed strandings."*

Was replaced by (L140)

*During this period of 730 days, only 110 days of observed beaching were recorded (i.e. 30 days in 2019 and 80 days in 2020). During the year of 2021, 78 days of beaching were observed in Guadeloupe.*

**L137: "above Barbados island" to "above the island of Barbados"**

*"above Barbados island"* was replaced by *"above the island of Barbados"*

**L142: "The similarity of the most similar fields is estimated per pair.." Improve phrasing. What do you refer to exactly? Per pair of Sargassum meshes?**

Line 141-150 the two paragraphs,

*"The second step was to group the information carried by the daily current velocity fields conditionally to the three given zones into histograms. The similarity of the most similar fields is estimated per pair and per zone based on the symmetrized Kullback-Leibler (KL) divergence computed from the histograms (Kullback and Leibler, 1951). This allows the entropy between two distributions to be expressed without having a priori reasoning concerning the probability distribution. The similarity between two histograms was quantified this way. The last step consisted in calculating the average of the divergence*

*values for each zone. This allows to have a single value, named Expert Distance (ED) quantifying the similarity between the individuals of the database during clustering."*

Was replaced by (L153):

*"Unsupervised learning methods such as Hierarchical Agglomerative Clustering (HAC) and K-means algorithms were used in the present study. Besides the measures and the classes of distance between objects such as the Euclidean distance for K-means and the Ward method which allows to identify homogeneous subsets of data (Ward, 1963), a new metric was also added. The Expert Distance (ED) which integrates image analysis within unsupervised learning methods (Clustering) was used. This method allowed significant improvement in clustering analysis dealing with climate data characterized by high spatio-temporal variability, such as precipitation (Biabiany et al., 2020). Clustering methods using Euclidean distance (L2) can lead to group different physical situations within the same cluster (Biabiany et al., 2020). The ED metric integrates a set of knowledge about the dynamics of the data to be partitioned as well as their spatio-temporal properties. "*

To clarify this part, the following schematic will be added in the methods section.

[Figure]

Figure 3: The schematic of the Expert Distance process.

**L148: "The SaMk index" to "The Silhouette (SaMk) index"**

*"The SaMk index"* was replaced by *"The Silhouette (SaMk) index"*

**L151: Define all variables of equation 2!**

This sentence was added below the equation (2) :

where k is the number of clusters, Cj the set of days from the cluster j, i a day form Cj and s(i) the silhouette index (Rousseeuw, 1987) value of day.

**L153-154: Improve phrasing.**

The sentence: *"To better understand current regime dynamics which may lead to Sargassum strandings on the coasts of Guadeloupe, the past stranding 30-day current backward sequences were analyzed."*

was replaced by (L182)

*"To better understand current dynamics which may lead to Sargassum beaching in Guadeloupe, we analyzed the 30-day current sequences before beaching."*

**L156: "January 2020" to "January 2019"**

*"January 2020"* was replaced by *"January 2019"*

**L165-L166: " surface currents with windage effects (Mercator, HYCOM and ERA-5)" to " surface currents (Mercator and HYCOM) with windage effects (ERA-5)"**

*" surface currents with windage effects (Mercator, HYCOM and ERA-5)"* was replaced (L197) by *"surface currents (Mercator and HYCOM) with windage effects (ERA-5)"*

**L186-L187: "The proposed tree in Fig. 2…". Move to new line, to separate it from the phrase explaining the terms in equation (4)**

The sentence starting with *"The proposed tree in Fig. 2…"* was moved to new line.

L191: **"do not exceed 2.57 m/s". Maybe better to say the maximum is 2.57 m/s, if not it sounds like 2.57 m/s is a key velocity value that should not be exceeded for some reason.**

*" For both models HYCOM and Mercator, the velocity magnitudes do not exceed 2.57 m s$^{-1}$ and 90% of them remain below 0.65 m s$^{-1}$."*

was replaced L224 by

*"For both models HYCOM and Mercator, the maximum surface velocity is 2.57 m s-1 and 90% of them remain below 0.65 m s-1"*

**L193-L194: add at end to which model it each value corresponds to e.g. ".. for HYCOM and Mercator, respectively."**

*"Figure 3 shows skewed distributions with skewness equal to 1.31 and 1.21"*

was replaced by

*"Figure 6 shows skewed distributions with skewness equal to 1.31 and 1.21 for HYCOM and Mercator, respectively."*

**L205: "Globally, at sea, the current.." Is it necessary to specify at sea? What do you exactly mean with at sea here, open ocean?**

*"at sea"* was replaced by *"at open sea"*

**L210: "into three magnitude groups of 45°" to "into three magnitude groups of 45° intervals"?**

*"into three magnitude groups of 45°"* was replaced by *"into three magnitude groups of 45° intervals"*

**L215: Improve phrasing, gives the impression you used equation (1) to perform the clustering.**

the part *"according to equation (1)"* will be removed.

**L244: "Table 3 shows results" to "Table 3 shows the results"**

*"Table 3 shows results"* was replaced by *"Table 3 shows the results"*

**L297: "remain with probabilities" add probabilities of… Help the reader follow better your study, recalling details**

This part was modified with new results produced by our improved version of Decision Support System. Please find below the modified Table 6 which includes recalling details.

| Time range around D (day) | Datasets | True positive (recall %) | True negative (recall %) | Accuracy (ratio %) |
|---|---|---|---|---|
| **0** | **HYCOM** | **46 (59.0%)** | 151 (52.8%) | **197 (54.1%)** |
| | **Mercator** | **43 (55.1%)** | 141 (49.3%) | **184 (50.6%)** |
| **+/- 1** | **HYCOM** | 52 (66.7%) | 175 (61.2%) | 227 (62.4%) |
| | **Mercator** | 47 (60.3%) | 151 (52.8%) | 198 (54.4%) |
| **+/- 2** | **HYCOM** | 55 (70.5%) | 189 (66.1%) | 244 (67.0%) |
| | **Mercator** | 51 (65.4%) | 155 (54.2%) | 206 (56.6%) |
| **+/- 3** | **HYCOM** | **57 (73.1%)** | 198 (69.2%) | **255 (70.1%)** |
| | **Mercator** | **51 (65.4%)** | 161 (56.3%) | **212 (58.2%)** |

Table 6: Decision tree performance scores.

**L317: Improve wording of Section 4.2 title, for example can simply remove "hazard"**

"Hazard" was removed.

**L320: "retroflexion" to "retroflection"**

"retroflexion" was replaced by *"retroflection"*

**L345 "The first peak of strandings, in March and seems.." to "The first peak of strandings, in March, seems.."**

*"The first peak of strandings, in March and seems.."* was replaced by *"The first peak of strandings, in March, seems.."*

**L373: Write as K-Means, and also in L217, write method in the same way.**

*"k-mean"* was replaced by *"K-Means"* everywhere in the revised manuscript."

**4. Figures and tables**

**Figure 2: Describe BASE abbreviation as in L175.**

*"BASE"* is not an abbreviation, to avoid misunderstanding, the term was written in the normal case (i.e., Base or base). Base was explained both in the Fig. 2a and in the caption.

[Figure]

Figure 5: (a) Scheme of the decision tree classifier to predict *Sargassum* beaching probability. (b) Combination base of oceanic current clusters labels obtained by KMS-ED from each beaching day to $\Delta t$ days before.

**Figures 4, 9 and 10: x-axis tick labels not clear, please improve.**

The x-axis tick labels were improved like below.

Figures 4, 9 and 10 were replaced by the following figures.

[Figure]

Figure 7: Relative frequency distribution of current speeds for the three offshore sub-regions around the Lesser Antilles (2019-2020), LA1 (blue), LA2 (red), LA3 (yellow). (a) Mercator with ERA-5 windage and (b) HYCOM with ERA-5 windage.

[Figure]

Figure 12: Relative frequency distribution of current speeds for the three offshore sub-regions: MC1 (a), MC2 (b), MC3 (c) and MC4 (d). The representative elements were obtained after KMS-ED clustering for Mercator.

[Figure]

Figure 13: Relative frequency distribution of current speeds for the three offshore sub-regions: HC1 (a), HC2 (b), HC3 (c) and HC4 (d). The representative elements were obtained after KMS-ED clustering for HYCOM.

**Table 1: Header mean to Mean**

*"mean"* was replaced by *"Mean"*

| Deciles (D$_i$) | 0.1 | 0.2 | 0.3 | 0.4 | 0.5 | 0.6 | 0.7 | 0.8 | 0.9 | Max | Mean | Sigma |
|---|---|---|---|---|---|---|---|---|---|---|---|---|
| Mercator (m s$^{-1}$) | 0.11 | 0.16 | 0.20 | 0.24 | 0.28 | 0.32 | 0.39 | 0.48 | 0.65 | 2.57 | 0.33 | 0.22 |
| HYCOM (m s$^{-1}$) | 0.13 | 0.18 | 0.23 | 0.28 | 0.32 | 0.38 | 0.44 | 0.52 | 0.65 | 2.49 | 0.36 | 0.21 |

Table 1: Boundaries of the histogram classes used to quantify surface currents velocity data with Sigma as Standard deviation.

**Table 5: Caption mention what n and % refer to exactly.**

Table 5 was modified as shown below:

| 30-day current sequence before beaching (HYCOM) | Seq1 | Seq2 | Seq3 | Seq4 |
|---|---|---|---|---|
| n | 18 | 40 | 7 | 42 |
| % | 16.8 | 37.4 | 6.5 | 39.3 |

Table 5: Distribution of sequence clusters, with (n) corresponding to the number of sequences in each cluster and (%) corresponding to the ratio of the number of sequences on the total.

**Answers to CC1**

**1. Throughout the text: *"Sargassum"* is the genus name of the pelagic, brown algae discussed. Accordingly, it should be italicized wherever used.**

Following your suggestion, *"Sargassum"* was italicized everywhere in the revised manuscript.

**2. Lines 79-80: Citing Putman et al. 2018 (already cited elsewhere) would be appropriate here as they model the % of *Sargassum* that follows these routes.**

Following your suggestion "Putman et al. 2018" was cited here.

The following sentence was added (L191):

*"Putman et al. (2018) modelled the percentage of Sargassum which follows these routes."*

**3. Lines 91-96: I am confused what HYCOM output you are using. What is reported here (GOMu0.04/expt_90.1m000 version) appears to only extend from latitude 18N to 32N and is thus outside of the area of this study. Can you please clarify? Did you run your own HYCOM at 1/25 degree resolution? The Global Analysis of HYCOM uses a grid of 0.04 degree longitude and 0.08 degree latitude, is this what you actually used?**

This was a mistake, we will correct it in the revised manuscript. The right version of HYCOM output used here is the HYCOM GLBy0.08 which has a grid resolution of 0.08 degree in longitude and 0.04 degree in latitude. To perform the present study, the native HYCOM fields have been preliminarily interpolated on the Mercator uniform lon/lat 0.08-degree grid with a bilinear method.

Line 91-96 :

*"**2.1 HYCOM surface current dataset***

*"Fine scale surface current data from the 1/25-degree HYCOM + NCODA Gulf of Mexico analysis model (GOMu0.04/expt_90.1m000 version, Hogan et al, 2014; Helber et al., 2013; Cummings and Smedstad, 2013; Cummings, 2005) between 1st January 2019 (i.e., available data starting date) and 31 December 2020 were analyzed. Daily 12Z fields giving the u and v components of the current at 50 cm depth were used. These fine resolution current data were not used in previous studies dealing with Sargassum hazard (Putman et al., 2018; Johns et al., 2020)."*

will be replaced by:

*2.1 HYCOM surface current dataset*

*"Daily 12 UTC (i.e. Coordinated Universal Time) surface current components from the 41-layer Hybrid Coordinate Ocean Model (HYCOM) global 1/12-degree analysis (HYCOM GLBy0.08 version), were examined. The HYCOM surface forcing including 10-m wind velocities are extracted from Climate Forecast System Version 2 (CFSv2). The Navy Coupled Ocean Data Assimilation (NCODA) system is used to assimilate available observational data: satellite altimeter sea surface height, satellite and in-situ sea surface temperature, temperature vertical profiles and salinity vertical profiles (Cummings, 2005; Cummings and Smedstad, 2013; Helber et al., 2013). The Bathymetry used is the GEBCO8 (Becker et al., 2009) with 30 arc second of resolution. The HYCOM GLBy0.08 grid resolution is 0.08 degree in longitude and 0.04 degree in latitude. To perform the present study, the native HYCOM fields were preliminarily interpolated on the Mercator uniform lon/lat 0.08-degree grid with a bilinear method. Putman et al. (2018) and Johns et al. (2020) used a previous version of HYCOM model including uniform lon/lat 0.08° scale grid to successfully simulate Sargassum trajectories."*

**4. Line 101-103: See also, Putman NF & He R (2013) Tracking the long-distance dispersal of marine organisms: sensitivity to ocean model resolution. Journal of the Royal Society Interface, 10:20120979**

We thank you for this suggestion.

**5. Line 104: I am confused, what is the basis for assuming the "optimal factors of Cw = 0.01"? Surely this is not the case based on data from Johns et al. 2020, which showed no evidence that a windage factor of 1% was appropriate for *Sargassum*. They simply picked the "reasonable" value that has been used in the earlier publication Putman et al. 2018. The value of 1% was chosen by Putman et al. 2018 to test the sensitivity of model predictions to windage and did not claim that it was optimal (or even somewhat correct). Work since that point has been conducted which seems to suggest that the situation is somewhat more complicated, see Putman et al. 2020 (already cited elsewhere) and Johnson, D.R., Franks, J.S., Oxenford, H.A. and Cox, S.A.L., 2020. Pelagic *Sargassum* Prediction and Marine Connectivity in the Tropical Atlantic. Gulf and Caribbean Research, 31(1), pp.GCFI20-GCFI30. Whether the best windage value is 0, 0.5%, 1%, 3% or something else likely depends on the oceanographic region and the ocean circulation model and wind product used.**

We agree with this remark,

the part Lines 104-108:

"*Surface wind influences the transport of floating seaweed rafts, with an optimal factor of $C_w$ = 0.01, which corresponds to the drag coefficient or windage, following Johns et al. (2020). A first clustering (KMS-L2) on Mercator analysis without windage had been proposed by Bernard et al. (2019). Berline et al. (2017), Putman et al. (2018) and Johns et al. (2020) have shown that the windage improves the Lagrangian simulations of Sargassum rafts transport in the Caribbean region. The windage was included in the present surface current clustering.*"

was replaced L131 by:

"*Surface wind influences the transport of floating seaweed rafts and a drag or windage coefficient must be added to the surface currents. The value of $C_w$ = 0.01 was used by Putman et al. (2018), Johns et al. (2020) and Berline et al. (2020). The use of other windage values should be investigated in a further study.*"

**6. Line 112: I think that "Putman et al. (2016)" should be "Putman et al. (2018)"**

"Putman et al. (2016)" was corrected to *"Putman et al. (2018)"*

**7. Line 206: change to "current speed differences are relatively small…"**

"*current speed differences are small*"

was replaced by

"*current speed differences are relatively small*"

**8. Line 334: change to "…due to the North Equatorial Current…"**

Line 334: *"The last identified factor is related to surface currents present in the North Atlantic region due to the North Current and the associated gyre circulation."*

was replaced by

*The last identified factor is related to the North Atlantic Gyre and the associated North Equatorial Current."*

**9. Lines 360-361: Another issue may be that ocean current patterns may be highly important for "non-beaching" events (e.g., the currents are directed so that material doesn't reach the island), but for Sargassum to beach there needs to be Sargassum present. Thus, currents might be in a state to transport material to the island, but if there is no Sargassum present, there can be no**

**beaching. Am I correct that this predictive model is based only on circulation/wind and not Sargassum abundance/coverage/distribution?**

Following your suggestion,

To improve the predictive model originally based on circulation/wind/past-beachings the module A producing the monthly probability of beaching was replaced by a new module based on satellite observations which produces the daily probability to reach the maximum observed cumulative floating algae density in an area of 100 km radius offshore Guadeloupe.

Moreover, to strengthen the performance evaluation, the testing period was extended from the first four months of 2021 (i.e., from January 2021 to April 2021) to the full year of 2021 including seasonal variations of the offshore *Sargassum* abundance.

The performance evaluation of the classifier was also extended by adding three temporal uncertainty ranges around the decision day, respectively: +/-1 day, +/-2 days, +/-3 days.

The following section was added in the "Datasets and method" section (L145):

*2.5 Satellite-based offshore abundance of Sargassum*

[revised manuscript text omitted]

**10. Lines 400-406: You may wish to draw reader's attention to the fact that there is considerable interest in monitoring and predicting coastal inundation by *Sargassum*. For instance, you may note how your smaller-scale study's goals might enhance the region-wide efforts such as the *Sargassum* Inundation Reports (SIR) discussed here:**

**Trinanes J, Putman NF, Goni G, Hu C, Wang M (2021) Monitoring pelagic *Sargassum* inundation potential for coastal communities. Journal of Operational Oceanography 14, in press (published online).**

Thank you for this useful suggestion.

"Trinanes et al. (2021)" was added to the references.

The following part was added in the Introduction section (Line 68):

*Trinanes et al. (2021) presented the Sargassum Inundation Reports (SIR), a product based on satellite observations to weekly predict Sargassum coastal inundation potential throughout the Caribbean Sea region, the Gulf of Mexico, and extending to the east coast of Florida and the Bahamas. As described by Trinanes et al. (2021), the SIR algorithm uses the Floating Algae density values within 50 km of each coastal pixel to predict three inundation potential levels (low, medium, and high). This algorithm does not include ocean currents, winds, and waves which may modify the movement of Sargassum.*

The following sentence was added in the Conclusion Line 454:

*"Like the Sargassum Inundation Reports (Trinanes et al., 2021), the present small-scale Sargassum beaching predictive model may contribute to the region-wide efforts to help coastal communities managing this hazard."*

---

## Referee Report (RR1)

**RC2 round 2**

Please find below, in green, the answers to round 2 of revisions. If no answer is provided, it implies agreement with your response.

1. General comments
Thank you very much for adding the new schematics to better understand the processing you used in this study.

2. Specific comments

**L69: A general definition of predictive modelling is missing in the introduction for the readers which do not know about this method and how it compares with a conventional forecast. For example could be included here (Line 69).**

The following sentences were added here L76:

"Predictive modelling refers to mathematical and computational methods of predicting future events based on the analysis of the repeatable patterns in the input dataset (Geisser, 1993; Friedl and Brodley, 1997; Kuhn and Johnson, 2013). Compared to other conventional forecast, predictive modelling methods requiring low computational costs are characterized by their flexibility, and their intuitive simplicity (Friedl and Brodley, 1997)."

" Compared to other conventional forecast…" to "Compared to other conventional forecast**s… "**

"predictive modelling methods requiring low computational costs are characterized by their flexibility" to "predictive modelling methods requir**e** low computational costs **and?** are characterized by their flexibility" add "and" or only the ones with low computational cost?

**L96-96: From what I understand this dataset was not used before to simulate Sargassum trajectories, but was it used in any other Lagrangian study? Any validation studies done on the velocity outputs of this dataset?**

Firstly, there was a mistake about the HYCOM output resolution, we will correct it in the revised manuscript. The right version of HYCOM output used here is the HYCOM GLBy0.08 which has a grid resolution of 0.08 degree in longitude and 0.04 degree in latitude. To perform the present study, the native HYCOM fields have been preliminarily interpolated on the Mercator uniform lon/lat 0.08-degree grid with a bilinear method.

[revised manuscript text omitted]

Section 2.2 suggested modification (in bold the specific changes made, and underlined phrases to be revised):

2.2 Mercator surface current dataset

The daily **(12 UTC)** surface current components from the 50-layer PSY4V3R1 Mercator 1/12-degree 3D analysis system (Lellouche et al., 2018; Gasparin et al., 2019) were also analyzed. The atmospheric surface forcing **is** extracted from the 3-hourly ECMWF (European Centre for Medium-Range Weather Forecasts) IFS (Integrated Forecast System). Assimilated observational data types are quite similar to HYCOM model. Unlike the HYCOM GLBy0.08 native grid including higher resolution in latitude (i.e. 0.04 degree), the Mercator native grid is uniform in longitude and latitude with 0.08-degree **horizontal grid resolution**. This would suggest that HYCOM may better reproduce small scale patterns than Mercator. Moreover, as described by Lellouche et al. (2018), the Mercator bathymetry includes GEBCO8 data in regions shallower than 200 m and the coarse 1 arc-minute ETOPO1 data (Amante and Eakins, 2009) in regions deeper than 300 m. The complex bathymetry of the Lesser Antilles Arc studied here could be less realistic **in the Mercator than in the HYCOM fields.**

"Assimilated observational data types are quite similar to HYCOM model." This phrase is not very clear, with the assimilated observational data types are you referring to the Mercator dataset? If so, please specify whether the HYCOM and Mercator datasets include data assimilation.

**Section 2.3: What is the spatial and temporal resolution of the ERA-5 wind dataset?**

The ERA-5 wind dataset has a spatial resolution of 31 km and hourly fields are available.

Line 108, the part "Surface wind data (at 1000 hPa) from the ERA-5 model for the time period 2019 to 2020 were integrated with Mercator currents following this formula:" was replaced (L133) by:

"The daily 12 UTC surface wind data (at 1000 hPa) from the 31-km scale ERA-5 model were integrated with Mercator and HYCOM currents following this formula:"

The above phrase and section 2.3 suggested to be re-writed for clarity in the following way:

"Surface wind influences the transport of floating seaweed rafts and a drag or windage coefficient must be added to the surface currents. The value of $Cw$ = 0.01 was used by Putman et al. (2018), Johns et al. (2020) and Berline et al. (2020). The use of other windage values should be investigated in a further study. The daily 12 UTC surface wind data (at 1000 hPa) from the 31-km scale ERA-5 model were integrated with Mercator and HYCOM currents following this formula:
$us(x, t) = um(x, t) + Cwuw(x, t)$ (1)
where $us$ represents the oceanic surface currents with windage, $um$ the oceanic surface currents velocity, $Cw$ the windage and $uw$ the surface winds velocity. This approach is consistent with Putman et al. (2018) and Johns et al. (2020) studies."

To:

"Surface wind influences the transport of floating seaweed rafts and a drag or windage coefficient must be added to the surface currents. Daily (12 UTC) from the **31-km horizontal resolution** ERA-5 model was used. The wind data was integrated with Mercator and HYCOM **ocean currents data** following this formula:
$us(x, t) = um(x, t) + Cwuw(x, t)$ (1)
where $us$ represents the oceanic surface currents with windage, $um$ the oceanic surface currents velocity, $Cw$ the windage and $uw$ the surface winds velocity. This approach is consistent with Putman et al. (2018) and Johns et al. (2020) studies. The value of $Cw$ = 0.01 was used, **following** Putman et al. (2018), Johns et al. (2020) and Berline et al. (2020). The use of other windage values should be investigated in a further study."

Information on the ERA5 data still missing: is it the reanalysis dataset? Any references of the dataset? Is the temporal resolution daily or hourly and just the 12 UTC fields used?

Lastly, please add here or in the Data availability section where this data was downloaded or obtained from.  Also, this information is missing for the HYCOM and Mercator datasets (sections 2.1 and 2.2).

**L128: "Ward's method for HAC" Please explain and add reference.**
The sentence "Besides the measures and the classes of distance between objects such as the Euclidean distance for K-means and the Ward's method for HAC, a new metric was also added (Biabiany et al. 2020)"

was modified like below (L154)
"Besides the measures and the classes of distance between objects such as the Euclidean distance for K-means and the Ward method which allows to identify homogeneous subsets of data (Ward, 1963), a new metric was also added."

Please clarify phrasing, maybe split sentence in 2 or 3?

**L186-L187: "was experimented on the first 120 days…". Was experimented to…? Recall aim of doing these tests. Also why 120 days and during this period of time? Could results vary a lot if done during the northern hemisphere Summer months instead?**

To strengthen the performance evaluation, the testing period was extended from the first four months of 2021 (i.e., from January 2021 to April 2021) to the full year of 2021 including seasonal variations of the Sargassum offshore abundance.

The sentence "The proposed tree in Fig. 2 was experimented on the first 120 days of the year 2021, from 1st January 2021 to 30 April 2021, i.e., 120 tests." was replaced by (L218)

The proposed tree in Fig. 5 was tested on the full year of 2021 except 31 December 2021 including missing data, i.e. in total 364 tests.

Phrase not clear, do you mean the 31$^{st}$ of December was not included because of missing data?

**L190: Can maybe start section 3.1 giving some context on why this analysis is done.**

The following sentence was added L223 at the beginning of the section 3.1:

"In view of the lack of study dealing with surface current patterns in the Lesser Antilles area, this preliminary analysis is presented here"

"the lack of study dealing" to "the lack of studies dealing"

**L191: "90% of them remain below 0.65 m/s". For both models exactly same?**

The sentence "For both models HYCOM and Mercator, the velocity intensities do not exceed 2.57 m s-1 and 90% of them remain below 0.65 m s-1"
was replaced by (L224)

"For both models HYCOM and Mercator, the maximum surface velocity is 2.57 m s-1 and 90% of them remain below 0.65 m s-1 (the respective 90th centile values are respectively 0.6515 m s-1 and 0.6458 m s-1 for HYCOM and Mercator)."

"90% of them remain" to "90% of the velocity values remain"

**L193: Figure 3 distributions how are they calculated? With histograms? Kernel Density Estimator or something else applied to obtain this "smooth" distribution curves?**

After the sentence "Figure 3 shows skewed distributions with skewness equal to 1.31 and 1.21." The following sentence was added (L229): "A normal kernel was used to obtain these distributions."

Thanks for the clarification, maybe to clarify further modify "A normal kernel was used to obtain these distributions." to "A **Gaussian** kernel was **applied** to obtain these distributions."

**L207-208: what are the implications of these differences?**

Firstly, the original figure was replaced by the following figure:

Figure 8: Comparison between Mercator and HYCOM surface currents from 2019 to 2020 on the same 0.08° grid: (a) median of magnitude absolute differences (Mercator-HYCOM) in m s-1 and (b) median of magnitude relative differences (Mercator-HYCOM) in m s-1 and (c) mode of current direction differences (Mercator-HYCOM) in degree.

For subplot c), colormap a bit confusing as the white regions suggest not difference in direction, but those regions are actually different. Maybe the same colormap as subplot a) would be more appropriate?

**L272-L273: "The monthly evolution of observed stranding days on the Guadeloupe coasts, the monthly evolution of Sargassum abundance over the Central Atlantic region (SaWS, https://optics.marine.usf.edu/projects/SaWS.html)" I imagine it should be: "Guadeloupe coasts and the monthly evolution...", to make clear you talking about two datasets. The observed stranding dataset is mentioned in the dataset section (section 2.4), but not the Sargassum abundance over the Central Atlantic region.**

Line 272, the sentences:
"The monthly evolution of observed stranding days on the Guadeloupe coasts, the monthly evolution of Sargassum abundance over the Central Atlantic region (SaWS, https://optics.marine.usf.edu/projects/SaWS.html) were also analyzed on the focused period 2019-2020 (Figs. 11 and 12). During these two years, the amount of Sargassum over the Central Atlantic region increased significantly from February to July, then decreased from July to November."

were replaced L304 by:
"The monthly evolution of observed stranding days on the Guadeloupe coasts, the monthly evolution of Sargassum abundance in the area 30-100 km offshore Guadeloupe were also analyzed on the focused period 2019-2020 (Figs. 11 and 12). During these two years, the amount of Sargassum which may enhance the beaching risk in Guadeloupe increased significantly from February to May, then decreased from May to November."

I think my remark here was not clarified. Please re-check.

**3. Technical corrections**

Some *Sargassum* in the Conclusions section forgot to put in italics.

**L46: Improve sentence, e.g. "… multi-year reanalysis of wind and current, and numerical models, both the role of subsurface nutrient supply and surface current transport were estimated."**

As you suggested the part:
"...multi-year reanalysis of wind and current, numerical models estimated both the role of subsurface nutrient supply and surface current transport."
Was replaced by
"...multi-year reanalysis of wind and current, and numerical models, both the role of subsurface nutrient supply and surface current transport were estimated."

L47-48: "current, and numerical models, both the roles of both subsurface nutrient supply and surface current transport were estimated." to "current, and numerical models; the roles of both subsurface nutrient supply and surface current transport were estimated."

**L116-117: "This period includes 730 observational days with 110 days of observed strandings." , phrasing not clear do you mean that out of the total 730 days of data, only 110 daysincluded observations of Sargassum strandings?**

During the two years 2019-2020, only 110 days of observed beachings in Guadeloupe have been recorded.

To clarify this point,

The following sentence:
This period includes 730 observational days with 110 days of observed strandings."

Was replaced by (L140)

During this period of 730 days, only 110 days of observed beaching were recorded (i.e. 30 days in 2019 and 80 days in 2020). During the year of 2021, 78 days of beaching were observed in Guadeloupe.

"During this period of 730 days, only 110 days of observed beaching were recorded" to "During this period of 730 days, only 110 days of *Sargassum* beaching were recorded"

**L151: Define all variables of equation 2!**

This sentence was added below the equation (2) :

where k is the number of clusters, Cj the set of days from the cluster j, i a day form Cj and s(i) the silhouette index (Rousseeuw, 1987) value of day.

"…value of day." to "..value of day **i**".

**L191: "do not exceed 2.57 m/s". Maybe better to say the maximum is 2.57 m/s, if not it sounds like 2.57 m/s is a key velocity value that should not be exceeded for some reason.**

" For both models HYCOM and Mercator, the velocity magnitudes do not exceed 2.57 m s-1 and 90% of them remain below 0.65 m s-1."
was replaced L224 by

"For both models HYCOM and Mercator, the maximum surface velocity is 2.57 m s-1 and 90% of them remain below 0.65 m s-1"

For the maximum velocity of 2.57 m/s for both models, is it exactly the same too? If so, why does in figure 7 the x-axis go up to 2.57 m/s for Mercator, but only 2.49 m/s for HYCOM?

**L297: "remain with probabilities" add probabilities of… Help the reader follow better your study, recalling details**

This part was modified with new results produced by our improved version of Decision Support System.
Please find below the modified Table 6 which includes recalling details.

Table 6: Decision tree performance scores.

Please add details to the table caption.  True/ negative positive of… and recall % referring to… Accuracy of… and ratio between… .

Moreover, in section 3.5 you first introduce table 6, but further details are still missing.  I assume that with true positive you mean beaching and true negative non-beaching days? Please specify this.  Moreover, the values in table 6 for true positive is the number of days beaching is observed and the percentage, the percentage of days with beaching observed? The corresponding for non-beaching and true negative?  Also, how is the accuracy calculated? And the ratio?

**4. Figures and tables**

Figure 2: "The schematic of the overall methodology." to "A schematic showing the overall methodology."

Figure 4: "The schematic of the clustering process on the current sequences leading to beachings." to "The schematic of the clustering process **used** on the **ocean** current sequences leading to beachings."  See whether best to say used on, applied on, or other.

---

## Author Response (AR2)

**Dear Editor,**

**Dear referees,**

We thank you for your minor and technical comments which helped us to improve the quality of the paper.

Please find below our answers to your remarks (in blue). The proposed changes in the text are marked in red.

**Reply to Reviewer 1**

Fig XX The schematic of the adopted methodology."
To clarify, I suggest using a generic 'circulation model' instead of repeating MERCATOR and HYCOM in each panel

Our answer:

Following your suggestion, the schematic of the adopted methodology was modified as below:

[Figure]

*Figure 2: A schematic showing the overall methodology.*

"Figure XX: The schematic of the Expert Distance process."

Beware of typos :

distribution.

KL 'divergence' rather than convergence I guess.

Our answer:

Following your remark, this schematic was modified: "convergence" was replaced by "divergence".

[Figure]

Figure 3: The schematic of the Expert Distance process.

Fig 11
As the monthly distribution of cluster MC1 and MC2 differ from HC1 and HC2, you should mention that the cluster number differ

Our answer:

To clarify this point, the following sentence was added L284:

*"The cluster numbering does not take into account these match percentages (e.g. MC1 and MC2 main patterns respectively differ from HC1 and HC2 patterns)."*

The caption of Fig 11 was also modified as below:

*"Figure 11: Representative elements of the clusters from HYCOM current data combined with ERA-5 windage (KMS-ED method with k = 4): HC1 (day 29-04-2019) (a), HC2 (day 06-01-2020) (b), HC3 (day 04-05-2020) (c), HC4 (day 11-11-2019) (d). The HYCOM clusters numbering differs from the Mercator clusters numbering."*

**Reply to Reviewer 2**

" Compared to other conventional forecast…" to "Compared to other conventional forecast**s… "**

"predictive modelling methods requiring low computational costs are characterized by their flexibility" to "predictive modelling methods requir**e** low computational costs **and?** are characterized by their flexibility" add "and" or only the ones with low computational cost?

Our answer :

Following your remarks,

L78: *" Compared to other conventional forecast…"* was replaced to *"Compared to other conventional forecasts… "*

*"predictive modelling methods requiring low computational costs are characterized"* was replaced by *"predictive modelling methods require low computational costs and are characterized"*

Section 2.1 suggested modification (in bold the specific changes made):

2.1 HYCOM surface current dataset

Daily **(12 UTC, i.e. Coordinated Universal Time)** surface current components from the 41-layer Hybrid Coordinate Ocean Model (HYCOM**) at 1/12-degree, global analysis** (HYCOM GLBy0.08 version), were examined. The HYCOM surface forcing including 10-m wind velocities are extracted from Climate Forecast System Version 2 (CFSv2). The Navy Coupled Ocean Data Assimilation (NCODA) system is used to assimilate available observational data: satellite altimeter sea surface height, satellite and in situ sea surface temperature, temperature vertical profiles and salinity vertical profiles (Cummings, 2005; Cummings and Smedstad, 2013; Helber et al., 2013). Th**e ba**thymetry used is the GEBCO8 (Becker et al., 2009) with 30 arc second of resolution. The HYCOM GLBy0.08 grid resolution is 0.08 degree in longitude and 0.04 degree in latitude. To perform the present study, the native HYCOM fields were **first** interpolated on the Mercator uniform lon/lat 0.08-degree grid with a bilinear method. Putman et al. (2018) and Johns et al. (2020) used a previous version of HYCOM model including uniform lon/lat 0.08° scale grid to successfully simulate *Sargassum* trajectories.

Section 2.2 suggested modification (in bold the specific changes made, and underlined phrases to be revised):

2.2 Mercator surface current dataset

The daily **(12 UTC)** surface current components from the 50-layer PSY4V3R1 Mercator 1/12-degree 3D analysis system (Lellouche et al., 2018; Gasparin et al., 2019) were also analyzed. The atmospheric surface forcing **is** extracted from the 3-hourly ECMWF (European Centre for Medium-Range Weather Forecasts) IFS (Integrated Forecast System). Assimilated observational data types are quite similar to HYCOM model. Unlike the HYCOM GLBy0.08 native grid including higher resolution in latitude (i.e. 0.04 degree), the Mercator native grid is uniform in longitude and latitude with 0.08-degree **horizontal grid resolution**. This would suggest that HYCOM may better reproduce small scale patterns than Mercator. Moreover, as described by Lellouche et al. (2018), the Mercator bathymetry includes GEBCO8 data in regions shallower than 200 m and the coarse 1 arc-minute ETOPO1 data (Amante and Eakins, 2009) in regions deeper than 300 m. The complex bathymetry of the Lesser Antilles Arc studied here could be less realistic **in the Mercator than in the HYCOM fields.**

Our answer: following your remarks, these paragraphs were modified as below (specific changes in red)

**2.1 HYCOM surface current dataset**

Daily (12 UTC, i.e. Coordinated Universal Time) surface current components from the 41-layer Hybrid Coordinate Ocean Model (HYCOM) at 1/12-degree, global analysis (HYCOM GLBy0.08 version, available at: https://www.hycom.org/data/glby0pt08/expt-93pt0, last access: 17 January 2022), were examined. The HYCOM surface forcing including 10-m wind velocities are extracted from Climate Forecast System Version 2 (CFSv2). The Navy Coupled Ocean Data Assimilation (NCODA) system is used to assimilate available observational data: satellite altimeter sea surface height, satellite and in-situ sea surface temperature, temperature vertical profiles and salinity vertical profiles (Cummings, 2005; Cummings and Smedstad, 2013;

Helber et al., 2013). The bathymetry used is the GEBCO8 (Becker et al., 2009) with 30 arc second of resolution. The HYCOM GLBy0.08 grid resolution is 0.08 degree in longitude and 0.04 degree in latitude. To perform the present study, the native HYCOM fields were first interpolated on the Mercator uniform lon/lat 0.08-degree grid with a bilinear method. Putman et al. (2018) and Johns et al. (2020) used a previous version of HYCOM model including uniform lon/lat 0.08° scale grid to successfully simulate *Sargassum* trajectories.

**2.2 Mercator surface current dataset**

The daily (12 UTC) surface current components from the 50-layer PSY4V3R1 Mercator 1/12-degree 3D analysis system (Lellouche et al., 2018; Gasparin et al., 2019) were also analyzed (available at: https://resources.marine.copernicus.eu/product-detail/GLOBAL_ANALYSIS_FORECAST_PHY_001_024/DATA-ACCESS, last access: 17 January 2022). The atmospheric surface forcing is extracted from the 3-hourly ECMWF (European Centre for Medium-Range Weather Forecasts) IFS (Integrated Forecast System). This version of Mercator model includes assimilation of observational data quite similarly to HYCOM NCODA system (i.e. satellite altimeter sea surface height, satellite and in situ sea surface temperature, temperature vertical profiles and salinity vertical profiles). Unlike the HYCOM GLBy0.08 native grid including higher resolution in latitude (i.e. 0.04 degree), the Mercator native grid is uniform in longitude and latitude with 0.08-degree horizontal grid resolution. This would suggest that HYCOM may better reproduce small scale patterns than Mercator. Moreover, as described by Lellouche et al. (2018), the Mercator bathymetry includes GEBCO8 data in regions shallower than 200 m and the coarse 1 arc-minute ETOPO1 data (Amante and Eakins, 2009) in regions deeper than 300 m. The complex bathymetry of the Lesser Antilles Arc studied here could be less realistic in the Mercator than in the HYCOM fields.

"Assimilated observational data types are quite similar to HYCOM model." This phrase is not very clear, with the assimilated observational data types are you referring to the Mercator dataset? If so, please specify whether the HYCOM and Mercator datasets include data assimilation.

Our answer:

To clarify this point, the sentence *"Assimilated observational data types are quite similar to HYCOM model."* was replaced by *"This version of Mercator model includes assimilation of observational data quite similarly to HYCOM NCODA system (i.e. satellite altimeter sea surface height, satellite and in situ sea surface temperature, temperature vertical profiles and salinity vertical profiles)."*

The above phrase and section 2.3 suggested to be re-writed for clarity in the following way:

[revised manuscript text omitted]

Phrase not clear, do you mean the 31st of December was not included because of missing data?

Our answer:

This sentence was replaced L223 by *"The proposed tree in Fig. 5 was tested on the full year of 2021 except 31 December 2021 which was not included because of missing data, giving a total of 364 tests"*

"the lack of study dealing" to "the lack of studies dealing"

Our answer:

Following your remark, *"the lack of study dealing"* was replaced by *"the lack of studies dealing"* (L228)

"90% of them remain" to "90% of the velocity values remain"

Our answer:

Following your remark, *"90% of them remain"* was replaced by *"90% of the velocity values remain"* (L230)

Thanks for the clarification, maybe to clarify further modify "A normal kernel was used to obtain these distributions." to "A **Gaussian** kernel was **applied** to obtain these distributions."

Our answer:

Following your remark, *"A normal kernel was used to obtain these distributions."* was replaced by *"A Gaussian kernel was applied to obtain these distributions."* (L234)

Figure 8: Comparison between Mercator and HYCOM surface currents from 2019 to 2020 on the same 0.08° grid: (a) median of magnitude absolute differences (Mercator-HYCOM) in m s-1 and (b) median of magnitude relative differences (Mercator-HYCOM) in m s-1 and (c) mode of current direction differences (Mercator-HYCOM) in degree.

For subplot c), colormap a bit confusing as the white regions suggest not difference in direction, but those regions are actually different. Maybe the same colormap as subplot a) would be more appropriate?

Our answer:

Following your remark, the colormap of the subplot c (Fig 8) was modified as below.

[Figure]

**Figure 8: Comparison between Mercator and HYCOM surface currents from 2019 to 2020 on the same 0.08° grid: (a) median of magnitude absolute differences (Mercator-HYCOM) in m s⁻¹ and (b) median of magnitude relative differences (Mercator-HYCOM) in m s⁻¹ and (c) mode of current direction differences (Mercator-HYCOM) in degree.**

**L272-L273: "The monthly evolution of observed stranding days on the Guadeloupe coasts, the monthly evolution of Sargassum abundance over the Central Atlantic region**

**(SaWS, https://optics.marine.usf.edu/projects/SaWS.html)" I imagine it should be: "Guadeloupe coasts and the monthly evolution…", to make clear you talking about two datasets. The observed stranding dataset is mentioned in the dataset section (section 2.4), but not the Sargassum abundance over the Central Atlantic region.**

Line 272, the sentences:

"The monthly evolution of observed stranding days on the Guadeloupe coasts, the monthly evolution of Sargassum abundance over the Central Atlantic region (SaWS, https://optics.marine.usf.edu/projects/SaWS.html) were also analyzed on the focused period 2019-2020 (Figs. 11 and 12). During these two years, the amount of Sargassum over the Central Atlantic region increased significantly from February to July, then decreased from July to November."

were replaced L304 by:

"The monthly evolution of observed stranding days on the Guadeloupe coasts, the monthly evolution of Sargassum abundance in the area 30-100 km offshore Guadeloupe were also analyzed on the focused period 2019-2020 (Figs. 11 and 12). During these two years, the amount of Sargassum which may enhance the

beaching risk in Guadeloupe increased significantly from February to May, then decreased from May to November."

I think my remark here was not clarified. Please re-check.

Our answer:

The observed stranding dataset is mentioned in the dataset section (section 2.4), and the Sargassum abundance in the dataset section (section 2.5).

L150

*"2.5 Satellite-based offshore abundance of Sargassum*

*Sargassum satellite observations were included in the present decision support system. To quantify the abundance of Sargassum in an area of 100 km radius offshore Guadeloupe, the 7-day Floating Algae (FA) density fields derived from the Alternative Floating Algae Index (Wang and Hu, 2016) were analyzed. As described by Trinanes et al. (2021), the 7-day Floating Algae (FA) density fields are accumulated on 7 days and have a 0.1° resolution. Due to optical complexity in nearshore waters, the FA density fields are masked with missing values within 30 km from shoreline (Trinanes et al. 2021). The cumulative FA density values were added up in the area 30-100 km offshore Guadeloupe (Fig. 1) then averaged over the two years 2019 and 2020 for each day."*

**3. Technical corrections**

Some *Sargassum* in the Conclusions section forgot to put in italics.

Our answer: Following your suggestion, these *"Sargassum"* were italicized in the conclusion section.

L47-48: "current, and numerical models, both the roles of both subsurface nutrient supply and surface current transport were estimated." to "current, and numerical models; the roles of both subsurface nutrient supply and surface current transport were estimated."

Our answer:

Following your remark, *"current, and numerical models, both the roles of both subsurface nutrient supply and surface current transport were estimated."* was replaced by *"current, and numerical models; the roles of both subsurface nutrient supply and surface current transport were estimated."* (L47)

"During this period of 730 days, only 110 days of observed beaching were recorded" to "During this period of 730 days, only 110 days of *Sargassum* beaching were recorded"

Our answer:

Following your remark, *"During this period of 730 days, only 110 days of observed beaching were recorded"* was replaced by *"During this period of 730 days, only 110 days of Sargassum beaching were recorded"* (L145)

**L151: Define all variables of equation 2!**
This sentence was added below the equation (2) :
where k is the number of clusters, Cj the set of days from the cluster j, i a day form Cj and s(i) the silhouette index (Rousseeuw, 1987) value of day.
"…value of day." to "..value of day i".

Our answer:

Following your remark, *"…value of day."* was replaced by *"..value of day i".* (L180)

For the maximum velocity of 2.57 m/s for both models, is it exactly the same too? If so, why does in figure 7 the x-axis go up to 2.57 m/s for Mercator, but only 2.49 m/s for HYCOM?

Our answer:
The sentence *"For both models HYCOM and Mercator, the maximum surface velocity is 2.57 m s-1 and 90% of them remain below 0.65 m s-1"* was replaced by:

*"The maximum surface velocity reaches 2.49 m s$^{-1}$ and 2.57 m s$^{-1}$, respectively for HYCOM and Mercator. For both models 90% of the velocity values remain below 0.65 m s$^{-1}$ (the respective 90th centile values are respectively 0.6515 m s$^{-1}$ and 0.6458 m s$^{-1}$ for HYCOM and Mercator). "* (L229)

Table 6: Decision tree performance scores.
Please add details to the table caption. True/ negative positive of… and recall % referring to… Accuracy of… and ratio between… .

Our answer:
The caption of Table 6 was changed to:
*"Table 6: Decision tree performance scores: "True positive/negative" respectively refer to the number of observed beaching/non-beaching days predicted by the decision system; "Recall" refers to the ratio in percentage between these respective numbers of days and the total number of tests (i.e. 364 days); "Accuracy" corresponds to the number of days with a true prediction and its ratio in percentage over the total number of tested days."*

Moreover, in section 3.5 you first introduce table 6, but further details are still missing. I assume that with true positive you mean beaching and true negative non-beaching days? Please specify this. Moreover, the values in table 6 for true positive is the number of days beaching is observed and the percentage, the percentage of days with beaching observed? The corresponding for non-beaching and true negative? Also, how is the accuracy calculated? And the ratio?

Our answer:
Following your remark, these sentences were added in section 3.5 (L339):
*"True positive/negative" respectively refer to the number of observed beaching/non-beaching days, predicted by the decision system. "Recall" refers to the ratio in percentage between these respective numbers of days and the total number of tests (i.e. 364 days). "Accuracy" corresponds to the number of days with a true prediction and its ratio in percentage was computed over the total number of tested days."*

**4. Figures and tables**
Figure 2: "The schematic of the overall methodology." to "A schematic showing the overall methodology."

Our answer:

Following your remark, the Figure 2 caption was modified: *"The schematic of the overall methodology."* was replaced by *"A schematic showing the overall methodology."*

Figure 4: "The schematic of the clustering process on the current sequences leading to beachings." to "The schematic of the clustering process **used** on the **ocean** current sequences leading to beachings."

Our answer:
Following your remark, the Figure 4 caption was modified: *"The schematic of the clustering process on the current sequences leading to beachings."* was replaced by *"The schematic of the clustering process used on the ocean current sequences leading to beachings."*